# In situ structure of a bacterial flagellar motor at subnanometre resolution reveals adaptations for increased torque

Tina Drobnič [1,9,11], Eli J. Cohen[1,11], Thomas Calcraft [2], Mona Alzheimer [3], Kathrin Froschauer [3], Sarah Svensson [3,10], William H. Hoffmann [4], Nanki Singh[1], Sriram G. Garg[5], Louie D. Henderson [1], Trishant R. Umrekar [1], Andrea Nans [6], Deborah Ribardo [7], Francesco Pedaci [4], Ashley L. Nord [4], Georg K. A. Hochberg [5,8], David R. Hendrixson[7], Cynthia M. Sharma [3], Peter B. Rosenthal [2] & Morgan Beeby [1] ✉

The bacterial flagellar motor, which spins a helical propeller for propulsion, has undergone evolutionary diversification across bacterial species, often involving the addition of structures associated with increasing torque for motility in viscous environments. Understanding how such structures function and have evolved is hampered by challenges in visualizing motors in situ. Here we developed a *Campylobacter jejuni* minicell system for in situ cryogenic electron microscopy imaging and single-particle analysis of its motor, one of the most complex flagellar motors known, to subnanometre resolution. Focusing on the large periplasmic structures which are essential for increasing torque, our structural data, interpreted with molecular models, show that the basal disk comprises concentric rings of FlgP. The medial disk is a lattice of PflC with PflD, while the proximal disk is a rim of PflB attached to spokes of PflA. PflAB dimerization is essential for proximal disk assembly, recruiting FliL to scaffold more stator complexes at a wider radius which increases torque. We also acquired insights into universal principles of flagellar torque generation. This in situ approach is broadly applicable to other membrane-residing bacterial molecular machines.

How evolution innovates remains a fundamental question. While innovations in eukaryotes often arise by rewiring existing gene transcriptional netwoks[1], examples of the emergence of evolutionary novelty at the molecular-scale focus on small protein complexes[2–5]. What is needed is a comprehensive case study of a family of molecular machines that have diversified, so we can infer how their diversity evolved.

Bacterial flagella, helical propellers rotated by cell-envelope-embedded rotary motors, are icons of the evolution of molecular novelty[6] (Fig. 1a). The flagellum, best studied in model organisms *Salmonella enterica* serovar Typhimurium and *Escherichia coli*, is composed of a ring of inner-membrane motor proteins ('stator complexes')

that harness ion flux to rotate a large cytoplasmic rotor ring (the 'C-ring'). Torque is transmitted through a chassis (the 'MS-ring') and periplasm-spanning axial driveshaft (the 'rod') to an extracellular propeller structure that generates thrust. Structures of the C-ring[7–9], parts of stator complexes[10,11], MS-ring and rod[12,13], and axial structures[14,15] have recently been determined from purified subcomplexes.

Because flagella may pre-date the emergence of the bacteria, their age renders understanding how these core structures first evolved challenging. More tractable is the promise of understanding how recent diversifications evolved. Many flagellar motors have recruited additional proteins that scaffold more stator complexes than the ~11 seen in

model motors[16–18]. By increasing the number of stator complexes and increasing the radius at which they exert leverage to rotate the C-ring, these adaptations increase the magnitude of the motor's maximum torque output[17]: while *Salmonella* and *E. coli* motors deliver torque up to ~1,300 pN nm, spirochaete motors (with 16 stator complexes and substantially wider C-ring) deliver up to 4,000 pN nm[19], and Campylobacterota motors (with 17 or 18 stator complexes and wide C-ring) up to 3,600 pN nm[20]. Such adaptations probably facilitated species with high-torque motors to inhabit viscous environments such as mucous[21]. Understanding the architecture of these relatively recently evolved adaptations is a prerequisite to understanding how they evolved.

The Campylobacterota, a phylum that includes gastrointestinal pathogens *Campylobacter jejuni* and *Helicobacter pylori*[17,21], feature some of the most complex motors known, incorporating additional proteins that scaffold their wider ring of stator complexes to generate greater torque than *E. coli* and *Salmonella*[18]. Comparisons of diverse Campylobacterota motors and contextualization against the *C. jejuni* motor, the best-studied Campylobacterota motor, have suggested a stepwise evolutionary recruitment of these scaffolding proteins[18]. First, inner-membrane-associated periplasmic proteins (PflA and PflB, from 'paralyzed flagellum' proteins A and B) were recruited to template assembly of a wider ring of the stator complex motor proteins[17]. This wider ring could exert greater leverage on a correspondingly wider C-ring to increase the motor's torque, an adaptation that may have enabled lifestyles in the highly viscous gut mucous that many contemporary Campylobacterota species inhabit[17,21]. Independently, a large (~100-nm-wide) outer membrane-associated 'basal disk' of FlgP evolved[17,18,22], which we recently found is needed for buttressing the motor while the flagellar filament wraps and unwraps from the cell[23]. We have speculated that these structures were initially independent but subsequently became co-dependent for assembly[18]. The physical basis for this and the origins of these new proteins remain unclear.

Understanding how and why these additional proteins were incorporated into the motor and how they contribute to function requires molecular-scale models, but their size, intimate membrane association and multiple moving parts have hampered structural determination. Cryogenic electron microscopy (cryoEM) subtomogram average structures at 16–18 Å resolution[12,24] and structures of purified flagellar motors by single-particle analysis cryoEM lacking dynamic components[7–9,12,13] do not study the molecular architecture of the additional structures. Meanwhile advances in imaging membrane proteins in situ are most often in spherical liposomes rather than in their native context[25,26].

To address these challenges, we engineered *C. jejuni* minicells for subnanometre-resolution structure determination of the flagellar motor in situ. The quality of our map enabled us to assemble a molecular model of its evolutionary elaborations. Our model provides insights into the contributions of new proteins, reveals distant homologies, identifies previously unknown components, contextualizes adaptations of pre-existing core machinery and acquires observations informative of how flagellar motors generate rotation.

## Results

### Single-particle analysis of a flagellar motor in situ
To increase particle number for high-resolution structure determination, we exploited the multiply flagellated, still-motile minicell phenotype produced by a Δ*flhG* mutation[27], but removed the flagellar filament by deleting filament-encoding genes *flaA* and *flaB* for centrifugal purification (Fig. 1b,c). The resulting ~200-nm-diameter minicells are more homogeneous than ~350-nm-diameter minicells from Enterobacteriaceae such as *E. coli, Shigella* and *Salmonella*[28]. The polar curvature remains comparable to wildtype cells, meaning motor structures are unperturbed (Fig. 1b,c).

We acquired micrographs of minicells for in situ single-particle analysis. Initial two-dimensional (2D) classification revealed features

including the stator complexes and their periplasmic scaffold, basal disk, C-ring, MS-ring and rod (Fig. 1d). Classification and refinement applying the dominant C17 symmetry[17] yielded a reconstruction to 9.4 Å resolution by gold-standard Fourier shell correlation (FSC) using 32,790 particles (Fig. 1e and Extended Data Fig. 1). The majority of the periplasmic structures exhibited 17-fold symmetric features consistent with discrete proteins (Fig. 1e,f). The basal disk is composed of concentric rings, but while the first ring featured 17-fold symmetry, subsequent rings were cylindrical averages, indicating that they do not share the 17-fold symmetry of the other periplasmic structures. The symmetry-mismatched core structures were also cylindrical averages. Given that these symmetry-mismatched regions comprise universally conserved structures (such as the P-, L-, MS- and C-rings) and the stator complexes which have already been structurally characterized[7–13], we focused on *C. jejuni*'s relatively recently evolved 17-fold symmetric periplasmic scaffold structures. Focused refinement resolved this region to 7.9 Å (Fig. 2a,b, and Extended Data Figs. 1d–f and 2a–c), sufficient to resolve α-helices and β-sheets.

### The basal disk is composed of concentric rings of FlgP
The basal disk forms a concave cup that pushes the outer membrane away at greater disk radii (Fig. 2a). We focused on the first, 17-fold symmetric ring. These 17 repeats were composed of 3-repeat units, making a ring of 51 similar subunits in 17 trimeric repeats. Knowing that the disk is composed of FlgP[17,22], we predicted monomeric and multimeric structures of FlgP, excluding the N-terminal signal sequence and linker[22] to the membrane[23], using AlphaFold2 (ref. 29). Oligomers laterally associated to form a continuous β-sheet, with interaction of 1 FlgP with the next 2 protomers. Bending the arc of these oligomers demonstrated that 17 trimeric repeats, that is, 51 protomers, fit into the 51 periodic densities of the innermost ring of the basal disk, with a map-model FSC of 9.9 Å at FSC = 0.5 (Fig. 2a–d, and Extended Data Figs. 2d, 3a and 4a). Each trimeric repeat also featured an additional medial disk-facing density (Fig. 2a–c). Although we could not discern the symmetries of rings beyond the first, assembly of the disk is entirely reliant upon *flgP*[17]. Subsequent rings all have comparable cross-section densities, making them unlikely to be composed of another protein. On the basis of ratios of circumferences, we predict that each subsequent ring adds 11 protomers, explaining the symmetry mismatch with the 17-fold symmetric structures (Fig. 2e). Attempts at focused refinement of discrete rings failed, possibly due to insufficient signal from the 11.5 kDa FlgP subunits that lack prominent features to assist alignment.

FlgP features a modified SHS2 domain fold from InterPro family IPR024952, with a three-stranded β-sheet and an α-helix spanning one face[30] (Fig. 2f). Structural similarity searches revealed that this fold is also present in dodecin[30], *Helicobacter* Lpp20 (ref. 31), and γ-proteobacterial FlgT, a flagellar component from sodium-driven flagellar motors (Fig. 2f). FlgP, however, features an additional C-terminal helix and long β-hairpin insert between the SHS2 α-helix and the second SHS2 β-strand. This β-hairpin insert extends 35 Å at a ~42° angle to the vertical axis and is the basis of the continuous β-sheet in the inner face of each FlgP concentric ring.

We have previously found that FlgQ is required for basal disk assembly, so wondered whether it is also a basal disk component[22]. We predicted its structure using AlphaFold2 (ref. 29), finding that it resembles a 2-protomer FlgP repeat (Fig. 2g). To clarify its location, we fused an mCherry tag to FlgQ and determined a subtomogram average structure. Although the mutant was motile, the structure featured no additional densities (Fig. 2h). We conclude that FlgQ is a low-abundance or irregular component, or an assembly chaperone.

The basal disk is adjacent to the outer face of the P-ring. While ascertaining the symmetry of the L- and P-rings was beyond the resolution of our map, cross-sections through L- and P-ring densities showed comparable radii to *Salmonella*[12,13] (Fig. 2i). We cylindrically averaged our map by applying arbitrary high-order symmetry during focused

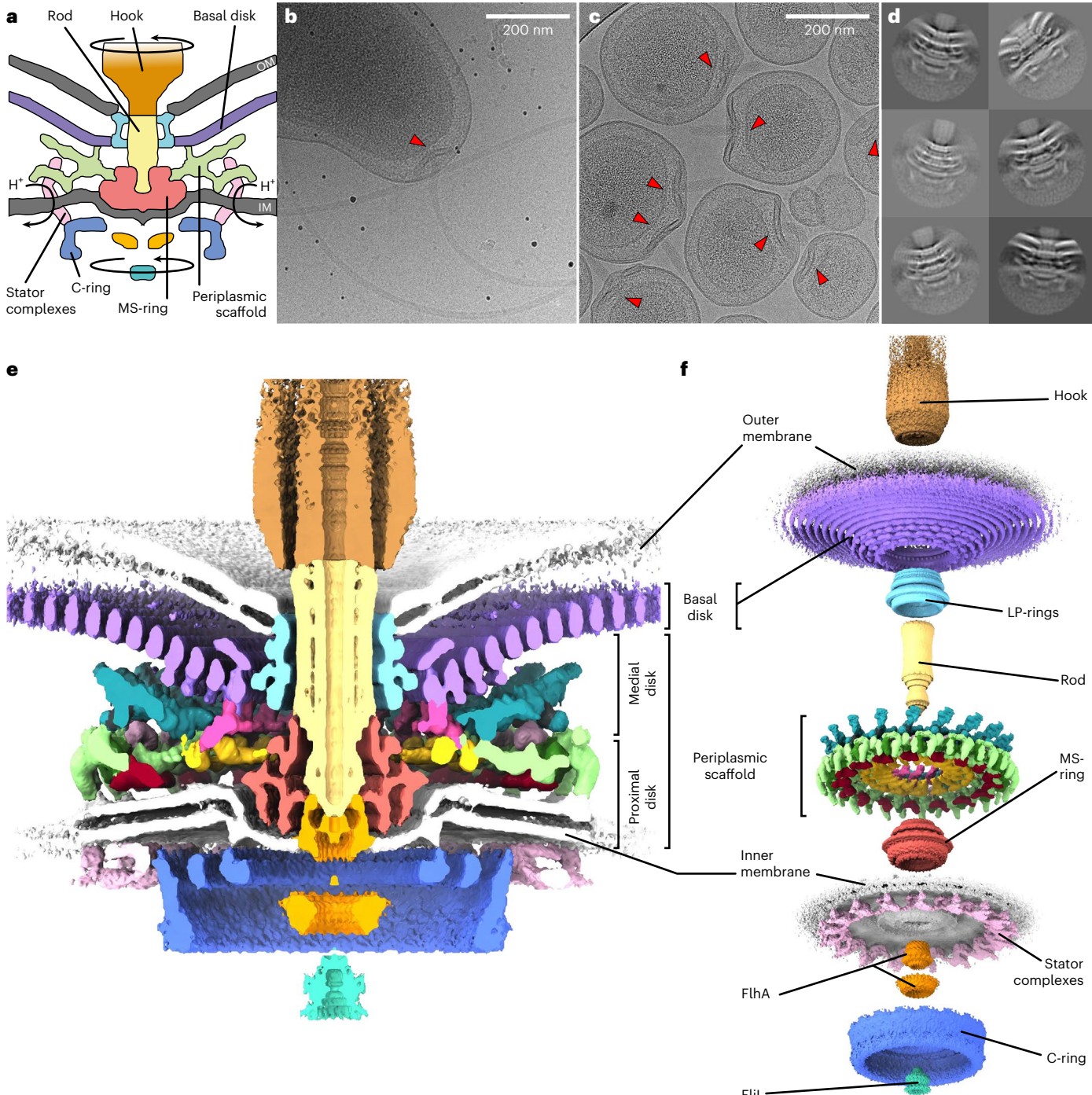

**Fig. 1 | Engineering of homogeneous *C. jejuni* minicells enabled determination of an in situ structure of a flagellar motor by single-particle analysis electron cryo-microscopy. a**, Schematic of the flagellar motor[6]. Proton flux through the stator complexes drives rotation of the C-ring, MS-ring, rod and hook/filament. In *C. jejuni* and other Campylobacterota, a basal disk and periplasmic scaffold have evolved that scaffold a wider ring of additional stator complexes thought to increase motor torque[17]. IM/OM, inner/outer membrane. **b**, Wildtype *C. jejuni* cells used in previous studies[17] typically provide 1 flagellar motor per field of view (arrowhead) as compared with **c**, many motors per field of view in

our minicell strain (arrowheads), greatly increasing throughput and reducing sample thickness for higher-quality electron micrograph acquisition. Note that curvature of minicells is comparable to wildtype cells. The 51781 raw micrographs are available from EMPIAR[90] (https://doi.org/10.6019/EMPIAR-11580). **d**, Periplasmic and cytoplasmic features are evident in single-particle analysis 2D classes of manually picked motors. **e**, Cross-section through an isosurface rendering of a C17 whole-motor 3D reconstruction (deposited as EMD-16723). **f**, Map from **e** segmented and exploded along the *z* axis to highlight component substructures.

refinement and compared these to low-pass filtered structures of similarly cylindrically averaged *Salmonella* LP-rings. The *C. jejuni* L- and P-rings have similar width to *Salmonella* (103% and 104% as wide, respectively) (Fig. 2i), and their sequences lack insertion and deletions, suggesting comparable stoichiometries. To experimentally

probe the *C. jejuni* LP-ring stoichiometries, we looked for steps in flagellar rotation−in *Salmonella* and *E. coli*, interactions between the asymmetric rod and 26-fold symmetric LP-rings impose 26 steps on flagellar rotation[12]. We attached beads to truncated flagella, and upon slowing rotation by de-energizing cells with carbonyl cyanide

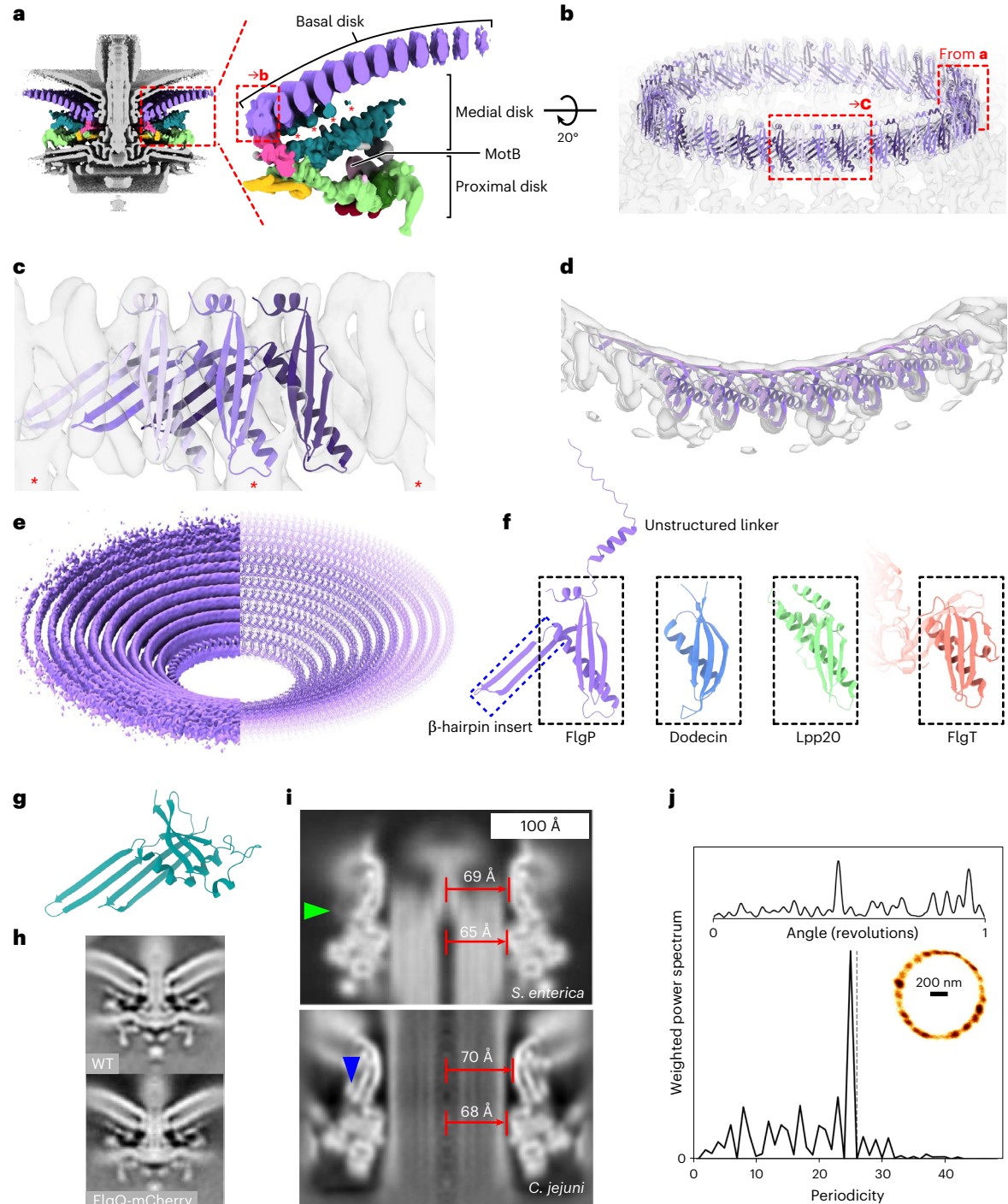

**Fig. 2 | The basal disk is composed of concentric rings of FlgP. a**, Left: focused C17 refinement of the periplasmic scaffold and inner ring of the multi-ring basal disk (deposited as EMD-16724) shows that the scaffold attaches to the innermost 5 concentric rings of the basal disk. Right: 1 asymmetric unit of the periplasmic scaffold. Only the innermost ring (dashed box) is 17-fold symmetric; subsequent rings were not part of the focused refinement. **b**, AlphaFold2 (ref. 29) structures of FlgP oligomers reveal that the innermost basal disk ring is composed of 51 FlgP monomers as 17 trimers. **c**, Fit of a FlgP trimer and interaction with the 17-fold symmetric density of the medial disk (asterisks). **d**, Top view of a fit of 7 FlgP monomers. **e**, Density of 10 concentric basal disk rings (left) and fitted FlgP models (right). **f**, The SHS2-like fold of FlgP (dashed box) is shared with dodecin[30] (PDBID: 1MOG), *Helicobacter pylori* Lpp20 (ref. 31) (PDBID: 5OK8) and *Vibrio alginolyticus* FlgT[49] (PDBID: 3W1E). FlgP uniquely features the ring-forming β-hairpin insert. **g**, AlphaFold2 (ref. 29) model of FlgQ structurally resembles a 2-FlgP repeat. **h**, 100 × 100 nm

slice through the subtomogram average structure of a motor in which FlgQ was fused to mCherry is indistinguishable from the WT motor (deposited as EMD-17419). **i**, The *C. jejuni* LP-rings have comparable diameters to the 26-fold symmetric *Salmonella* LP-rings. The density map of the *Salmonella* LP-rings[12] (EMD-12183) was low-pass filtered to 15 Å resolution and cylindrically averaged for comparison with the *C. jejuni* LP-rings (Additional data for EMD-16723). The *C. jejuni* L- and P-rings are of comparable diameters to those from *Salmonella*. Green arrowhead indicates *Salmonella* YecR, blue arrowhead an unidentified *C. jejuni* density. **j**, Support for the *C. jejuni* LP-rings having comparable stoichiometry to *E. coli* and *Salmonella* from ~26 steps in flagellar rotation. Top: kernel density estimation of bead position as a function of rotation. Bottom: weighted power spectrum of angular position; grey dashed line highlights 26. Inset: *x,y*-position histogram with density represented by darkness of coloration (Extended Data Fig. 5 shows 10 other traces).

3-chlorophenylhydrazone (CCCP), observed 25 to 27 phase-invariant dwell positions (Fig. 2j and Extended Data Fig. 5). Taking these structural and biophysical observations together, we conclude that the *C. jejuni* LP-rings have not expanded to match the symmetry of the basal disk or other more recently evolved components and have a symmetry similar to the 26-fold symmetry of the *Salmonella* LP-rings. Although our density maps show close association of the P-ring with the basal disk, we previously showed that the basal disk can assemble even if pushed out of axial register with the P-ring by increasing the length of the FlgP N-terminal linker[23]. Together with their 17:26 symmetric mismatch, this indicates that the basal disk and P-ring do not form a stable complex, despite their proximity.

We neither saw evidence of YecR, which in *Salmonella* forms a belt around the LP-rings[12], nor found a homologue in the *C. jejuni* genome, although an unidentified adjacent density hints at an as-yet-unidentified analogue (Fig. 2i).

## The medial disk is a lattice of PflC with PflD

We speculated that our map might help us understand the medial disk, a lattice between the basal and proximal disks of unknown composition (Fig. 3a). Our unpublished transposon insertion sequencing-based infection screen using *C. jejuni* NCTC11168 (ref. 32) revealed that polarly localized Cj1643 and Cj0892c are required for motility but not flagellar filament assembly, suggesting them as potential candidates. Co-immunoprecipitation with proximal disk component PflA in *C. jejuni* 81–176 recovered orthologues CJJ81176_1634 and CJJ81176_0901, respectively (Supplementary Table 1), which we renamed PflC and PflD (Paralyzed flagellum C and D) after subtomogram average structures revealed destablization of the proximal and medial disks upon deletion of *pflC*, and loss of a peripheral cage-like structure between the medial and proximal disks upon deletion of *pflD* (Fig. 3b).

PflC is a 364-residue periplasmic protein featuring an N-terminal serine protease domain followed by two PDZ-like domains. Generating an AlphaFold2 (ref. 29) model revealed 2 folding regions, $PflC_N$ (protease and PDZ1, residues 16–252) and $PflC_C$ (PDZ2, residues 265–364), connected by a proline-rich linker. By separately rigid-body docking $PflC_N$ and $PflC_C$, we located 17 copies of PflC in the densities projecting from each third FlgP subunit of the first basal disk ring, yielding a map-model FSC of 9.3 Å at FSC = 0.5 (Fig. 3c, and Extended Data Figs. 2e, 3a,b and 4b). $PflC_C$ docks to the underside of the first ring of FlgP, while $PflC_N$ forms the inner band of the medial disk; a density corresponding to the linker bridges adjacent PflC protomers, indicating that the $PflC_C$ domain of one chain assembles with the $PflC_N$ domain of another in a daisy chain of domain-swapped protomers (Fig. 3c). PDZ domains usually interface with binding partners via a hydrophobic pocket, but $PflC_C$ interactions are unlikely to be mediated by canonical binding[33] as its ligand-binding groove is oriented away from FlgP, lacks the conserved binding loop[34], and is predicted to be occupied by residues from its own chain.

The remainder of the medial disk is a lattice of α-helices that we could interpret with 17 asymmetric units of 6 additional $PflC_N$

protomers in a lattice (Fig. 3d and Extended Data Fig. 2f). $PflC_N$ protomers could be docked into our density map with a map-model FSC of 9.7 Å at FSC = 0.5 (Extended Data Figs. 3a,c and 4c). We refer to the subunits in one asymmetric unit as $PflC_{2–7}$; $PflC_{2–6}$ form radial spokes with a slight twist, while $PflC_7$ connects adjacent spokes. Pulldowns using FLAG-tagged PflC verified interactions with FlgP, while coIPs of FLAG-tagged proximal disk components PflA and PflB co-purified PflD (Fig. 3e). Supporting our modelling of PflC, we noted an additional rodlike density protruding from the location of N239 in all PflC protomers, which in a homolog from a closely related species is glycosylated with an N-linked heptasaccharide[35] (red asterisks, Fig. 3f).

The PflC lattice consists of diverse oligomerization interfaces (Fig. 3f). Three types of symmetric dimerization interface have rotational axes perpendicular to the plane of the lattice: one in $PflC_2$:$PflC_3$ and $PflC_4$:$PflC_5$; one in $PflC_3$:$PflC_4$ and $PflC_5$:$PflC_6$; and a third in $PflC_3$:$PflC_7$, although the latter is substantially distorted. The remaining interfaces ($PflC_1$:$PflC_1$, $PflC_1$:$PflC_2$, $PflC_2$:$PflC_7$ and $PflC_7$:$PflC_4$) are asymmetric and differ in each case. Based on our findings with $PflC_1$, the densities on the underside of the basal disk are probably the C-terminal PDZ domains of $PflC_{2–7}$ (asterisks in Fig. 2a, opaque teal in 3a), although the lack of C17 symmetry renders linker helices invisible.

These promiscuous self–self interfaces predicted that PflC oligomerizes ex situ. Curiously, however, size exclusion chromatography (Extended Data Fig. 6a,b) and mass photometry (MP) (Extended Data Fig. 6c) of purified PflC revealed mainly monomers. This suggested that the domains of discrete PflC polypeptides might self-associate until the context of the motor facilitates correct assembly. To test this, we removed $PflC_C$, which might inhibit oligomerization. Consistent with our speculation, $PflC_{Δ236–349}$ formed substantially more dimers than full-length PflC (Extended Data Fig. 6d–f). We speculate that interaction of $PflC_N$ and $PflC_C$ from the same polypeptide prevents cytoplasmic oligomerization, and binding to the basal disk facilitates medial disk assembly only in the context of the assembled motor. This would explain why we have never seen isolated PflC lattices in *C. jejuni* tomograms.

To better understand the origin of the PflC-based medial disk, which is absent from other Campylobacterota species, we performed a DALI search of PflC's predicted structure against the Protein Databank. Our search revealed that its protease:PDZ:PDZ domain architecture is similar to trypsin-like HtrA serine proteases[36] (Fig. 3g). Indeed, despite 20% and 27% sequence identity between the N- and C-terminal regions of PflC and HtrA, respectively, PSI-BLAST of PflC against the non-redundant protein sequence database returned hits to serine proteases from the first iteration, and a hidden Markov model derived from PSI-BLAST alignments returned four significant hits in the *C. jejuni* proteome, with the highest confidence to PflC and HtrA (e-values of $2.4 × 10^{-108}$ and $3 × 10^{-57}$, respectively; HtrA is CJJ81176_1242). The two other hits were also to proteases (CJJ81176_1086 and CJJ81176_0539, with e-values of $6.7 × 10^{-8}$ and

**Fig. 3 | The medial disk is a lattice of PflC and PflD that interacts with the basal and proximal disks. a**, Asymmetric unit of the periplasmic scaffold highlighting the medial disk. Dashed box regions are enlarged in **c**, **d** and **f**. **b**, Comparing the WT subtomogram average motor structure (100 × 100 nm cross-section of EMD-3150 (ref. 17)) to a pflC deletion (100 × 100 nm cross-section, deposited as EMD-17415) reveals loss of the medial disk (open arrowhead; filled on WT structure), while pflD deletion (deposited as EMD-17416) abolishes assembly of a peripheral post-like density (open arrowhead; filled on WT structure). **c**, Close-up below the innermost basal disk ring shows a ring of 17 domain-swapped PflC protomers attached to FlgP trimeric repeats. A single PflC (dashed outline) contributes domains to 2 protomeric units. Asterisks denote the interdomain linker. **d**, View of the medial disk from outside the cell depicting 17 asymmetric units (dashed box highlights 1 asymmetric unit) of 7 PflC protomers and 1 PflD. $PflC_1$ in pink; $PflC_{2,4,6}$ in teal; $PflC_{3,5}$ in cyan; $PflC_7$ in blue. Inset: differential oligomerization interfaces

of PflC protomers. Twofold symmetry axis symbols highlight symmetric dimerization; empty circles represent asymmetric interfaces. See focus in **f**. **e**, PflC and PflD interact with known flagellar components. Top: western blot of coIP of PflC-3×FLAG. Detected heavy (HC) and light (LC) antibody chains are indicated. C, culture; L, lysate. Middle: western blot of coIP of PflA-3×FLAG, PflD-sfGFP double-tagged strain. Sn1/2, supernatant1/2; W, wash; E, eluate. Bottom: western blot of coIP of PflB-3×FLAG, PflD-sfGFP double-tagged strain. **f**, Top: molecular model of the PflC lattice denoting symmetry elements, enlarged from red box in **d**. Densities adjacent to every Asn239 denoted by asterisks correspond to a glycosylation site of PflC from a related species. Symmetry elements as in **d**. Bottom: side view of PflD beneath $PflC_{4,5}$. **g**, The predicted structure of PflC (top) highlights common fold with HtrA (bottom), a periplasmic protease (PDB 6Z05 (ref. 36)). Left panel aligned to $PflC_1$ in **c**. Protease and 2 PDZ domains labelled for comparison. See Extended Data Figs. 6 and 7 for further analysis.

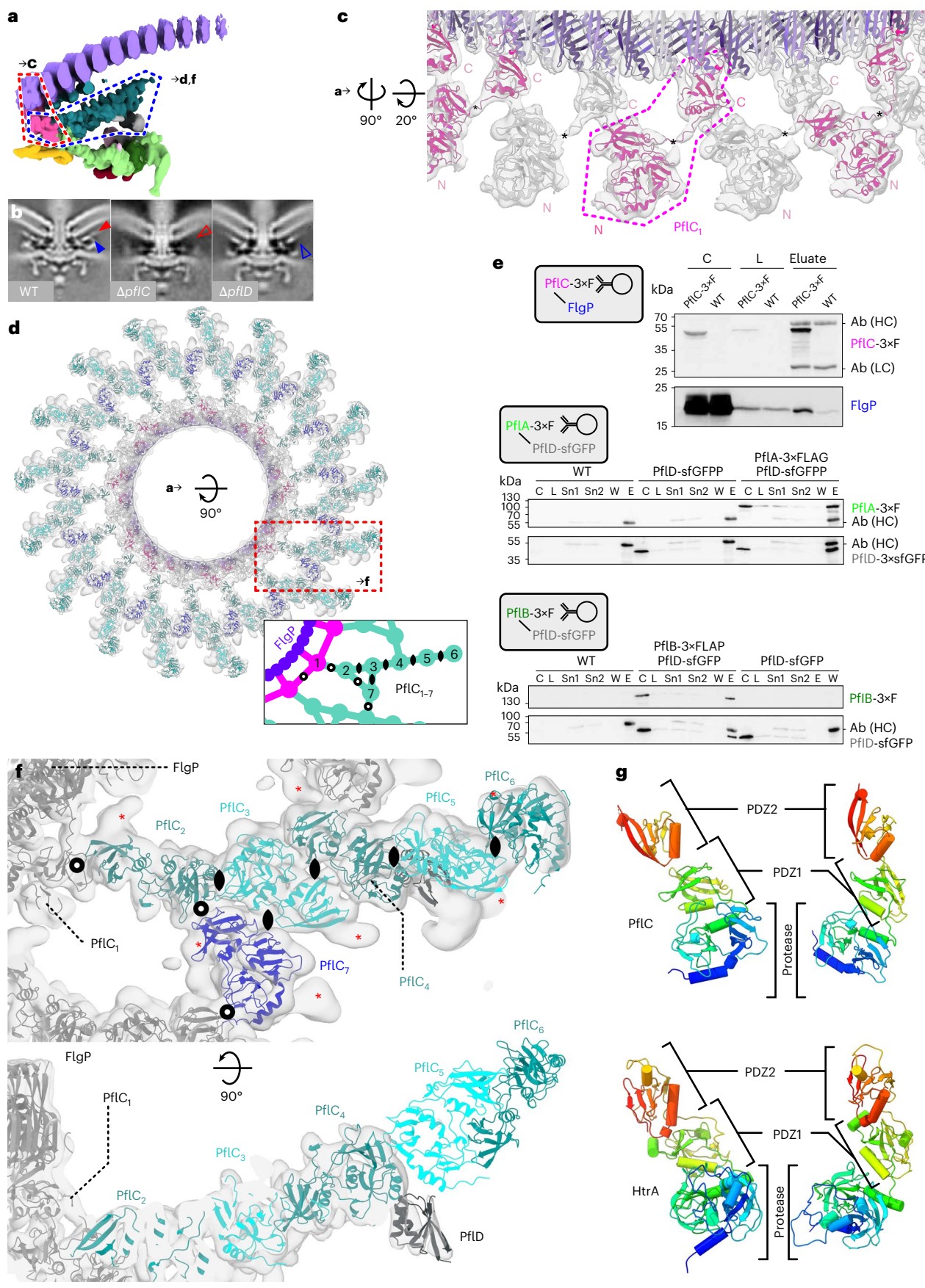

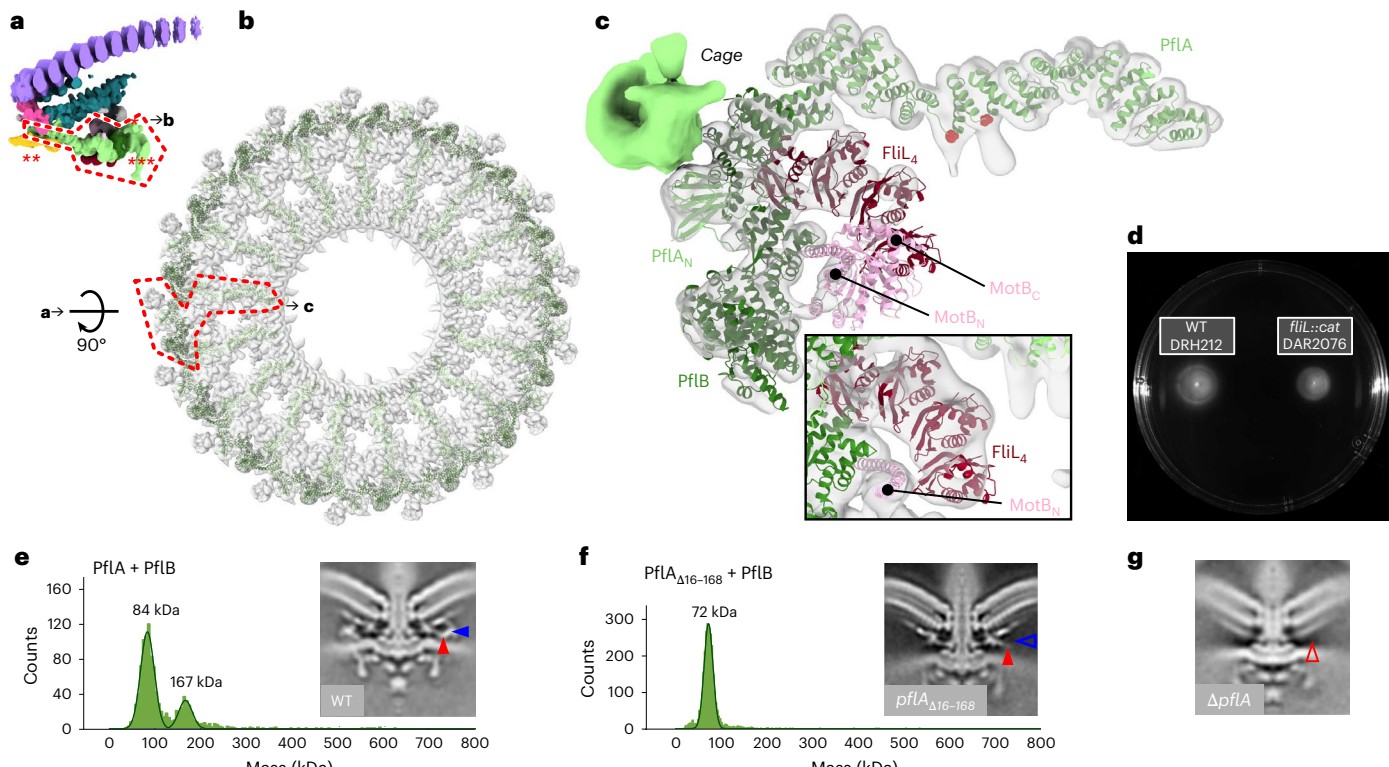

**Fig. 4 | PflA and PflB form a spoke-and-rim structure that scaffolds 17 stator complexes. a**, Side view of an asymmetric unit of the periplasmic scaffold highlighting the proximal disk (dashed red line). Asterisks denote unassigned PflD-adjacent density (*), E-ring (**) and peripheral cage (***). **b**, Top view of the proximal disk. Dashed red box denotes the asymmetric unit in **c**. **c**, Every asymmetric unit features 1 PflA, 1 PflB, 4 FliL and 1 stator complex (composed of 5 copies of MotA and 2 copies of MotB). PflA (light green) forms spokes whose N-terminal domain interacts with a rim of PflB (dark green) at the periphery of the scaffold. An arc of FliL (red) and periplasmic domain of MotB (pink, residues 68–247) are also evident at lower confidence. Inset: focus on FliL at lower threshold to demonstrate match of 4 FliL models into 4 periodic densities. **d**, A representative motility agar plate stabbed with WT and *fliL::cat*

demonstrates that *fliL* knockout has only a minor effect on motility. **e**, Mass photometry measurements confirm that the PflAB dimer (167 kDa peak) forms in vitro. Inset: 100 × 100 nm cross-section through the subtomogram average density map of the WT motor exhibits PflA$_C$ (filled red arrowhead) and PflB densities (filled blue arrowhead). Structure from EMD-3150 (ref. 17). **f**, Mass photometry shows that deleting the PflA β-sandwich and linker abolishes dimerization with PflB. Inset: 100 × 100 nm cross-section through a density map of the subtomogram average of this mutant reveals a vestigial PflA$_C$ density (filled red arrowhead) and loss of PflB (open blue arrowhead) (deposited as EMD-17417), whereas **g**, a 100 × 100 nm cross-section through a density map of the subtomogram average of a *pflA* deletion further lacks the vestigial PflA$_C$ density (open red arrowhead) (structure from EMD-3160 (ref. 17)).

$3.2 \times 10^{-6}$, respectively). Sequence-guided structural alignment of PflC$_N$ to *C. jejuni* HtrA yielded alignment of a core of 118 pruned residue pairs (of 216) with root-mean-square deviation (RMSD) of 2.448 Å (Extended Data Fig. 7a,b). Because PflC$_C$ is connected to PflC$_N$ by a flexible linker, we aligned it separately to the C-terminal PDZ domain of HtrA with RMSD between 42 pruned residue pairs (of 64) of 1.425 Å. The region of PflC$_N$ corresponding to the active site cleft of the serine protease domain is most divergent from HtrA (Extended Data Fig. 7c) and PflC lacks the catalytic triad residues His119, Asp150 and Ser224: the large loop that His119 occupies has been lost in PflC, Asp150 is replaced by Asn65, and although Ser224 aligns with PflC's Ser122, these residues do not align structurally (Extended Data Fig. 7d). This atrophy of the catalytic triad region, however, is the most substantial difference, and the global topologies of the two proteins are otherwise similar (Extended Data Fig. 8).

The second candidate medial disk component, PflD, is a 162-residue periplasmic protein that pulls down with PflA of the proximal disk (Supplementary Table 1). We inspected the peripheral part of the medial disk adjacent to PflC$_4$ which disappeared when we deleted *pflD*, and found that a model of PflD was consistent with this density, despite the lower resolution of this area of our map yielding only a modest mean main chain correlation coefficient of 0.63 (Fig. 3f and Extended Data Fig. 3b). This location is consistent with our co-purification of PflD with both PflA and PflB (Fig. 3e).

### PflAB spokes and rim and FliL arcs make the proximal disk

Finally, we sought to interpret our density map of the proximal disk, which contains PflA, PflB and stator complex protein MotB[17]. Consistent with this, the proximal disk features the extensive short antiparallel α-helical motifs characteristic of the repetitive TPR motifs[37,38] predicted for PflA and PflB (Fig. 4a).

To model PflA and PflB, we sought to fit their AlphaFold2 (ref. 29) predictions into our map. PflA was predicted to form an elongated superhelix consisting of 16 TPR motifs connected to an N-terminal β-sandwich domain by an unstructured linker, while PflB was mainly α-helical except two 5-residue β-strands. The predicted structure of a PflAB heterodimer (Extended Data Fig. 4d) indicated that PflA's N-terminal domain and linker wraps around PflB. We flexibly fitted these structures into our map, resulting in a structure in which 17 radial spokes of PflA position a continuous rim of 17 PflBs (Fig. 4b and Extended Data Fig. 2g). PflA fitted the map with a map-model FSC of 9.1 Å at FSC = 0.5 (Extended Data Fig. 3a). Similar to PflC, PflA is a glycoprotein, with N-linked glycans attached at N458 and N497 (ref. 39), and this modelling is validated by densities consistent with N-linked heptasaccharides emanating from our modelled locations of both residues (Fig. 4c, red atoms). The resolution of PflB was lower, presumably because it is more peripheral and flexible, with map-model mean main chain correlation coefficient of 0.71 (Extended Data Fig. 3b).

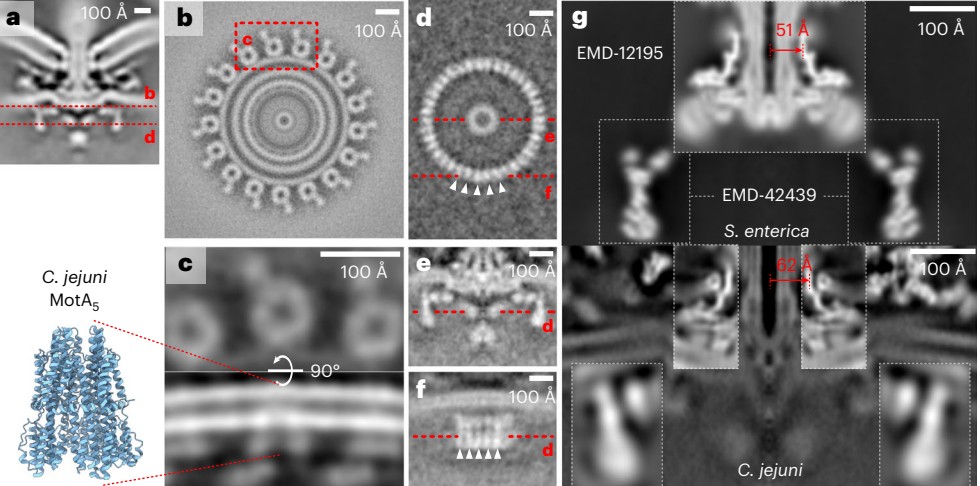

**Fig. 5 | Wider rings of additional stator complexes are incorporated in the *C. jejuni* periplasmic scaffold, while the rotor components are correspondingly wider. a**, A 100 × 100 nm cross-section through the subtomogram average of the wildtype *C. jejuni* motor (from EMD-3150 (ref. 17)) depicting locations of stator and rotor cross-sections illustrated in **b** and **d**. **b**, Cross-section through the whole-motor map just beneath the outer membrane shows 17 circular densities at the expected location of MotA. **c**, Top: focused refinement of the stator complexes reveals pentameric densities that, in cross-section (bottom), have the distinctive thimble-like shape of a MotA pentamer (from PDB 6YKM[11], with N-terminal helices of MotB removed). **d**, Cross-section through the *C. jejuni* C-ring showing 38-fold periodic structure (deposited as EMD-19642). Arrowheads highlight 5 of the 38 puncta. Labels E and F denote cross-sections depicted in respective panels. **e**, Cross-section through the centre of the *C.*

*jejuni* C-ring. **f**, Cross-section through the edge of the C-ring showing post-like densities corresponding to the periodicity shown in **d** (arrowheads) as have been reported in the *Salmonella* C-ring. **g**, Comparison of *Salmonella enterica* serovar Typhimurium and *C. jejuni* MS-ring and C-rings. Top: cross-section through a composite map of the *Salmonella* MS-ring (middle top dashed box, from EMD-12195 (ref. 12)) and C-ring (two lower dashed boxes, from EMD-42439 (ref. 8)) rotor components depicting the 51-Å-radius MS-ring β-collar. Both maps were low-pass filtered to 15 Å resolution and cylindrically averaged for like-for-like comparison with Bottom: cross-section through a composite map of the whole-motor *C. jejuni* map, with superimposed corresponding cross-sections through focused, cylindrically averaged *C. jejuni* MS-ring and C-ring maps (deposited as Additional data for EMD-16723), highlighting the wider 62-Å-radius β-collar.

To test the prediction that PflAB heterodimer formation is mediated by the linker between the N- and C-terminal domains of PflA binding a TPR-superhelical groove in PflB, with the N-terminal β-sandwich domain wrapping around PflB and the C-terminal α-helical PflA spoke pointing toward the motor axis (Fig. 4c), we measured their interaction using mass photometry. PflAB heterodimerized even at nanomolar concentrations (Fig. 4e), while deleting the linker and β-sandwich domain of PflA (residues 16–168) abolished dimer formation (Fig. 4f and Extended Data Fig. 9). A subtomogram average structure of the motor in a PflA$_{\Delta16-168}$ mutant confirmed that PflB was unable to assemble into the motor (Fig. 4e), although the C-terminal end of PflA remained evident, unlike a full *pflA* deletion[17] (Fig. 4g). Together with PflAB interaction seen in pulldowns[40], we conclude that PflAB dimerization is essential for completion of proximal disk assembly.

Although our previous work and the topology of the globular periplasmic MotB$_C$ dimer make its location unambiguous[17], the resolution of this region was low; this is presumably due to flexibility between this domain and the periplasmic portion of the helical linker between globular MotB$_C$ and its N-terminal transmembrane helix, meaning we could only crudely position a MotB model. This is consistent with the MotB$_C$ domain not being resolved in recent structures of purified MotA$_5$B$_2$ complexes[10,11]. The linker itself, however, was evident (Fig. 4c inset).

An arc of density partially encircling the periplasmic portion of the helical MotB$_N$ linker had similar radius and location as the tertiary structures of complete circles of FliL in other motors[41,42]; we found that a curved tetrameric homology model of FliL fitted into this arc with a mean main chain correlation coefficient of 0.61 (Fig. 4d and Extended Data Fig. 3b). Co-immunoprecipitation assays confirmed that FliL is found in pulldowns of PflA and PflB (Supplementary Table 1), suggesting that the FliL arc is augmented by PflB and PflA to scaffold MotB, and explains why PflA and PflB are both required for the high occupancy or static anchoring of stator complexes into the *C. jejuni* motor[17]. Indeed,

we found that deletion of *fliL* in *C. jejuni* had only a modest impact on motility (Fig. 4d) in contrast to *H. pylori* where FliL is essential, reinforcing that FliL's role is partially fulfilled by PflA and PflB in *C. jejuni*. The presence of the stator complexes in the *C. jejuni* structure, in contrast to their absence in *Salmonella*, indicates either high occupancy or static anchoring, probably mediated by their interactions with PflA, PflB, FliL and unidentified cytoplasmic proteins.

We could not assign identities to three remaining densities in the scaffold: the so-called E-ring that spaces the MS-ring from PflA, a cage previously observed in *H. pylori*[41] on the periphery of the PflB rim that extends through the membrane to wrap around the stator complexes, and a small density adjacent to PflD (Fig. 4a, opaque regions, single, double and triple asterisks, respectively).

## Conserved structures have adapted to a high-torque role

To better understand the conserved torque-generation machinery, we focused on the stator complexes, C-ring and MS-ring. By symmetry expansion and classifying stator complexes from our whole-motor map, we observed a pentameric structure in contact with the C-ring that is directly beneath the periplasmic peptidoglycan-binding domain of MotB (Fig. 5a–c). In cross-section, the dimensions and shape of this density match MotA$_5$ from the purified *C. jejuni* stator complex[11] (Fig. 5c). The consistent rotational register of this pentameric density even after symmetry expansion and classification, with a pentameric corner pointing toward the C-ring, indicates that stator complexes are most frequently in this rotational register.

We wondered how the *C. jejuni* rotor components, that is, the C-ring, MS-ring and rod, have adapted to interact with a wider ring of stator complexes. The *C. jejuni* C-ring is wider than that of *Salmonella*[17]; to ascertain its stoichiometry, we determined its structure by subtomogram averaging. To remove the strong 17-fold signal of MotA, we used a motile *C. jejuni* mutant whose stator complexes have lower occupancy than WT motors. Our subtomogram average revealed a 38-fold periodic

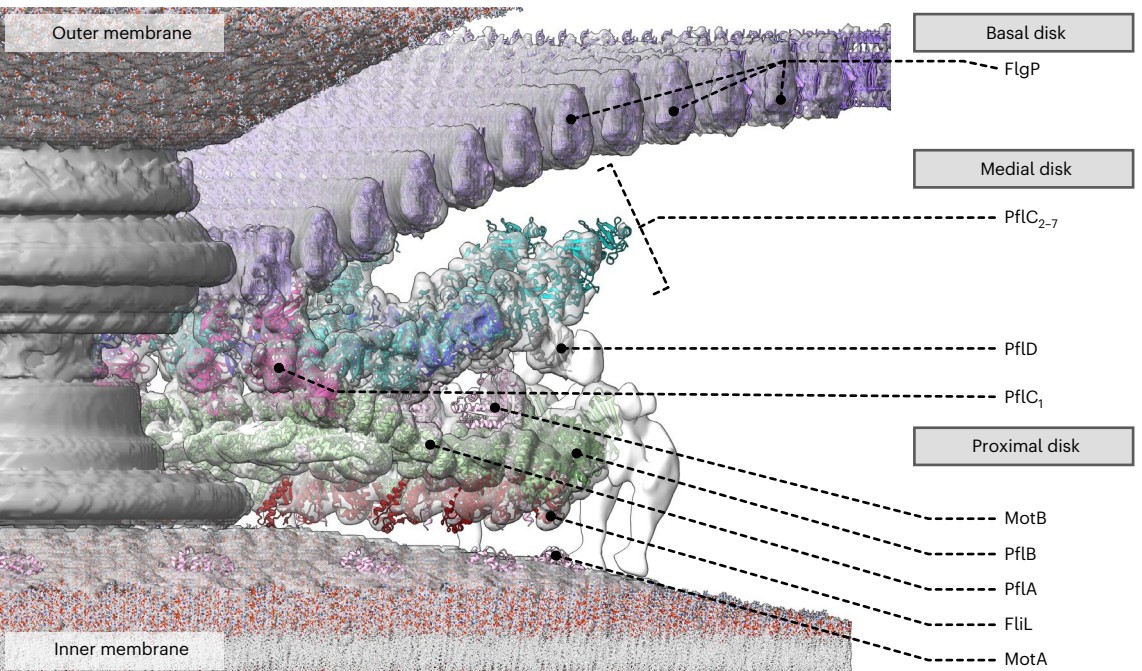

**Fig. 6 | A 'parts list' of protein adaptations to increase torque by scaffolding a wider ring of additional stator complexes, thus exerting greater leverage on the axial flagellum.** A partial cut-away schematic of the structure of the *C. jejuni* flagellar motor contextualizing the protein components modelled in this study. The basal disk is formed of FlgP; the medial disk of PflC and PflD; and the proximal disk of PflA, PflB and FliL together with stator complex components MotA and MotB.

structure whose vertical post-like architecture resembles that of the near-atomic-resolution *Salmonella* C-ring[7–9] (Fig. 5d–f).

C-ring diameter is reliant on templating on the C terminus of the MS-ring protein FliF[8,43]. To examine possible mechanisms behind templating a wider C-ring, we compared the *C. jejuni* MS-ring to that from *Salmonella*. The distinctive FliF β-collar has a radius of ~51 Å in *Salmonella* (Fig. 5g, top). Focused refinement of the *C. jejuni* MS-ring with imposed C38 symmetry expected from 1:1 stoichiometry of FliF and FliG[43,44] demonstrated a 62-Å-radius FliF β-collar (Fig. 5g, bottom). We could not resolve azimuthal features corresponding to discrete FliF subunits, but imposition of other arbitrary high-order symmetries did not alter the radius of FliF's β-collar. We conclude that the wider C-ring is achieved by *C. jejuni* assembling a wider MS-ring, which in turn templates the wider C-ring.

## Discussion

This study provides a near-complete structural inventory of the proteins incorporated into the *C. jejuni* motor during evolution of higher-torque output and adaptations of pre-existing components, providing insights into evolutionary origins and the universal principles of flagellar rotation (Fig. 6 and Supplementary Table 2). This work lays the foundation for understanding where these new proteins came from and how they became incorporated into a pre-existing machine.

Our structures explain how the *C. jejuni* motor produces approximately three times the torque of the *E. coli* and *Salmonella* motors. We previously inferred from *E. coli* and *Salmonella* that a single stator complex exerts a force of 7.3 pN[17]. Given the approximation that torque from multiple stator complexes is additive[45], the number of stator complexes, together with our force estimate and the radius at which stator complexes exert force on the C-ring make predictions that nicely explain observations of 3,600 pN nm torque for Campylobacterota flagellar motors (our structure-based prediction is 3,288 pN nm) versus observations ranging from 1,260 to 2,000 pN nm[45,46] for *E. coli* and *Salmonella* flagellar motors (our structure-based prediction is 1,606 pN nm). For the *C. jejuni* motor to build this wider ring of additional stator complexes to produce its higher torque, the spoke-and-rim scaffold of PflA and PflB,

anchored by PflC and PlfD, positions stator complexes in their wider ring. This scaffold facilitates the static positioning (or high occupancy) of the *C. jejuni* stator complexes[17]; the radius of the MS-ring is correspondingly larger than that of *Salmonella* to template a wider C-ring to maintain contact with the larger stator ring.

The structures of PflA and PflB hint at their possible origins. Both are composed of arrays of TPR motifs, which are widespread building blocks of structural scaffolds. The widespread nature of TPR motifs would have made them available for co-option to form PflA or PflB. Intriguingly, other flagellar proteins are homologous to parts of the widespread Tol/Pal cell-envelope maintenance system[6], and the Tol/Pal component YbgF features TPR motifs[47], making it tempting to contemplate a scenario in which PflA and PflB originated from another Tol/Pal co-option. Our structures support our previous speculation that PflA may be widespread among other species such as *Bdellovibrio*[18].

In *C. jejuni*, the PflAB-containing proximal disk is linked to the basal disk by the PflC-composed medial disk, while other species lack a medial disk and have separate proximal and basal disks. Our structure provides a foundation for understanding the function and mechanism of the relatively recent evolution of the medial disk[17]. We recently showed that the basal disk buttresses the motor when unwrapping flagellar filaments from the cell body[23]. Here we identify PflC as the mediator of the extensive connections between the proximal and basal disks (Figs. 3 and 4). PflC's shared domain organization and structure with HtrA-family enzymes suggests that the medial disk originated as a co-option of an ancestral member of the HtrA family[48]. Indeed, HtrA family proteins are periplasmic, meaning that they already co-localized with the flagellar motor. Curiously, HtrA also forms higher-order oligomers[36], suggesting that this oligomerization tendency may have been a pre-existing property easily exapted for its role as a structural scaffold.

The basal disk is comparatively ancient. While the basal disk protein FlgP comes from a broad protein family, ring oligomerization in the family correlates with the subset of sequences that feature a sequence insertion between the SHS2 α-helix and the second β-strand, as seen with outer membrane-associated rings formed by FlgT[49]. Curiously,

the basal disk from *Wolinella* has been proposed to form a spiral, not concentric rings[50], raising the possibility that FlgP forms concentric rings only in the presence of a medial disk.

Our study also provides clues to the universal mechanism of flagellar rotation. In our averaged structure, MotA$_5$ pentamers all have similar rotational orientations relative to the C-ring, meaning that the majority of MotA pentamers are in this orientation, as averaging an unbiased range of orientations would yield a circle. This, on face value, is inconsistent with the dominant 'cogwheel' model of the flagellar rotation mechanism, which posits that rotation of a MotA$_5$ pentamer around its MotB anchor enables the pentamers to act as cogs that drive rotation of the larger C-ring cogwheel. This meshing means that MotA$_5$ pentamers would always be engaged with the C-ring, that is, they would have a 'high duty ratio'. Because the number of small cogs (17 in *C. jejuni*, 11 in *E. coli* and *Salmonella*) has a symmetry mismatch with the number of teeth in the big cog (38 in *C. jejuni*, 34 in *E. coli* and *Salmonella*), once assembled, each of the small cogs, which are evenly distributed around the circumference of the larger cog, would need to be rotationally orientated differently to enable all to mesh simultaneously with the symmetry-mismatched teeth of the larger cog. Thus in an averaged structure, MotA$_5$ would appear as smooth circles, inconsistent with our observation.

We can imagine two explanations for this inconsistency. First, the symmetry mismatch might be capacitated by structural tolerances, such as flexibility in the so-called FliG$_C$ 'torque helix' (a 'tooth' of the large cog) that interacts with MotA, which would enable all pentamers to retain the same orientation while the FliG$_C$ cogteeth distort to facilitate maintaining the interface. Previous models of high-duty-ratio stator complexes cite tolerance from an elastic linker from peptidoglycan through MotB to MotA[51], although this could not be universal, as species such as *C. jejuni* have stator complexes held rigidly in proteinaceous cages. With such tolerances, all cogs would rotate in unison to drive a single rotational step of the C-ring, all starting and finishing a work cycle in the orientation that we observe. Alternatively, the tolerances might be insufficiently broad to enable engagement of all stator complexes, in which case the duty ratio might be lower than a naïve interpretation of the cogwheel model suggests. A second explanation is that the cogwheel model does not adequately describe the mechanism underlying flagellar rotation. Indeed, a recent study hints that previous results suggesting a high stator complex duty ratio derived from artefacts from stator complex dissociation at low load, and that stator complexes actually have a low duty ratio[52].

While our single-particle analysis approach achieves substantially higher resolution than previous studies, our approach is not without limitations. Our study focused on the 17-fold rotationally symmetric periplasmic scaffold, but flagella feature diverse symmetry-mismatched subcomponents. The approach can in principle identify arbitrary symmetries, but we did not resolve other substructures such as the C-ring, which instead required a subtomogram averaging approach, probably due to excess noise owing to the size, heterogeneity and the in situ context. Different data acquisition schemes are needed to understand how to extract signal. Nevertheless, using minicells to study polar machinery has promise for increases in resolution in other systems, including Tad pili[53] in *Caulobacter crescentus* minicells[54], type II secretion systems[55] in *Pseudomonas* minicells[56], chemoreceptors in *C. jejuni*[57] and flagellar motors in other polar flagellates[58,59].

Our approach builds a foundation to study molecular evolution mechanisms and provides information on how torque is generated. Our approach confirms that imaging protein machines in situ can provide subnanometre resolution without sectioning, and lays a foundation for future studies expanding to other symmetry elements and structures.

## Methods

### Materials availability

Plasmids and strains generated in this study (see Supplementary Table 3) are available from the lead author.

### Bacterial strains and culture conditions

*C. jejuni* 81–176 or NCTC11168 were cultured from frozen stocks on Mueller–Hinton (MH) agar (1.5% w/v) supplemented with trimethoprim (10 μg ml$^{-1}$) (MHT) for 1–2 days at 37 °C under microaerophilic conditions (5% $O_2$, 10% $CO_2$, 85% $N_2$) in a Heracell 150i trigas incubator (Thermo Fisher). Additional antibiotics were added to the agar medium when required: kanamycin at 50 μg ml$^{-1}$, streptomycin at 2 mg ml$^{-1}$. All 81–176 mutants were constructed in DRH212 (ref. 60), a streptomycin-resistant derivative of *C. jejuni* 81–176, which is the reference wildtype strain in this work unless otherwise stated. When working with *E. coli*, cultures were grown at 37 °C on Luria-Bertani (LB) agar plates (1.5% w/v) or in LB medium with agitation, both supplemented with carbenicillin at 100 μg ml$^{-1}$. Please see Supplementary Table 3 for details of strains and plasmids used in this study.

### Strain construction

The minicell ($\Delta flhG \Delta flaAB$) and PflA truncation ($pflA_{\Delta18-168}$) strains were constructed as described previously[60,61]. Briefly, $aphA$-$rpsL^{WT}$ cassettes flanked by ~500 bp overhangs with homology to the targeted chromosomal loci and EcoRI sites at the 5' and 3' termini were synthesized by 'splicing by overlap extension' PCR. Linear DNA fragments were methylated at their EcoRI sites with EcoRI methyltransferase (New England Biolabs) and transformed into *C. jejuni* using the biphasic method[62]. Transformants were selected for on MH agar supplemented with 50 μg ml$^{-1}$ kanamycin. Replacement of the $aphA$-$rpsL^{WT}$ with the desired mutation was achieved using the same method, but with transformants being selected for on MH agar supplemented with 2 mg ml$^{-1}$ streptomycin sulphate. Kanamycin-sensitive, streptomycin-resistant transformants were single-colony purified and checked by Sanger sequencing (Source Bioscience). For the minicell background, in-frame deletion of $flhG$ left the first and last 20 codons intact, while the $\Delta flaAB$ allele spanned from 20 base pairs upstream of the $flaA$ translational start site to codon 548 of $flaB$.

To construct the *C. jejuni fliL* mutant, we made a *cat* insertional knockout and confirmed absence of polar effects. To preserve expression of the essential *acpS* gene downstream of *fliL*, we constructed a *fliL* mutant that disrupted *fliL* with an antibiotic-resistance cassette containing an intact *flaA* promoter positioned to maintain expression of *acpS*. First, the *fliL* locus from *C. jejuni* 81–176 was PCR amplified with an HpaI site engineered within the *fliL* coding sequence. This fragment was then cloned into the BamHI site of pUC19 to create pDAR1712. The *flaA* promoter and start codon were PCR amplified from *C. jejuni* 81–176 and cloned into the XbaI and BamHI sites of pUC19 to create pDAR2039. The chloramphenicol-resistance cassette containing *cat* was digested as a PstI fragment from pRY109 and cloned into PstI-digested pDAR2039 to create pDAR2045. The *cat-flaA* promoter was then digested from pDAR2045 as an EcoRI-BamHI fragment, treated with T4 DNA polymerase to create blunt ends and cloned into the HpaI site of pDAR1712 to create pDAR2072. pDAR2072 was verified to contain the *cat-flaA* promoter in the correct orientation to maintain expression of *acpS*. DRH212 was then electroporated with pDAR2072 and chloramphenicol-resistance transformants were recovered. Colony PCR verified creation of a *fliL* mutant (DAR2076).

Deletion mutants of *C. jejuni* NCTC11168 were constructed by double-crossover homologous recombination with an antibiotic-resistance cassette to remove most of the coding sequence using overlap PCR products. As an example, deletion of *cj1643* (*pflC*) is described. First, ~500 bp upstream of the *cj1643* start codon was amplified using CSO-3359/3360 and ~500 bp downstream of the *cj1643* stop codon was amplified with CSO-3361/-3362 from genomic DNA (gDNA) of the wildtype strain (CSS-0032). A non-polar kanamycin-resistance cassette ($aphA$-3, Kan$^R$) was amplified from pGG1 (ref. 63) with primers HPK1/HPK2. To fuse the up- and downstream regions of *cj1643* with the resistance cassette, the three fragments were mixed and subjected to overlap-extension PCR with CSO-3359/3362. PCR products were

electroporated into the WT strain as previously described[64]. The final deletion strain (CSS-4087; NCTC11168 Δ*cj1643*) was verified by colony PCR with CSO-3363/HPK2. Deletion of *cj0892c* (*pflD*) in *C. jejuni* strain NCTC11168 was generated in a similar fashion: *cj0892c*::*aphA-3* (CSS-4081; NCTC11168 Δ*cj0892c*).

To fuse *sfgfp* to the penultimate codon of *cj0892c* (*pflD*), its coding sequence was first amplified with CSO-3611/3612, digested with *Nse*I/*Cla*I and inserted into similarly digested pSE59.1 (ref. [65]), which had been amplified with CSO-0347/CSO-0760, to generate pSSv106.5, where *cj0892c* transcription was driven from the *metK* promoter. The plasmid was verified by colony PCR with CSO-0644/3270 and sequencing with CSO-0759. Next, *sfgfp* was amplified from its second codon from pXG10-SF[66] with CSO-3279/3717, digested with *Cla*I and ligated to pSSv106.5 (amplified with CSO-3766/0347 and also digested with *Cla*I). This generated pSSv114.1, which was verified by colony PCR with CSO-0644/0593 and sequencing with CSO-0759/3270. The fusion of *rdxA*::P*~metK~*-*cj0892c*-sfGFP was amplified from pSSv114.1 with CSO-2276/2277 and introduced into the *rdxA* locus of Δ*cj0892c* (CSS-4081) by electroporation. Clones were verified via colony PCR and sequencing with CSO-0349 and CSO-0644. Colony PCR was also used to confirm retention of the original deletion with CSO-3343 and HPK2.

Similar to construction of deletion mutants, C-terminal epitope-tagged strains were generated by homologous recombination at the native locus by electroporation of a DNA fragment. The 3×FLAG sequence was fused to the penultimate codon of the coding sequence to allow in-frame translation of the tag. The DNA fragment contained ~500 bp upstream of the penultimate codon of the gene of interest, the sequence of the epitope tag, a non-polar resistance cassette and the ~500 bp downstream sequence of the gene. As an example, 3×FLAG tagging of PflA (CSS-5714) is described. The upstream fragment was amplified with CSO-4224 and CSO-4225 from *C. jejuni* NCTC11168 WT gDNA. The downstream fragment was amplified using CSO-4226 and CSO-4227. The fusion of the 3×FLAG tag with the gentamicin-resistance cassette was amplified from *fliW*::3xFLAG-*aac*(3)-IV[63] using CSO-0065 and HPK2. Next, a three-fragment overlap PCR using CSO-4224 and CSO-4227 was performed and the resulting PCR product was electroporated into CSS-4666. The obtained clones were validated by PCR using CSO-4223 and HPK2 and by sequencing using CSO-4223. PflB-3×FLAG (CSS-5716) and PflC-3×FLAG (CSS-4720) were generated similarly. The 3×FLAG with a non-polar kanamycin-resistance cassette was amplified from *csrA*::3×FLAG-*aphA*-3 (ref. [63]).

To construct a FlgQ-mcherry fusion protein for expression, a 76-bp DNA fragment containing the cat promoter and start codon with an in-frame BamHI site from pRY109 was amplified by PCR and cloned into the XbaI and XmaI sites of pRY112 to create pDAR1003. PCR was then used to amplify mcherry from codon to the stop codon, which was then inserted into XmaI and EcoRV sites of pDAR1003 to create pDAR1006. This plasmid contains a start codon that is in-frame with DNA for BamHI and XmaI sites followed by the mcherry coding sequence. Primers were then designed and used for PCR to amplify flgQ from codon 2 to the stop codon from *C. jejuni* 81–176. This fragment was inserted in-frame into the BamHI site of pDARH1006 to create pDRH7476, which was then conjugated into DRH2071 to result in DRH7516.

### CryoEM sample preparation

*C. jejuni* Δ*flhG* Δ*flaAB* cells were grown on MH plates and resuspended in phosphate-buffered saline (PBS) buffer (137 mM NaCl, 2.7 mM KCl, 10 mM Na$_2$HPO$_4$, 1.8 mM KH$_2$PO$_4$, pH 7.4). Cells were spun at 1,500 × *g* for 20 min to pellet whole cells. The minicell-enriched supernatant was removed and spun in a tabletop microcentrifuge at 21,000 × *g* for 5 min to pellet the minicells. The pellet was then resuspended to a theoretical optical density (OD)$_{600}$ of ~15.

Minicells were vitrified on QUANTIFOIL R0.6/1 or R1.2/1.3 holey carbon grids (Quantifoil Micro Tools) using a Vitrobot Mark IV (Thermo Fisher).

For electron cryo-tomography and subtomogram averaging, whole cells were grown on MHT agar, restreaked on fresh plates and grown overnight before use. Freshly grown cells were suspended into ~1.5 ml of PBS buffer and concentrated to an approximate theoretical OD$_{600}$ of 10 by pelleting at 1,500 × *g* for 5 min on a tabletop microcentrifuge and resuspending appropriately. Of the concentrated cell sample, 30 µl was mixed with 10 nm gold fiducial beads coated with bovine serum albumin (BSA). Of this mixture, 3 µl was applied to freshly glow-discharged QUANTIFOIL R2/2, 300 mesh grids. Grids were plunge frozen in liquified ethane–propane using a Vitrobot mark IV.

### Single-particle analysis image acquisition

Micrographs of the minicell sample were collected using 300 keV Thermo Fisher Titan Krios TEMs across two sessions using EPU acquisition software. The first dataset was collected on a microscope with a Falcon III direct electron detector (Thermo Fisher), the second dataset using a K2 direct electron detector equipped with a GIF energy filter (Gatan), using a slit width of 20 eV. Due to our large particle size relative to that of the holes, we collected one shot per hole. Gain correction was performed on the fly. Details of data-collection parameters are described in Extended Data Table 1.

### Tilt-series acquisition

Tilt series of motors in *pflA*$_{Δ18–168}$ were collected using a 300 keV Titan Krios TEM (Thermo Fisher) equipped with a K2 direct electron detector and a GIF energy filter (Gatan) using a slit width of 20 eV. Data were collected in Tomography 5 (Thermo Fisher) using a dose-symmetric tilt scheme across ±57° in 3° increments. We used a dose of 3 e$^-$ Å$^{-2}$ per tilt, distributed across 4 movie frames. Pixel size was 2.2 Å and defocus range from −4.0 to −5.0 µm. To determine the *C. jejuni* C-ring architecture, tilt series of 194 motors in DRH8754 were collected using a 200 keV Glacios TEM (Thermo Fisher) equipped with a Falcon 4 direct electron detector and a Selectris energy filter (Thermo Fisher) using a slit width of 10 eV. Data were collected in Leginon automated data-collection software using a dose-symmetric tilt scheme across ±51° in 3° increments. We used a dose of 2.7 e$^-$ Å$^{-2}$ per tilt, distributed across 5 movie frames. Pixel size was 1.9 Å and defocus was −4.0 µm. All other tilt-series datasets were acquired on a 200 keV FEI Tecnai TF20 FEG transmission electron microscope (FEI) equipped with a Falcon II direct electron detector camera (FEI) using Gatan 914 or 626 cryoholders. Tilt series were recorded from −57° to +57° with an increment of 3° collected defocus of approximately −4 µm using Leginon automated data-collection software at a nominal magnification of ×25,000 and were binned two times. Cumulative doses of ~120 e$^-$ Å$^{-2}$ were used. Overnight data collection was facilitated by the addition of a 3 litre cold-trap Dewar flask and automated refilling of the Dewar cryoholder triggered by a custom-written Leginon node interfaced with a computer-controlled liquid nitrogen pump (Norhof LN2 Systems).

### Single-particle analysis

Movie frames were aligned and dose weighted according to exposure, as implemented in MotionCor2 (v.2.1)[67]. All subsequent processing was done in RELION (3.1)[68,69]. Contrast transfer function (CTF) correction was performed using CTFFIND4 (ref. [70]), using the RELION wrapper. Flagellar motor positions were picked manually, yielding 79,287 particle coordinates for the K2 dataset and 14,605 particle coordinates for the F3 dataset.

The two datasets were first processed separately in RELION 3.1 before merging for a final round of refinement. For the K2 dataset, 79,287 particles were extracted at a box size of 800 px. A round of 2D classification removed junk and membrane particles, and an initial model was created using these particles with imposed C17 symmetry, which is known from past structural characterization of the motor by subtomogram averaging[17]. A round of mask-free 3D classification and refinement with applied C17 symmetry produced the first consensus refinement. A total of 27,164 particles were then re-extracted,

centring on the periplasmic structures. After another round of 3D classification, 19,736 particles were refined in C17 symmetry to produce a whole-motor reconstruction at 9.88 Å using gold-standard refinement. For the F3 data, 14,605 particles were extracted at a box size of 1,000 px rescaled to 500 px. They underwent 2D classification to remove junk, 3D classification and refinement to arrive at an initial consensus 3D structure. The particles were again recentred on the periplasmic structures and underwent another round of refinement. Finally, the 13,054 particles were re-extracted at an unbinned 1,000 px box size for a final round of refinement. The two recentred refined datasets were merged, assigning them different RELION 3.1 optics groups, and refined to a global resolution of 9.36 Å (32,790 total particles).

Signal subtraction was used to further refine the structure of the periplasmic scaffold. A mask encompassing the regions of interest was made by segmenting and smoothing the whole-motor map using UCSF Chimera (1.16)[71,72] and its Segger plugin[73], binarizing and adding a soft-edge in RELION 3.1. The mask included the periplasmic scaffold and first ring of the basal disk, as the scaffold appears to attach onto it. This mask was used to computationally remove signal outside of the periplasmic regions of interest, as implemented in RELION 3.1. The signal subtraction and subsequent masked classification and refinement were conducted for the combined dataset, as well as K2 and F3 datasets separately. The highest resolution was reached with the merged data, with the periplasmic scaffold map reaching 7.68 Å from 32,790 particles.

The periplasmic scaffold map was post processed using LAFTER[74] as implemented in the CCP-EM 1.6.0 software suite[75] to suppress noise and enhance signal between the half maps. The LAFTER-filtered map of the scaffold was used for docking and modelling of periplasmic regions.

### Focused refinement of LP-, MS- and C-rings

After convergence of the full-motor refinement in C17 symmetry, focused refinement of the MS-, LP- and C-rings was carried out using a lathed map lacking azimuthal features. The C17 map was lathed by applying C360 symmetry using EMAN2, producing a map with axial and radial features but no azimuthal features. This map was used as a reference for asymmetric refinement, so that no bias towards a particular cyclic symmetry could be imposed.

For the LP-ring, a tubular mask covering the LP-ring was used for signal subtraction and focused refinement. Subtracted particles were recentred on the LP-ring in a smaller 256-pixel box. A recentred subvolume of the full lathed map was used for the initial reference. Refinement with imposition of C26 symmetry produced the LP-ring localized reconstruction.

For the MS-ring, a tubular mask covering the MS-ring was used for signal subtraction and focused refinement. Subtracted particles were recentred on the MS-ring in a smaller 256-pixel box. A recentred subvolume of the full lathed map was used for the initial reference. Refinement with imposition of C38 symmetry produced the MS-ring localized reconstruction.

For the C-ring, a toroidal mask covering the C-ring was used for signal subtraction and focused refinement against the full-size lathed motor map. Refinement with imposition of C38 symmetry produced the C-ring localized reconstruction. This map was then post processed and sharpened with an empirically determined B factor of −1,200 Å$^{-2}$.

### Focused classification of stator complexes

From the full-motor refinement in C17 symmetry, particles were symmetry expanded and rewindowed into a smaller 360-pixel box centred on a stator complex, with subtraction of the surrounding signal using a spherical mask encompassing a triplet of neighbouring stator complexes. Classification without alignment produced the stator maps shown.

### Subtomogram averaging

Fiducial models were generated and tilt series were aligned for tomogram reconstruction using the IMOD package[76]. Tomo3D[77] was used

to reconstruct tomograms with the SIRT (Simultaneous Iterative Reconstruction Technique) method due to approximately 1 particle per tomogram. All steps were automated by in-house custom scripts.

Subtomogram averaging was performed using the Dynamo package[78] unless otherwise stated. Motors were picked using 3dmod from the IMOD suite and imported into Dynamo as 'oriented particles' using an in-house script, and subtomograms were extracted for averaging. For each structure, an initial model was obtained by reference-free averaging of the oriented particles, with randomized Z-axis rotation to alleviate missing wedge artefacts. This initial model was used for a first round of alignment and averaging steps, implementing an angular search and translational shifts, with cone diameter and shift limits becoming more stringent across iterations. The resulting average was used as a starting model for a round of masked alignment and averaging. In this round, custom alignment masks were implemented, focusing on the periplasmic and inner-membrane-associated parts of the motor. This excluded dominant features that would otherwise drive the alignment, most prominently the outer membrane and extracellular hook. A 17-fold rotational averaging was applied. The final $pflA_{\Delta18-168}$ average was derived from 103 particles, the $\Delta pflC$ average from 101 particles, $\Delta pflD$ average from 195 particles, and the $flgQ$-mCherry average from 155 particles.

The C. jejuni C-ring subtomogram average using DRH8754 used the PEET package[79]. Tomograms were CTF corrected using IMOD, motors picked using 3dmod and subtomograms extracted for averaging in PEET. After an initial C1 whole-motor alignment, a subsequent alignment recentred on the tightly masked C-ring was performed. To capitalize on the redundancy of C-ring archecture and to increase the number of particles, each motor was then symmetry expanded so that the C-ring beneath each of the 17 asymmetric units could be treated as a separate particle. Subsequent alignment and classification to remove poorly aligned particles yielded the final structure.

### De novo modelling

Signal sequences predicted with SignalP (6.0)[80] were removed from all sequences before modelling. ColabFold[81], the community-run implementation of AlphaFold2 (ref. [29]), was used to create structural models of PflA, PflB, a dimer of PflA-PflB, PflC, PflD, FlgP (as a monomer, trimer and heptamer), FlgQ, FliL and a dimer of MotB$_C$ (with transmembrane residues 1–67 removed).

The PflAB dimer model was created by merging two predicted structures: that of a dimer of PflA residues 16–455 and PflB residues 113–820, and full-length PflA.

### Docking and refinement

For regions where α-helices and β-sheets were resolved, we refined our AlphaFold-predicted protein models into the scaffold map using ISOLDE[82] in UCSF Chimera X 1.6. We imposed torsion, secondary structure and reference distance restraints in our modelling due to resolution limitations. The PflB chain of the PflAB dimer was rigidly docked into the map, the PflA chain was refined into the appropriate density and any clashes resolved. Models of FlgP multimers showed that each subunit interacts with subunits i − 1 and i − 2. To avoid artefacts due to this, a heptamer of FlgP was docked into the innermost basal disk ring and then refined into the map. Two subunits were then removed from each end, resulting in a fitted FlgP trimer. The PflC model was first separated into three domains, PflC$_N$ (residues 16–252), linker (253–263) and PflC$_C$ (264–364). Each domain was rigidly docked into its appropriate density, the three chains merged and the resulting protein was fitted using ISOLDE to resolve clashes or poor geometry at merging points. Six copies of PflC$_N$ were rigidly docked into an asymmetric unit of PflC lattice in the medial disk, forming PflC$_{2-7}$, PflD, MotB and an arc of FliL were rigidly docked into the LAFTER-filtered scaffold map.

We used 'phenix.validation_cryoem' of the Phenix package[83] to evaluate map and model quality for PflA, PflB, PflC, PflC$_{2-7}$, PflD, FlgP

trimer, MotB dimer and FliL. For each model, a soft mask of the region was first created and applied to the map. The resulting masked volume and corresponding structural model were input to 'phenix.validation_cryoem' which generated map-model FSC plots or cross-correlation scores.

### Flagellar motility assays

WT *C. jejuni* and DAR2076 were grown from freezer stocks on MH agar containing trimethoprim for 48 h in microaerobic conditions at 37 °C. Strains were restreaked on MH agar containing trimethoprim and grown for 16 h at 37 °C in microaerobic conditions. After growth, strains were resuspended from plates and diluted to an $OD_{600}$ of 0.8. Strains were then stabbed in MH motility agar (0.4% agar), incubated at 37 °C in microaerobic conditions for 30 h and assessed for migration from the point of inoculation.

### Bead assay of rotational steps

The *C. jejuni* bead assay strain was constructed by deleting *flaA* (to shorten the flagellar filament), making a cysteine substitution in *flaB* (to allow for filament biotinylation via a malemide-PEG2-biotin linker (Thermo Scientific)) making an alanine substitution in *flhF* (which moves motors to a subpolar position permitting better bead coupling), and deleting *cheY* (to prevent motor switching). From frozen stocks, this strain was cultured on MH agar (1.4% w/v) supplemented with trimethoprim (10 µg ml⁻¹) overnight at 37 °C in a pouch with a gas-controlling sachet (EZ CampyPouch System BD, 260685) and a damp paper towel. The growth from this culture was restreaked onto another plate and grown in the same conditions overnight. Passages to new plates on the following 2 days were optionally made to propagate the culture for multiple days of experiments.

After growth, bacteria were washed off the plate with MH broth and washed twice with PBS (centrifuged for 2 min at 8,900 × *g*) to create a 500 µl suspension of $OD_{600}$ = 2. Next, 25 µl of 20 mM malemide-PEG2-biotin linker in PBS was added and left to react for 10 min. The bacteria were then washed twice more in PBS. A custom tunnel slide composed of a slide with two drilled holes, a parafilm spacer and another glass slide was constructed and connected to a peristaltic pump (Lambda Multiflow) to draw flow through the slide. Poly-L-lysine solution (Sigma-Aldrich, P4707) was flowed into the slide channel and left for 5 min to coat the glass surface. Excess poly-L-lysine was flushed with PBS. Bacteria were added and left for 5 min to settle on the surface. Unattached bacteria were flushed sequentially with PBS and MH broth. Beads (1,300 nm diameter, streptavidin-coated) (Sigma-Aldrich, 49532) were washed once in PBS and dispersed in MH broth. Beads were added to the flow slide and left for 5 min to settle and conjugate to filaments. Excess beads were flushed away with MH broth. Spinning beads were observed with a bright-field microscope with a high numerical aperture (NA) objective (×100, 1.46 NA, Zeiss, 420792-9800-720) and recorded with a high-speed scientific complementary metal-oxide semiconductor camera (Optronis CL600X2-FULL-M-FM). Positions of slowly rotating beads attached to cells treated with CCCP at concentrations of 0–5 µM in MH broth revealed preferred phase-invariant dwell positions.

### Plasmid construction and cloning in *E. coli*

*C. jejuni* proteins for recombinant expression in *E. coli* were cloned into the pLIC plasmid backbone, which confers resistance to ampicillin and places the gene of interest under an isopropyl β-D-1-thiogalactopyranoside (IPTG)-inducible T7 promoter for high levels of controlled expression. We used WT *C. jejuni* genomic DNA as template for gene amplification (extracted using the Wizard genomic DNA purification kit by Promega), and the Gibson Assembly method[84] to seamlessly assemble all plasmid constructs. For all constructs, primer pairs were designed to amplify (1) the pLIC backbone and (2) the gene to be expressed, while also introducing a 25-30 bp complementary overlap between the two fragments. The pLIC plasmid primers also introduced an N-terminal hexahistidine tag. After vector linearization and purification of PCR product, it was digested with DpnI (New England Biolabs) to remove template vector.

The resulting linear DNA fragments were assembled using the Gibson Assembly master mix (New England Biolabs). Of the mix, 5 µl was added to 15–20 fmol of linearized vector and 4× excess of insert, and topped up to 10 µl with double-distilled water (ddH$_2$O). The tube was incubated at 50 °C for 15 min and kept on ice until transformation.

Before transformation, 30 µl of ddH$_2$O was added to the 10 µl reaction. Of the diluted Gibson mix, 2 µl was added to 25 µl of chemically competent *E. coli* DH5α and transformed using the heat shock method[85]. The entire volume of the tube was then plated onto a carbenicillin-supplemented LB agar plate.

After confirmation by Sanger sequencing (Source Bioscience), each assembled construct was isolated from the cloning strain (QIAprep Spin Miniprep kit, QIAGEN) and transformed into *E. coli* BL21(DE3) for recombinant overexpression.

### Protein overexpression and purification

All proteins encoded on pLIC expression vectors were purified using the same protocol. A small (5 ml) overnight liquid culture of *E. coli* BL21(DE3) carrying the appropriate expression vector was prepared and diluted 1:50 in 1,000 ml of LB medium. Shaking at 37 °C, the culture was grown to $OD_{600}$ of 0.4–0.6, after which protein expression was induced by addition of 0.5 mM IPTG. Temperature was reduced to 18 °C and protein was expressed overnight.

Cells were collected at 5,000 × *g* at 4 °C for 20 min. All subsequent steps were done on ice using buffers chilled to 4 °C. The cell pellet was gently resuspended in ~35 ml of wash buffer (50 mM Tris-HCl, 100 mM NaCl, 30 mM imidazole, pH 7.5). DNase and protease inhibitor were added (cOmplete Protease Inhibitor Cocktail, Roche). Cells were lysed using a LM10 Microfluidizer Processor cell disrupter (Analytik) at 15,000 psi. Lysate was centrifuged at 17,000 × *g* at 4 °C for 30 min to pellet debris. The resulting supernatant was filtered through a 0.45 µm syringe filter (Whatman).

A 5 ml HisTrap HP affinity chromatography nickel column (Cytiva) was first equilibrated with wash buffer. Supernatant was loaded onto the column with a peristaltic pump at a flow rate of 3 ml min⁻¹ The column was washed with 50 ml of wash buffer and then transferred onto a Fast protein liquid chromatography system (Bio-Rad). The column was further washed until the UV trace was flat. Then, protein was eluted from the column using a high-imidazole buffer (50 mM Tris-HCl, 100 mM NaCl, 500 mM imidazole, pH 7.5) at a flow rate of 2 ml min⁻¹ using 'reverse flow'.

The purified protein was kept at 4 °C or flash frozen in liquid nitrogen for longer-term storage before characterizing them by mass photometry.

### Analytical size exclusion

Analytical size exclusion of PflC and PflC$_N$ (Δ236–349) was performed with an ENrich SEC 650 column (Bio-Rad), equilibrated with 1× PBS at a flow rate of 0.1 ml min⁻¹ and a total sample injection volume of 400 µl. The column was calibrated using the Protein Standard Mix 15–600 kDa (Supelco, 69385). Absorption was recorded at 280, 220 and 495 nm to follow elution profiles and plotted using GraphPad Prism.

### PflC PSI-BLAST and hidden Markov model construction

*C. jejuni* PflC (CJJ81176_1634) was used as seed for a PSI-BLAST search[86] against the NCBI clustered non-redundant protein sequence database (nr_clustered, 5 December 2024) using default settings. After four iterations using default settings, the top 500 hits were used to build an HMM using hmmbuild from the hmmer 3.3 package[87]. This HMM was then used with hmmsearch to search predicted proteins from *C. jejuni* 81–176 (ASM1552v1).

## Mass photometry

Microscope coverslips (24 × 60 mm, Carl Roth) and CultureWell Gaskets (CW-50R-1.0, 50-3 mm diameter × 1 mm depth) were cleaned with alternating $ddH_2O$ and 100% isopropanol washes, then dried roughly with pressurized air and left to dry further overnight at room temperature. Before use, gaskets were assembled onto coverslips and placed on the lens of a One$^{MP}$ mass photometer (Refeyn) with immersion oil.

For each measurement, a gasket was first filled with 18 µl of PBS buffer and the instrument was focused. Then, 2 µl of sample was added to the droplet and rapidly mixed by pipetting. Measurements were then started using AcquireMP v.1.2.1 (Refeyn). For each measurement, data were acquired for 60 s at 100 frames per second. Mass photometry data were processed and analyzed in DiscoverMP software v.1.2.3 (Refeyn).

Measurements were conducted using affinity chromatography-purified proteins diluted to 200–800 nM, calculated from absorption at 280 nm. Of sample, 2 µl was added to 18 µl PBS droplet and mixed. For measurements of hetero-oligomers, the different proteins were first combined and mixed in a separate tube and subsequently applied to the PBS droplet. MP measurements were calibrated against molecular masses of commercial NativeMark unstained protein standards (Thermo Fisher). A volume of 1 µl of NativeMark was diluted 30-fold in PBS and 2 µl of this solution was added to 18 µl PBS for measurement. Detected peaks corresponded to 66 kDa, 146 kDa, 480 kDa and 1,048 kDa, and were used to calibrate subsequent measurements in DiscoverMP.

## Co-immunoprecipitation (coIP) of PflA/B-3×FLAG with PflD-sfGFP and PflC-3×FLAG

Chromosomally epitope-tagged fusions of PflC-3×FLAG (CSS-4720) or PflA-3×FLAG and PflD-sfGFP (CSS-5714), and PflB-3×FLAG and PflD-sfGFP (CSS-5716) were used together with the untagged *C. jejuni* NCTC11168 WT (CSS-0032) and PflD-sfGFP only (CSS-4666) as controls for immunoprecipitation. Co-purification of FlgP or PflD-sfGFP was investigated by western blot (WB) analysis using FlgP-specific antisera[22] or an anti-GFP antibody (Roche, 11814460001, RRID: AB_390913), respectively. In brief, strains were grown to an $OD_{600}$ of 0.6, and 60 $OD_{600}$ of cells were collected (4,600 × *g*, 20 min, 4 °C) and washed in buffer A (20 mM Tris-HCl pH 8, 1 mM $MgCl_2$, 150 mM KCl, 1 mM dithiothreitol). In parallel, 1 $OD_{600}$ of cells was collected as 'culture' control and boiled in 1× protein loading buffer (PL; 62.5 mM Tris-HCl, pH 6.8, 100 mM dithiothreitol, 10% (v/v) glycerol, 2% (w/v) SDS, 0.01% (w/v) bromophenol blue; 8 min at 95 °C, shaking at 1,000 r.p.m.). Next, 60 $OD_{600}$ cell pellets were lysed with a FastPrep system (MP Biomedical, matrix B, 1 × 4 m s$^{-1}$, 10 s) in 1 ml lysis buffer [buffer A including 1 mM phenylmethylsulfonyl fluoride (Roche), 20 U DNase I (Thermo Fisher), 200 U RNase Inhibitor (moloX and Triton X-100, 2 µl ml$^{-1}$ lysis buffer)]. Cleared lysates (14,500 × *g*, 10 min, 4 °C) were incubated with 35 µl anti-FLAG antibody (Sigma-Aldrich, F1804-1MG, RRID: AB_262044) for 30 min at 4 °C with rotation. Before and after incubation, a 1 $OD_{600}$ aliquot was taken aside as lysate and supernatant 1 samples. Lysates with anti-FLAG antibody were then incubated for an additional 30 min (4 °C, rotating) with 75 µl per sample pre-washed (3× in buffer A) Protein A-Sepharose beads (Sigma-Aldrich, P6649). Afterwards, the supernatant/unbound fraction was removed after centrifugation (15,000 × *g*, 1 min, 4 °C; supernatant 2) and Protein A-Sepharose beads with bound proteins were washed 5× with buffer A. Elution of the bound proteins was performed with boiling of the beads in 400 µl 1× PL (8 min at 95 °C, 1,000 r.p.m.). Six volumes of acetone were used to precipitate eluted proteins overnight at −20 °C. Next, precipitated proteins were collected by centrifugation (21,000 × *g*, 1 h, 4 °C), air-dried and resuspended in 1× PL. Culture, lysate, supernatants 1 and 2, wash (aliquots corresponding to 0.1 $OD_{600}$) and eluate samples (corresponding to 10 $OD_{600}$) were analysed by WB. Western blots were performed as described previously[63] and probed with the appropriate primary antibodies (anti-FLAG, anti-GFP (1:1,000 in 3% BSA/TBS-T)) or FlgP antisera (1:20,000 in 3% BSA/TBS-T) and secondary antibodies (anti-mouse or anti-rabbit IgG, HRP-conjugate (1:10,000) in 3% BSA/TBS-T; GE Healthcare, RPN4201 and RPN4301, respectively).

## Construction of plasmids and strains for PflA and PlfB co-immunoprecipitation experiments

Plasmids were constructed with specific promoters for expression of FLAG-tagged proteins in *C. jejuni* mutants for co-immunoprecipitation experiments. To express a C-terminal FLAG-tagged PflA protein, a 206-base-pair DNA fragment from *C. jejuni* 81–176 (containing the promoter for *flaA* encoding the major flagellin with its start codon and an in-frame SpeI restriction site followed by an in-frame BamHI restriction site) was amplified by PCR. This fragment was cloned into the XbaI and BamHI sites of pRY108 to result in pDAR1425. Primers were then constructed to amplify DNA from codon 2 to the penultimate codon of *pflA* from *C. jejuni* 81–176 with an in-frame C-terminal FLAG tag epitope and stop codon. This DNA fragment was then cloned into the BamHI site of pDAR1425 so that *pflA*-FLAG was expressed from the flaA promoter to create pDAR3417. As a control, a 229-base-pair DNA fragment from *C. jejuni* 81–176 (containing the promoter for *flaA* encoding the major flagellin with its start codon and DNA encoding an in-frame FLAG tag epitope followed by an in-frame BamHI restriction site) was cloned into the XbaI and BamHI sites of pRY108 to create pDAR1604. pDAR1604 and pDAR3417 were then moved into DH5a/pRK212.1 for conjugation into DAR1124. Transconjugants were selected for on media with kanamycin and verified to contain the correct plasmids to result in DAR3447 and DAR3477.

To express a C-terminal FLAG-tagged PflB protein, primers were constructed to amplify DNA from codon 2 to the penultimate codon of *pflB* from *C. jejuni* 81–176 with an in-frame C-terminal FLAG tag epitope and stop codon. This DNA fragment was then cloned into the BamHI site of pECO102 so that *pflB*-FLAG was expressed from the cat promoter to create pDAR3414. pDAR965 and pDAR3414 were then moved into DH5a/pRK212.1 for conjugation into DAR981. Transconjugants were selected for on media containing chloramphenicol and verified to contain the correct plasmids to result in DAR3451 and DAR3479.

## PflA and PflB co-immunoprecipitation experiments

*C. jejuni* Δ*pflA* and Δ*pflB* mutants containing plasmids to express a FLAG-tag alone or C-terminal FLAG-tagged PflA or PflB proteins were grown from freezer stocks on MH agar containing chloramphenicol for 48 h in microaerobic conditions at 37 °C. Each strain was restreaked onto two MH agar plates containing chloramphenicol and grown for 16 h at 37 °C in microaerobic conditions. After growth, strains were resuspended from plates in PBS and centrifuged for 10 min at 5,210 × *g*. Each cell pellet was resuspended in 2 ml of PBS. Formaldehyde was added to a final concentration of 0.1% and suspensions were gently mixed for 30 min at room temperature to crosslink proteins. After crosslinking, 0.4 ml of 1 M glycine was added to each sample and suspensions were then gently mixed for 10 min at room temperature to quench the crosslinking reaction. Bacterial cells were collected by centrifugation for 10 min at 5,210 × *g*. Cells were then disrupted by osmotic lysis and FLAG-tagged proteins with associated interacting proteins were immunoprecipitated with α-FLAG M2 affinity resin as previously described[88,89].

To identify potential proteins interacting with PflA and PflB, resin with immunoprecipitated proteins were resuspended in SDS-loading buffer and electrophoresed on a 4–20% TGX stain-free gel (Bio-Rad) for 10 min. The gel was then stained with Coomassie blue for 30 min and then destained overnight. After equilibration of the gel in $dH_2O$ for 30 min, a 1 cm region of the gel containing a majority of the co-immunoprecipitated proteins was excised and diced into 1 mm pieces and then submitted for analysis by LC–MS/MS. After identification of proteins that co-immunoprecipitated with the FLAG-tagged bait protein and with the resin from the FLAG-tag only sample (the

negative control), a ratio for each protein was determined by dividing the abundance of each protein detected in the FLAG-tagged bait protein sample by the abundance of each protein in the negative control. The top 20 proteins with the highest ratios for co-immunoprecipitation with the FLAG-tagged bait proteins are reported. The top 20 proteins that co-immunoprecipitated only with the FLAG-tagged bait proteins and were not detected in the negative control samples are also reported with their respective raw abundance counts.

### Reporting summary

Further information on research design is available in the Nature Portfolio Reporting Summary linked to this article.

## Data availability

CryoEM maps have been deposited at the Electron Microscopy Data Bank (EMDB) and are publicly available as of the date of publication. The whole-motor map has been deposited with accession code EMD-16723 together with the original micrographs deposited to the EMPIAR repository with public accession code EMPIAR-11580 (https://doi.org/10.6019/EMPIAR-11580). The refined periplasmic scaffold map has been deposited in the EMDB with accession code EMD-16724. Subtomogram average maps were deposited as follows: Δ*pflC* - EMD-17415; Δ*pflD* - EMD-17416; *pflA*$_{\Delta16-168}$ - EMD-17417; FlgQ-mCherry - EMD-17419. The α-carbon coordinates of the atomic model of the periplasmic scaffold have been deposited to the PDB with public accession code 9HMF. Source data are provided with this paper.

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

## Acknowledgements

We thank P. Simpson at the Imperial College London Electron Microscopy Centre and the Francis Crick Institute Structural Biology science technology platform for electron microscopy assistance. This work was supported by Medical Research Council grant MR/V000799/1 to E.J.C. and M.B., Human Frontier Science Program grant RGP0028/2021-HOCHBERG to F.P., M.B. and G.K.A.H., NIH grant R01AI065539 to D.R.H., BBSRC doctoral training grant BB/M011178/1 and a short-term DAAD grant to T.D. T.C. and P.B.R. were supported by the Francis Crick Institute, which receives its core funding from Cancer Research UK (CC2106), the UK Medical Research Council (CC2106), and the Wellcome Trust (CC2106). C.M.S. was supported by research grants Sh580/7-2 and Sh580/8-2 in the framework of the DFG (Deutsche Forschungsgemeinschaft) priority program SPP2002 (Small proteins in prokaryotes, an unexplored world), the DFG Research Training Group GRK2157 '3D-Infect', and a 'CampyRNA' junior consortium grant within the 2nd call of Infect-ERA (ERA-NET; www.infect-era.eu)/ Bundesministerium für Bildung und Forschung (BMBF; www.bmbf. de). A.L.N. was supported by two French National Research Agency (ANR) grants (PHY-BABIFO ANR-22-CE30-0034 and PHYBION ANR-23-ERCB-0005-01). The CBS is a member of France-BioImaging (FBI) and the French Infrastructure for Integrated Structural Biology (FRISBI), two national infrastructures supported by the ANR (ANR-10-INBS-04-01 and ANR-10-INBS-05, respectively).
For the purpose of open access, the authors have applied a Creative Commons Attribution (CC BY) licence to any author-accepted manuscript version arising from this submission.

## Author contributions

T.D. performed sample preparation, image processing, analysis, molecular modelling, mass photometry, and wrote the paper. E.J.C. conducted strain construction, sample preparation, image processing and analysis. T.C. performed image processing, and manuscript and figure preparation. N.S. conducted image processing. T.R.U. performed sample preparation, data acquisition and image processing. A.N. conducted data acquisition and image processing. D.R. performed flgQ-mCherry strain construction, pulldowns and fliL knockout. S.S., K.F., M.A. and C.M.S. conducted PflC and PflD identification, pulldowns and funding acquisition (C.M.S.). W.H.H., F.P. and A.L.N. performed experimental design, data acquisition, data analysis and funding acquisition. L.D.H. conducted PflD subtomogram averaging. G.K.A.H. and S.G.G. performed mass photometry data acquisition supervision and analysis, and funding acquisition. D.R.H. carried out strain construction, pulldowns and funding acquisition. P.B.R. conducted image processing, manuscript and figure preparation, and funding acquisition. M.B. performed conceptualization, funding acquisition, supervision, and wrote the paper.

## Competing interests

The authors declare no competing interests.

## Additional information

**Extended data** is available for this paper at https://doi.org/10.1038/s41564-025-02012-9.

**Correspondence and requests for materials** should be addressed to Morgan Beeby.

[1]Department of Life Sciences, Imperial College London, London, UK. [2]Structural Biology of Cells and Viruses Laboratory, The Francis Crick Institute, London, UK. [3]University of Würzburg, Institute of Molecular Infection Biology, Department of Molecular Infection Biology II, Würzburg, Germany. [4]Centre de Biologie Structurale, Universite de Montpellier, CNRS, INSERM, Mont-pellier, France. [5]Max Planck Institute for Terrestrial Microbiology, Marburg, Germany. [6]Structural Biology Science Technology Platform, The Francis Crick Institute, London, UK. [7]Department of Microbiology, University of Texas Southwestern Medical Center, Dallas, TX, USA. [8]Department of Biology, Marburg university, Marburg, Germany. [9]Present address: MRC Laboratory of Molecular Biology, Francis Crick Avenue, Cambridge, UK. [10]Present address: The Center for Microbes, Development and Health, CAS Key Laboratory of Molecular Virology and Immunology, Shanghai Institute of Immunity and Infection, Chinese Academy of Sciences, Shanghai, China. [11]These authors contributed equally: Tina Drobnič, Eli J. Cohen. ✉e-mail: mbeeby@imperial.ac.uk

**Extended Data Table 1 | Data collection statistics of two cryoEM data collection sessions, deposited collectively as EMPIAR-11580 (https://doi.org/10.6019/EMPIAR-11580)**

| Parameter | Dataset 1 ('F3') | Dataset 2 ('K2') |
|---|---|---|
| Voltage (kV) | 300 | 300 |
| Detector | Falcon III | K2 |
| Pixel size (Å/px) | 1.75 | 2.2 |
| Exposure dose per frame (e⁻/Å/frame) | 5 | 1.53 |
| Total exposure dose (e⁻/Å) | 50 | 50 |
| Defocus range (μm) | −1.5 to −3.0 | −1.5 to −3.0 |
| Total micrographs | 8,774 | 42,988 |

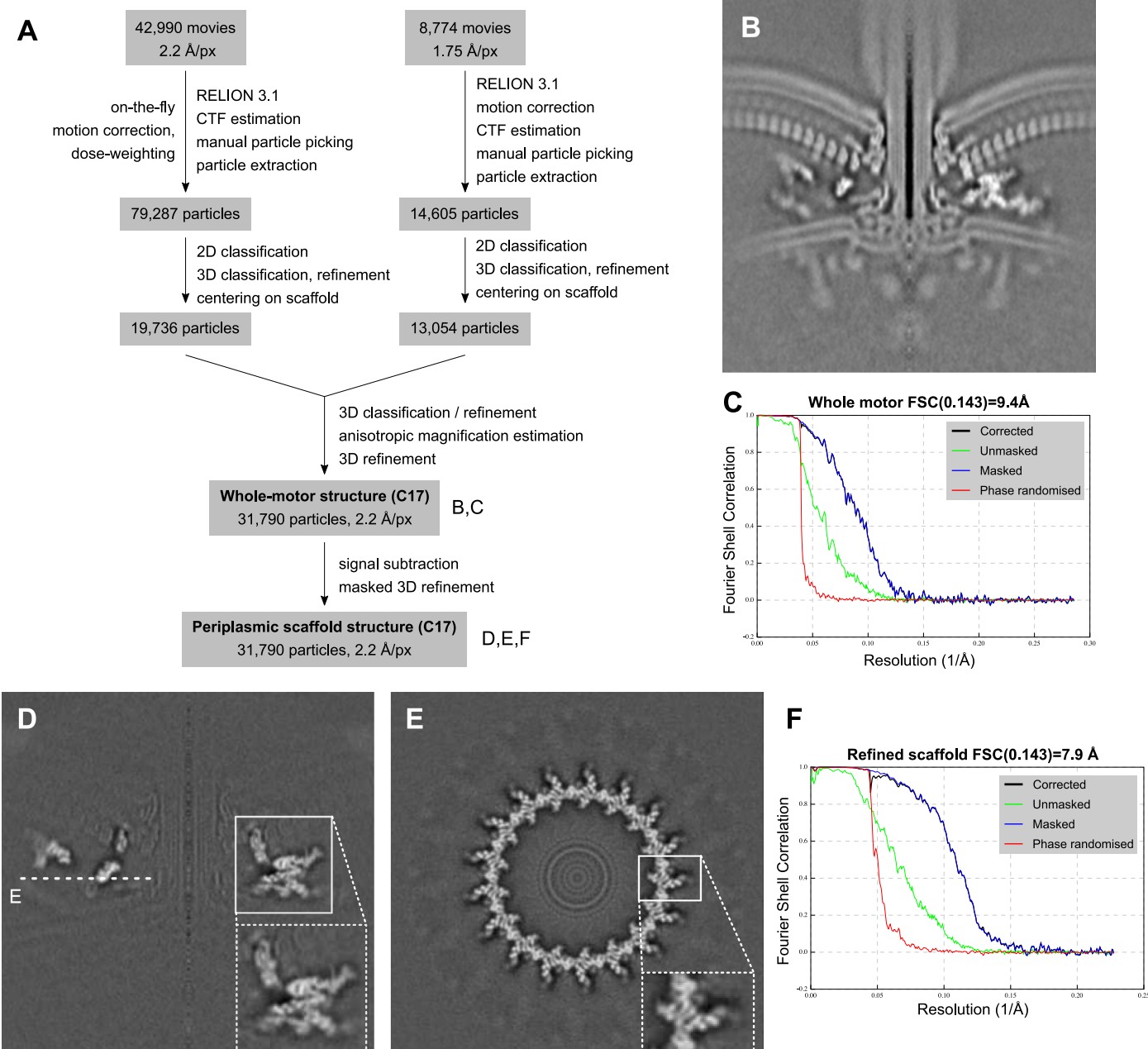

**Extended Data Fig. 1 | Flowchart and resolution estimates of structure determination of the *Campylobacter jejuni* bacterial flagellar motor using in situ single particle analysis. (a)** Simplified flowchart showing the generation of cryoEM volumes. **(b)** Central slice through the refined whole-motor structure. **(c)** FSC curve for **B**. **(d)** and **(e)** show slices through the volume of the refined, signal-subtracted periplasmic scaffold. **(f)** FSC curve for **D**, **E**.

| Description | Relevant figure | Top context | Diagonal context | Side context |
|---|---|---|---|---|
| **(A)** Periplasmic scaffold (yellow) in context of whole map (transparent) | *Fig. 1* | | | |
| **(B)** Zoom of periplasmic scaffold alone | *Fig. 1* | | | |
| **(C)** Periplasmic scaffold asymmetric unit | *Fig. 2-4,6* | | | |
| **(D)** Innermost basal disk ring of 51 FlgP | *Fig. 2* | | | |
| **(E)** Innermost basal disk ring of 51 FlgP and innermost medial disk ring of 17 PflC$_1$ | *Fig. 3* | | | |
| **(F)** Innermost basal disk ring of 51 FlgP and medial disk lattice of 17 PflC$_{1-7}$ | *Fig. 3* | | | |
| **(G)** Proximal disk of 17 PflAB FliL$_4$ | *Fig. 4* | | | |

**Extended Data Fig. 2 | See next page for caption.**

**Extended Data Fig. 2 | Global context of components discussed in the manuscript. (a)** Context of the periplasmic scaffold focussed refinement map (EMD-16724) in the context of the whole motor map (EMD-16723). **(b)** Structure of the periplasmic scaffold density in isolation. **(c)** Location of one asymmetric unit from the 17-fold-symmetric periplasmic scaffold in pink as depicted in

Figs. 2a, 3a, 4a, and 6. **(d)** Innermost basal disk ring of 51 FlgP as depicted in Fig. 2. **(e4)** Innermost basal disk ring of 51 FlgP and innermost medial disk ring of 17 PflC$_1$ as depicted in Fig. 3c. **(f)** Innermost basal disk ring of 51 FlgP and medial disk lattice of 17 PflC$_{1-7}$ as depicted in Fig. 3d, f. **(g)** Proximal disk of 17 PflAB FliL$_4$ as depicted in Fig. 4b, c.

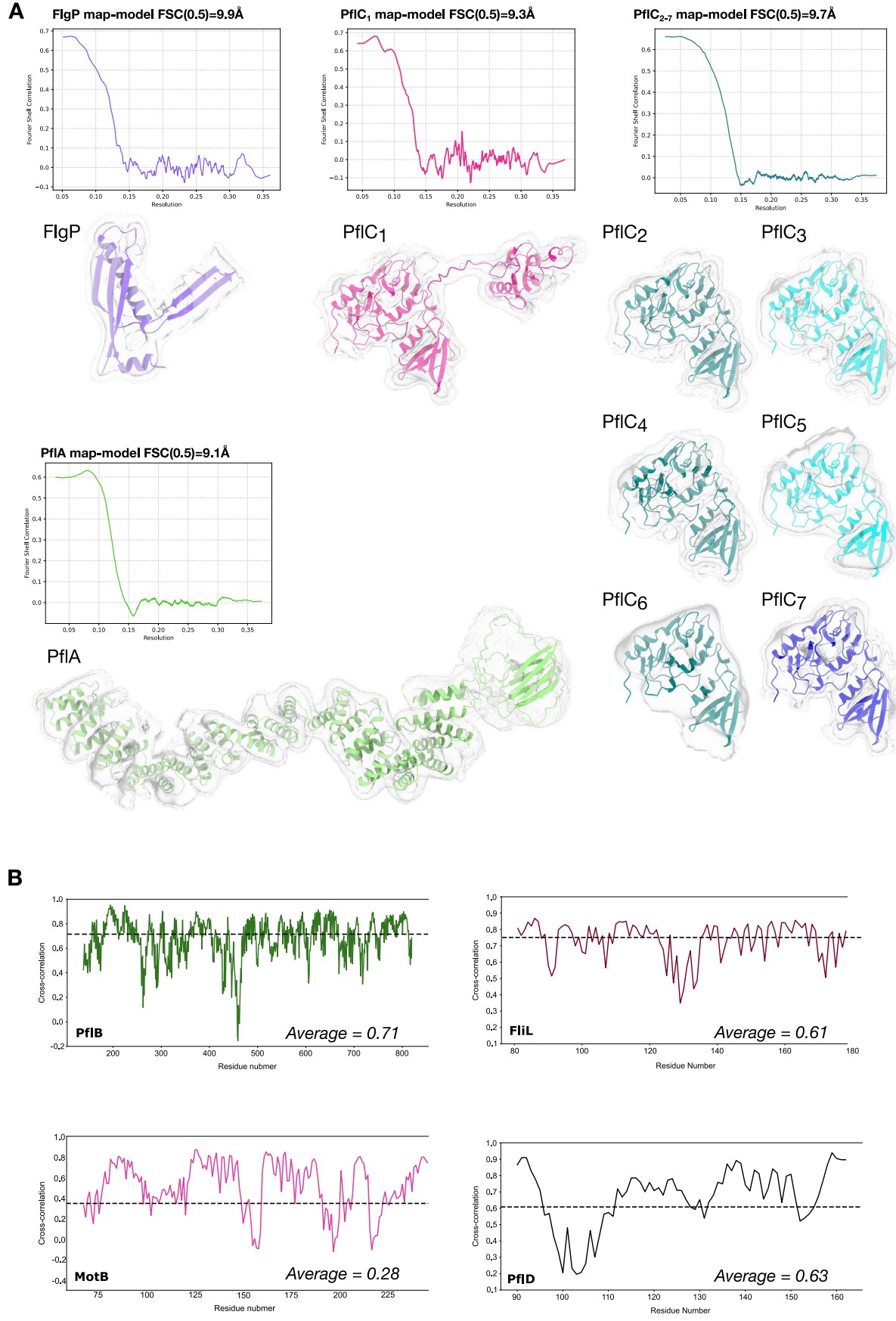

**Extended Data Fig. 3 | See next page for caption.**

**Extended Data Fig. 3 | Validation of those protein chains modelled in the scaffold map to subnanometre resolution. (a)** Map-model FSC curves for protein models refined into the scaffold map: FlgP, PflA, PflC$_1$, and PflC$_{2-7}$ calculated in Phenix[83], alongside images of excised pieces of the density map corresponding to individual proteins. High and low isosurface thresholds are denoted by solid and mesh surfaces, respectively, to illustrate fit of models into secondary structure densities. **(b)** Cross-correlation per residue plots of proteins docked into the map: PflB, FliL, MotB, PflD, calculated using Phenix. Mean CC values are shown with dashed line and in text inset.

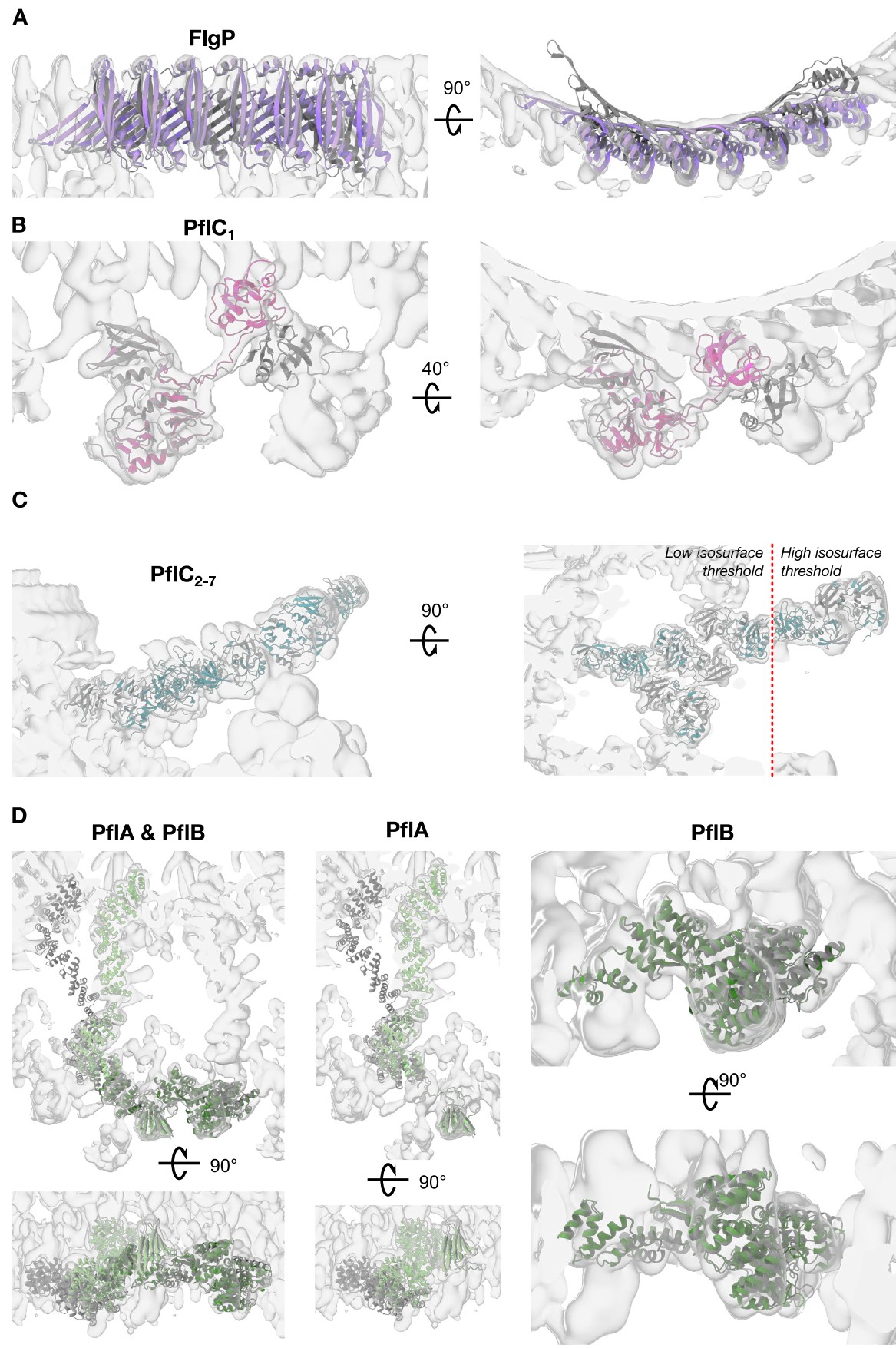

**Extended Data Fig. 4 | See next page for caption.**

**Extended Data Fig. 4 | Comparison of starting predicted protein models with final structures after flexible fitting into density map.** (**a**) AlphaFold2 (ref. 29) prediction of FlgP (dark grey) required bending of the oligomer to fit the curvature of the first ring of the basal disk (purple). (**b**) AlphaFold2 prediction of $PflC_1$ required independent rigid-body docking of the N- and C-terminal domains (magenta). (**c**) The resulting N-terminal domain of PflC was rigid-body docked into density multiple times for $PflC_{2-7}$. C-terminal domains were not modelled. Higher thresholds required for more peripheral, and presumably more flexible, components. (**d**) PflA and PflB AlphaFold2 models (dark grey) required bending to fit into density maps (pale and dark green for PflA and PflB, respectively).

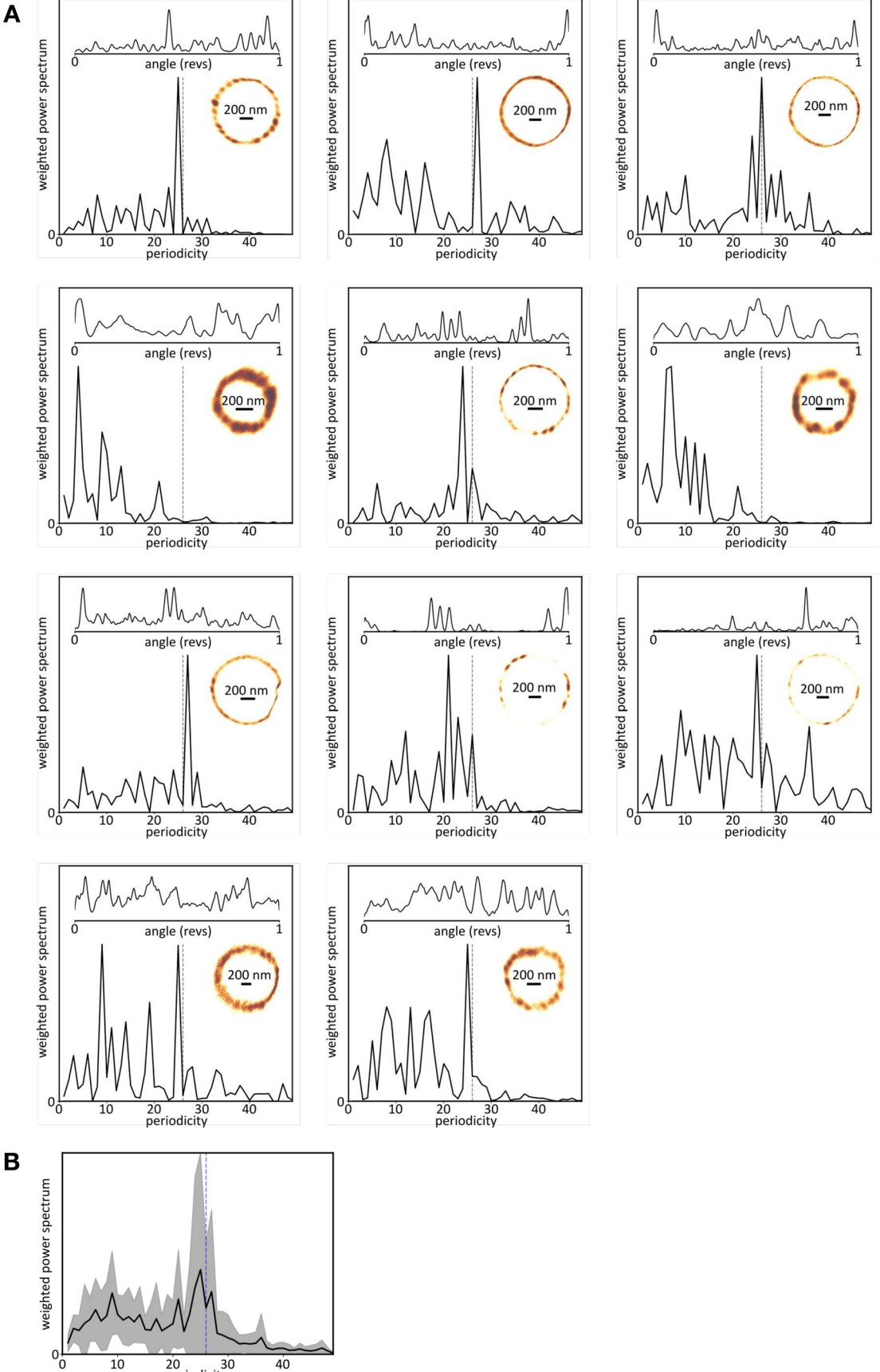

**Extended Data Fig. 5 | Discrete steps in flagellar rotation are similar to those in *Escherichia coli.* (a)** Kernel density estimation bead position plots over many rotations for 11 cells. Measurements were of duration 20 to 240 s, substantially de-energized by CCCP to slow rotation. Their (x,y) histograms, a kernel density of the angular position, and the weighted power spectrum are depicted. The top left is the trace shown in Fig. 2. **(b)** Weighted power spectrum of all 11 traces.

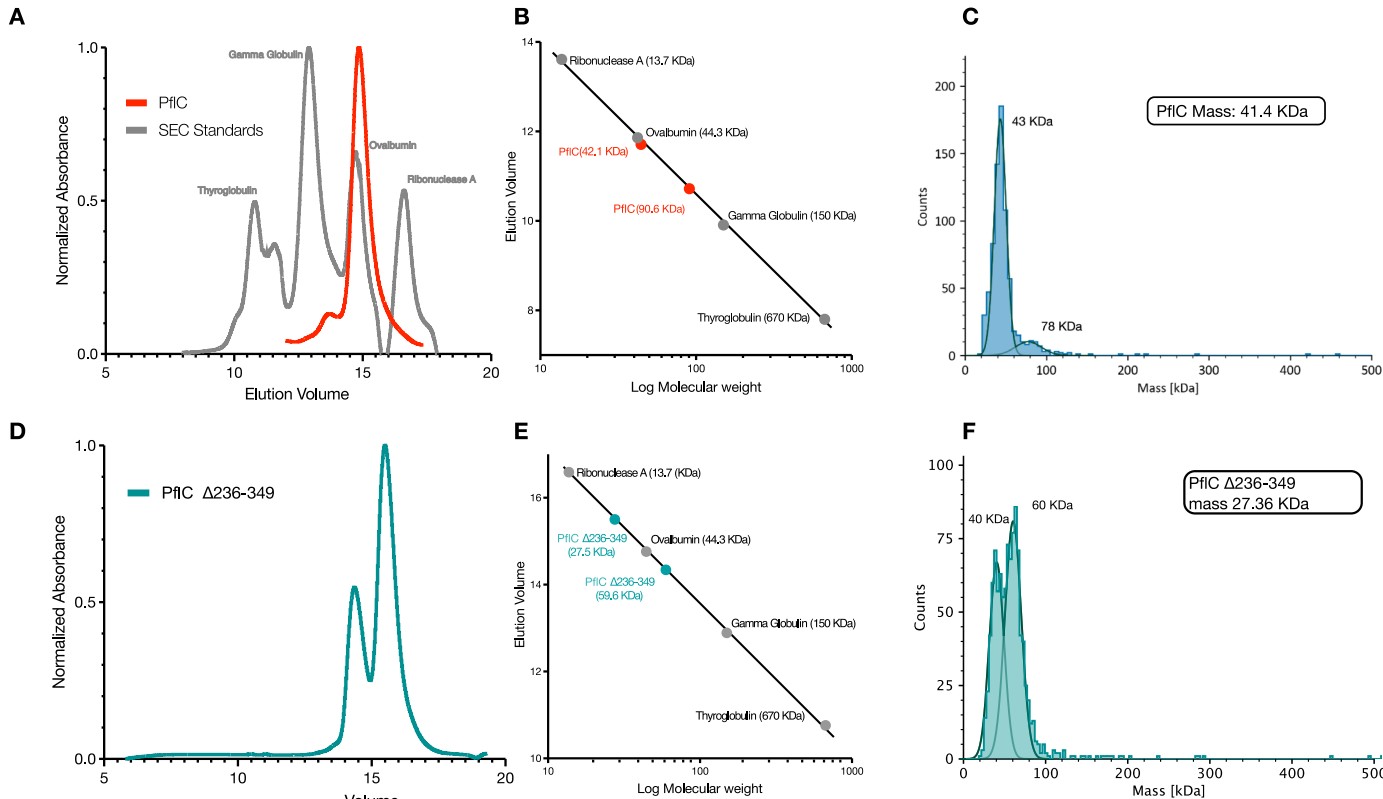

**Extended Data Fig. 6 | PflC forms oligomers *in vitro* regulated by its C-terminal PDZ domain. (a)** Size Exclusion Chromatography of PflC (red) along with protein standards (grey) showing elution volume. **(b)** Calibration graph showing a dimer of PflC (red) corresponding to retention times during elution. **(c)** Mass photometry measurements of purified PflC (replicates). **(d)** Size Exclusion Chromatography of PflC$_N$ (Δ236–349, green) along with protein standards (grey) showing elution volume. **(e)** Calibration graph of PflC$_N$ (green) corresponding to retention times during elution. **(f)** Mass photometry measurements of purified PflC$_N$. Theoretical masses are shown in the inserts. The instrument's limit of detection is 35 kDA, meaning the monomer mass is larger than expected.

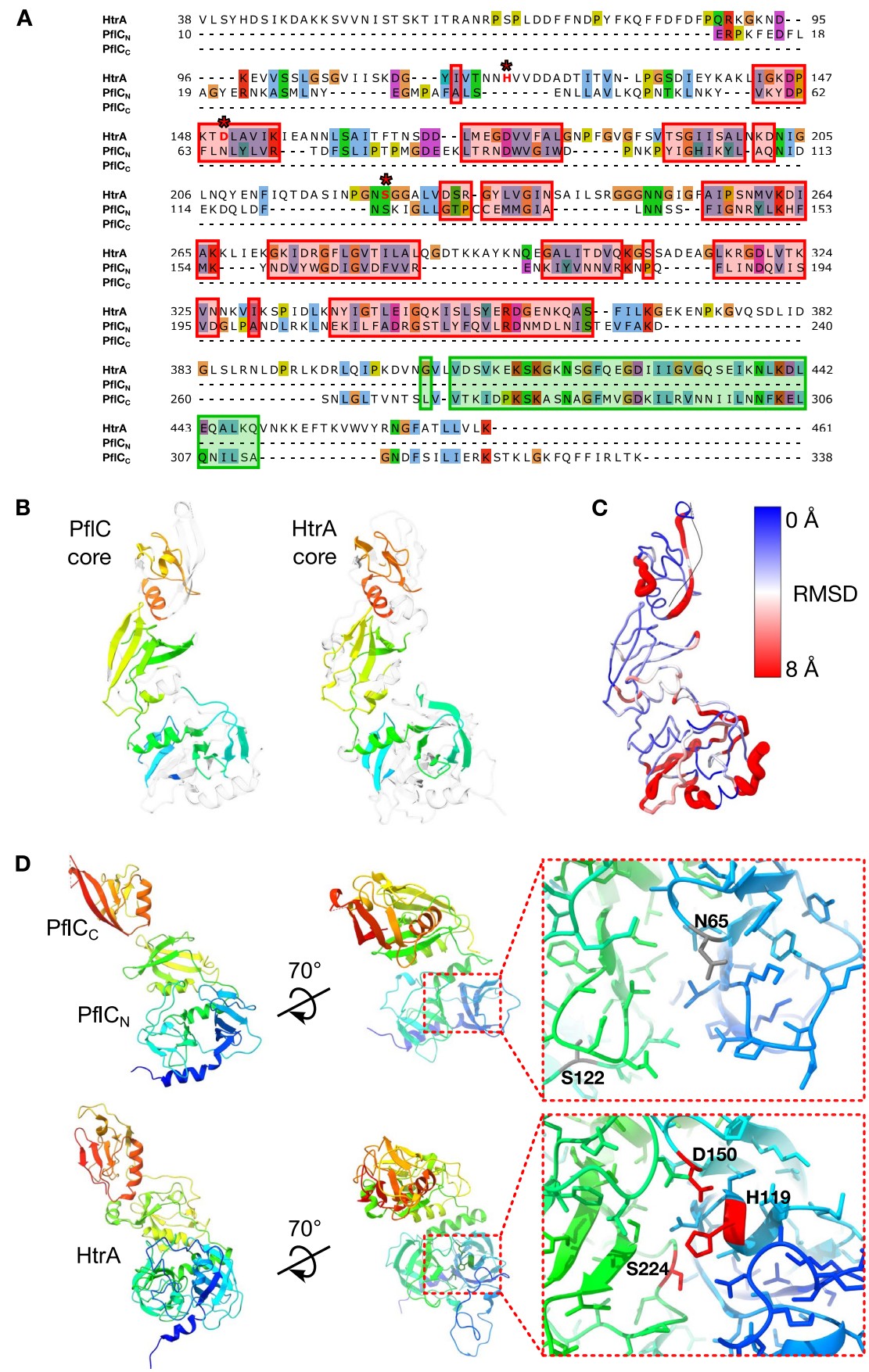

**Extended Data Fig. 7 | See next page for caption.**

**Extended Data Fig. 7 | PflC is a homolog of serine protease HtrA but lacking a catalytic triad. (a)** Needleman-Wunsch global sequence alignments of *Campylobacter jejuni* PflC$_N$ and PflC$_C$ with *C. jejuni* HtrA from ChimeraX MatchMaker[91] using default parameters. Asterisks highlight HtrA His-Asp-Ser catalytic triad not conserved in PflC. Coloured boxes indicate pruned residue pairs between PflC$_N$:HtrA$_N$ (red) and PflC$_C$:HtrA$_C$ (green) from MatchMaker algorithm using a 4 Å RMSD cutoff **(b)** Core folds of PflC and HtrA from ChimeraX structural alignment using pruned residue pairs of PflC$_N$ and PflC$_N$ (left) to HtrA (right) (PDBID: 6Z05[36]) guided by the global alignment in panel A. Pruned core

residues are depicted in colour, while non-core residues are depicted in grey. **(c)** RMSD of PflC alignment to HtrA depicted on the structure of PflC as worm diameter and colour (blue: 0 Å, white: 4 Å, red: 8 Å). **(d)** Left panel shows PflC and HtrA oriented as with the PflC protomer in Fig. 3c Right panels depict PflC and HtrA rotated similarly to reveal the HtrA active site cleft. Close-up boxes include stick representation of residues, highlighting HtrA's H119:D150:S224 catalytic triad, and PflC's N65 and S122 residues that sequence align to D150 and S224, revealing divergence and atrophy of the active site region in PflC.

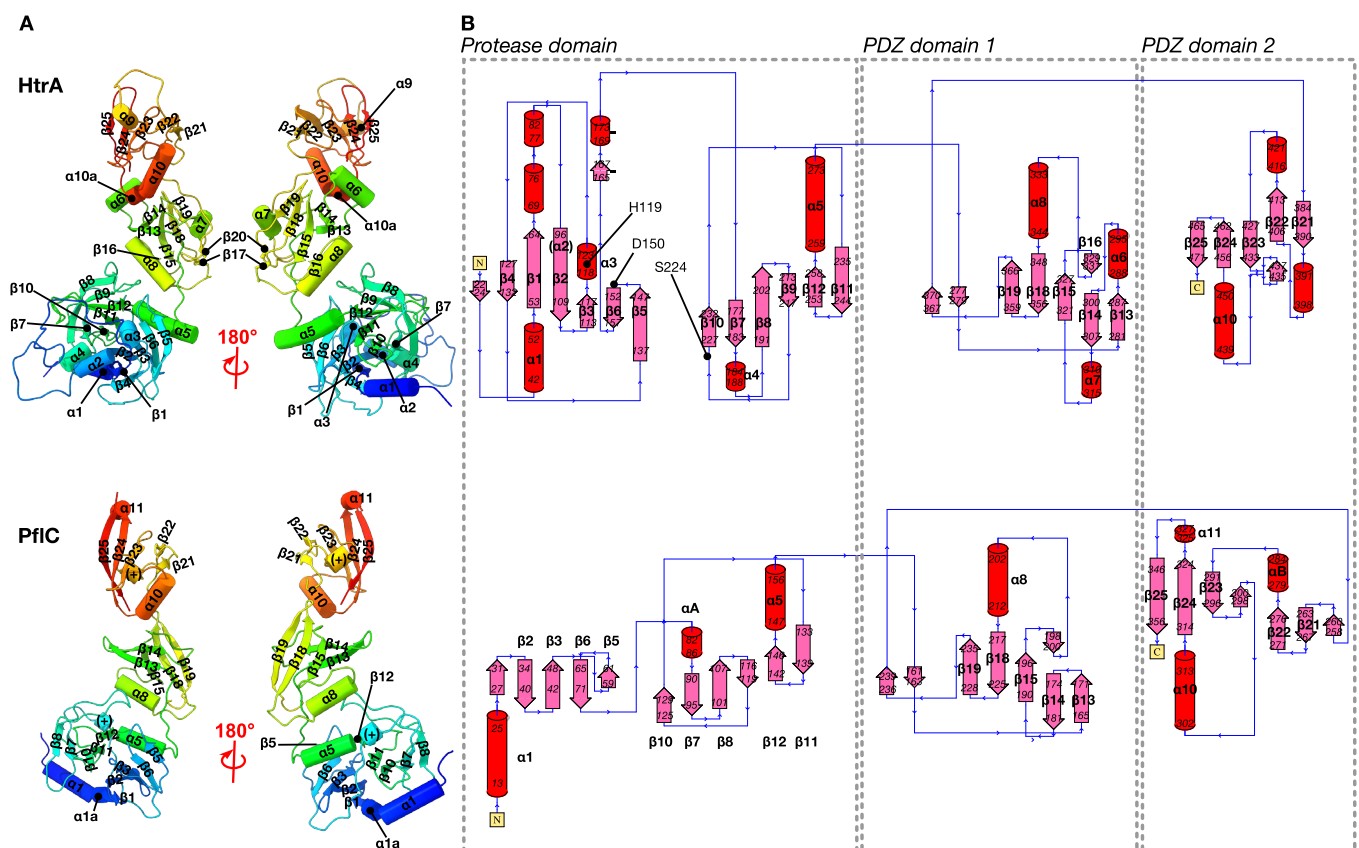

**Extended Data Fig. 8 | PflC and HtrA share a common fold and topology.**
**(a)** Automated secondary structure annotation of HtrA using the ChimeraX implementation of the Defining the Secondary Structure of Proteins algorithm (top); and PflC (bottom) with HtrA nomenclature transferred based on structural correspondence. **(b)** Topology diagram of HtrA and PflC generated using PDBsum[92]. Note that due to algorithmic nuances, PDBsum's secondary structure definitions do not exactly correspond to those of DSSP. Locations of the three catalytic triad residues H119, D150 and S224 are labelled.

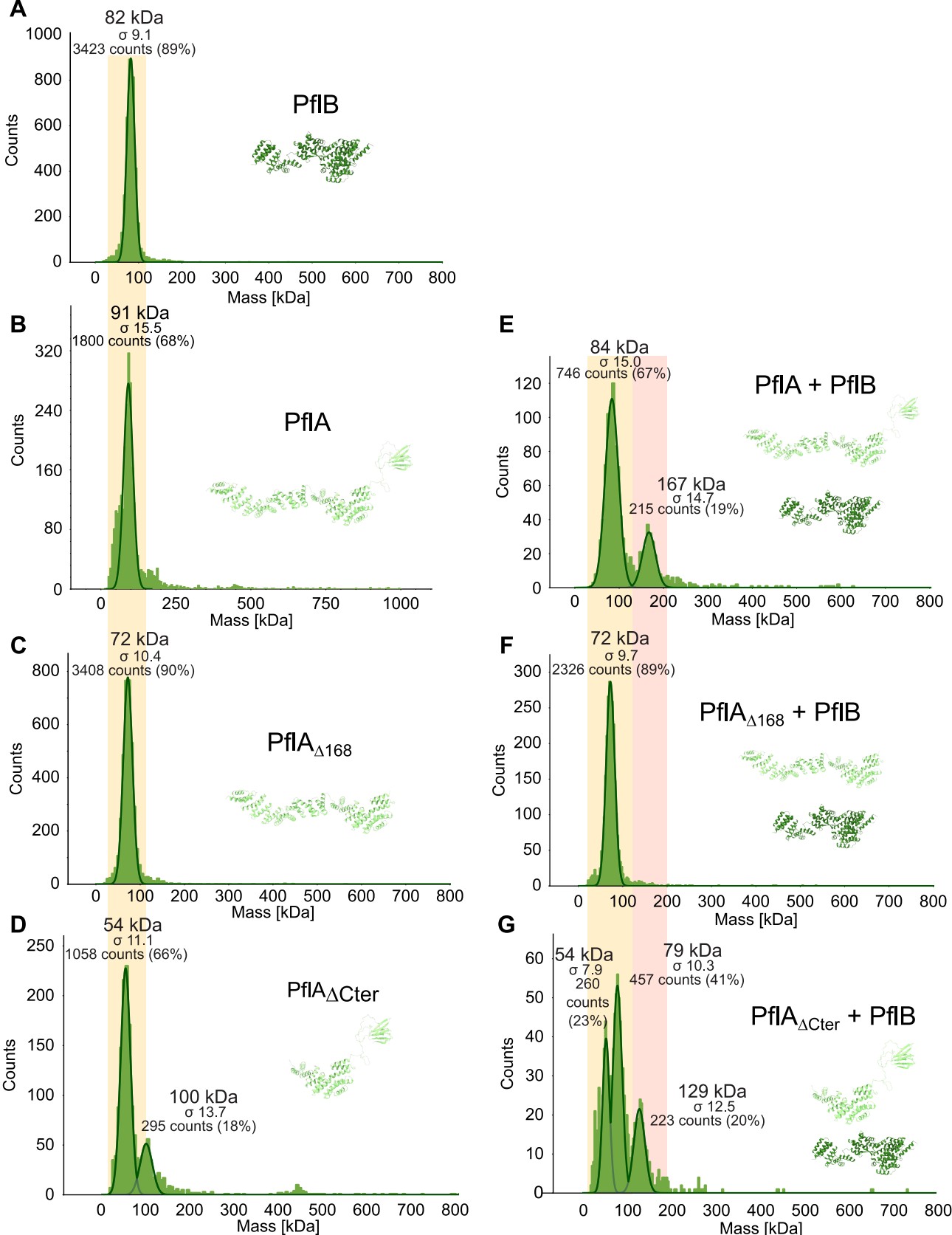

**Extended Data Fig. 9 | Mass photometry shows that PflA dimerises with PflB via its N-terminal β-sandwich domain. (a–d)** Mass photometry measurements of purified PflA and PflB constructs show the proteins are mainly monodisperse. There is a dimer peak present for the PflA$_{\Delta Cter}$ construct, likely due to a reduction in stability and solubility. **(e–g)** Mass photometry measurements of mixtures of PflB and PflA variants. Dimer peaks appear only when β-sandwich and linker domain of PflA is present. In panels **E** and **G**, monomer peaks of PflA and PflB are not resolved due to their similar molecular weights. In the bottom panel, the 54 kDa peak corresponds to PflA$_{\Delta Cter}$, and the 79 kDa peak to PflB. Broadly, monomer peaks have a yellow background, dimer peaks red.

# Reporting Summary

## Statistics

For all statistical analyses, confirm that the following items are present in the figure legend, table legend, main text, or Methods section.

| n/a | Confirmed | |
|---|---|---|
| ☐ | ☒ | The exact sample size (*n*) for each experimental group/condition, given as a discrete number and unit of measurement |
| ☒ | ☐ | A statement on whether measurements were taken from distinct samples or whether the same sample was measured repeatedly |
| ☒ | ☐ | The statistical test(s) used AND whether they are one- or two-sided<br>*Only common tests should be described solely by name; describe more complex techniques in the Methods section.* |
| ☒ | ☐ | A description of all covariates tested |
| ☒ | ☐ | A description of any assumptions or corrections, such as tests of normality and adjustment for multiple comparisons |
| ☒ | ☐ | A full description of the statistical parameters including central tendency (e.g. means) or other basic estimates (e.g. regression coefficient) AND variation (e.g. standard deviation) or associated estimates of uncertainty (e.g. confidence intervals) |
| ☒ | ☐ | For null hypothesis testing, the test statistic (e.g. *F*, *t*, *r*) with confidence intervals, effect sizes, degrees of freedom and *P* value noted<br>*Give P values as exact values whenever suitable.* |
| ☒ | ☐ | For Bayesian analysis, information on the choice of priors and Markov chain Monte Carlo settings |
| ☒ | ☐ | For hierarchical and complex designs, identification of the appropriate level for tests and full reporting of outcomes |
| ☒ | ☐ | Estimates of effect sizes (e.g. Cohen's *d*, Pearson's *r*), indicating how they were calculated |

*Our web collection on statistics for biologists contains articles on many of the points above.*

## Software and code

Policy information about availability of computer code

| | |
|---|---|
| Data collection | All data collection software is fully described in Methods. In brief, single particle analysis data was acquired using EPU (Thermo Fisher Scientific). Tomography data was acquired with either Tomo5 (Thermo Fisher Scientific) or Leginon (developed by an independent group, cited appropriately). |
| Data analysis | All data analysis software is fully described in Methods. In brief, single particle analysis was performed entirely using RELION 3.1 (cited appropriately in Methods). Subtomogram averaging was performed in IMOD and PEET (cited appropriately). |

For manuscripts utilizing custom algorithms or software that are central to the research but not yet described in published literature, software must be made available to editors and reviewers. We strongly encourage code deposition in a community repository (e.g. GitHub). See the Nature Portfolio guidelines for submitting code & software for further information.

## Data

Policy information about availability of data

All manuscripts must include a data availability statement. This statement should provide the following information, where applicable:
- Accession codes, unique identifiers, or web links for publicly available datasets
- A description of any restrictions on data availability
- For clinical datasets or third party data, please ensure that the statement adheres to our policy

Cryo-EM maps have been deposited at the Electron Microscopy Data Bank (EMDB) and are publicly available as of the date of publication. The whole-motor map

has been deposited with accession code EMD-16723 together with the original micrographs deposited to the EMPIAR repository with public accession code EMPIAR-11580 (DOI: 10.6019/EMPIAR-10016). The refined periplasmic scaffold map has been deposited in the EMDB with accession code EMD-16724. Subtomogram average maps were deposited as follows: ΔpflC - EMD-17415; ΔpflD - EMD-17416; pflAΔ16-168 - EMD-17417; FlgQ-mCherry - EMD-17419.  The α-carbon coordinates of the atomic model of the periplasmic scaffold has been deposited to the PDB with public accession code 9HMF.

# Research involving human participants, their data, or biological material

Policy information about studies with human participants or human data. See also policy information about sex, gender (identity/presentation), and sexual orientation and race, ethnicity and racism.

| Reporting on sex and gender | N/A |
|---|---|
| Reporting on race, ethnicity, or other socially relevant groupings | N/A |
| Population characteristics | N/A |
| Recruitment | N/A |
| Ethics oversight | N/A |

Note that full information on the approval of the study protocol must also be provided in the manuscript.

# Field-specific reporting

Please select the one below that is the best fit for your research. If you are not sure, read the appropriate sections before making your selection.

☒ Life sciences   ☐ Behavioural & social sciences   ☐ Ecological, evolutionary & environmental sciences

For a reference copy of the document with all sections, see nature.com/documents/nr-reporting-summary-flat.pdf

# Life sciences study design

All studies must disclose on these points even when the disclosure is negative.

| Sample size | We acquired approximately 95,000 particles for single particle analysis limited primarily by microscope access time. |
|---|---|
| Data exclusions | Particles were rejected during 2-D and 3-D classification as per field conventions as outlined in Extended Data Figure 1. |
| Replication | All statistics are calculated by splitting the dataset into two halves as per field convention of 'Gold Standard Fourier Shell Coefficient' as reported in Methods. |
| Randomization | N/A |
| Blinding | N/A |

# Reporting for specific materials, systems and methods

We require information from authors about some types of materials, experimental systems and methods used in many studies. Here, indicate whether each material, system or method listed is relevant to your study. If you are not sure if a list item applies to your research, read the appropriate section before selecting a response.

## Materials & experimental systems

| n/a | Involved in the study |
|---|---|
| ☐ | ☒ Antibodies |
| ☒ | ☐ Eukaryotic cell lines |
| ☒ | ☐ Palaeontology and archaeology |
| ☒ | ☐ Animals and other organisms |
| ☒ | ☐ Clinical data |
| ☒ | ☐ Dual use research of concern |
| ☒ | ☐ Plants |

## Methods

| n/a | Involved in the study |
|---|---|
| ☒ | ☐ ChIP-seq |
| ☒ | ☐ Flow cytometry |
| ☒ | ☐ MRI-based neuroimaging |

## Antibodies

| Antibodies used | FlgP specific antisera Ref21 FlgP specific antisera<br>Anti-GFP Roche #11814460001, RRID:AB_390913<br>Anti-FLAG Sigma-Aldrich #F1804-1MG, RRID:AB_262044<br>Anti-mouse IgG GEHealthcare #RPN4201<br>Anti-rabbit IgG GEHealthcare #RPN4301 |
|---|---|
| Validation | As per above for the four commercial antibodies. FlgP specific antisera derives from Sommerlad, S. M. & Hendrixson, D. R. "Analysis of the roles of FlgP and FlgQ in flagellar motility of Campylobacter jejuni." J. Bacteriol 189, 179–186 (2007), in which wild-type and FlgP-deficient strains as positive and negative controls demonstrate positive and negative responses to the antisera respectively. |

## Plants

| Seed stocks | N/A |
|---|---|
| Novel plant genotypes | N/A |
| Authentication | N/A |

