## [Peer Review File · Nature Microbiology]

In situ structure of a bacterial flagellar motor at subnanometre resolution reveals adaptations for increased torque

Corresponding Author: Dr Morgan Beeby

Version 0:

Reviewer comments:

Reviewer #1

(Remarks to the Author)

This is a very nice piece of work revealing the in situ flagellar motor structure of *Campylobacter jejuni* in sufficient detail to allow identification of component proteins by docking AlphaFold predicted models into the cryoEM density map. The authors have done a good job producing minicells of *Campylobacter jejuni* at minimal sizes close to 200 nm so that they can carry out cryoEM single particle analysis of the flagellar motor structure, instead of tomography data collection and subtomogram averaging as usually done for in situ structural analysis, to successfully achieve a sub nanometer resolution. The density map is impressively visualizing the secondary structures or domains of many of the component proteins to allow unambiguous docking of predicted models. Although the depth of biological and functional insights into the mechanism of torque generation by the flagellar motor is limited due to the limited resolution, this study still reveals much greater detail of the flagellar motor structure of *Campylobacter jejuni* than those obtained in previous studies by tomography of normal cells to advance our understanding of the structural and functional variations of the flagellar motor between bacterial species.

The authors also faithfully revised their original manuscript submitted to other journal according to the comments by reviewers, and their responses are satisfactory. I have a few minor points listed below for the final revision of the manuscript.

1. Lines 135-137: The authors predict that each subsequent rings of FlgP incorporates 11 additional protomers, but this could be examined by applying +11-fold symmetry to the 17-fold of the innermost ring. Has this been tried?
2. Lines 142 and 336: "second beta-sheet" should be "second strand of the beta-sheet".
3. Paragraph from line 308: A reference should be cited for "approximately three times the torque of the E. coli and Salmonella motors". Also, if the structure of *C. jejuni* flagellar motor can explain the three times higher torque, the authors should give a quantitative mechanical explanation by referring to the number of stator units and the diameter of the C-ring.
4. Paragraph from line 341: The consistent rotational register of the MotA pentamer to the C-ring edge could also be interpreted as all the MotA pentamers are bound to the C-terminal torque helix of FliG. This possibility should be considered and discussed here.
5. Line 353: It is not clear what is meant by "our projection imaging approach".
6. Line 358: "such as the MS-ring"?
7. Lines 367-368: "provide invaluable information on the mechanisms of torque generation" seems to be an overstatement.
8. Lines 369-371: It may be good to insert the words "without sectioning" after "subnanometer resolution" to make is a bit more precise.

Keiichi Namba

Reviewer #2

(Remarks to the Author)

The manuscript "Molecular model of a bacterial flagellar motor in situ reveals a parts-list of protein adaptations to increase torque," from the Beeby lab uses a delta-flhG mutation to produce large numbers of *C. jejuni* flagellar motors in mini-cells, then determines the cryoEM structures. This is a major tour de force and provides the highest resolution view of a fully intact flagellar motor in situ. Part of this work discovers roles for two previously unassigned proteins, now FlgC/D. The introduction sets up biological questions about how the *C. jejuni* motor achieves high torque and how motors might have evolved. The authors also appear to want to discuss accompanying technological advances. This turns the results section into a true 'parts-list' that describes the different regions of the motor, the methods to collect data, and the strategy used to assign and interpret the maps. The authors seem to run out of space and don't squarely address the biological questions that they pose in the intro. I recognize that it is a challenge to pick and choose when there is a massive amount of information and a short report format. Despite these concerns and the detailed comments below, I think that this manuscript is well suited to a broad audience and could help explain

differences in chemotaxis behavior between different motile bacteria. The structure itself is an absolutely fantastic accomplishment. Most concerns directly or indirectly relate to the lack of focus of the text in the context of a short format, which makes this major advance feel smaller than it actually is.

Points for the authors to consider.

HtrA. The HtrA analysis felt tangential to the main manuscript and lacks clarity on whether the authors are proposing convergent or divergent evolution. Moreover, the figures do not convincingly illustrate the authors' claims of a shared fold and some of the depictions do not match the legend. For instance, in the ED5A legend, the term "local sequence realignments" suggests manual manipulation, which absolutely compromises a structure-based alignment. Closer inspection identifies that this almost certainly cannot be a proper structure-based sequence alignment. Specifically, ED5C shows the HtrA secondary structural elements containing the catalytic residues, i.e. a helix containing the catalytic His and a loop containing the Ser. These secondary structure elements appear to be absent in the overlaid PflC in the same ED5C panel. But panel ED5A shows the HtrA sequences aligned to something in the PflC sequence as if the catalytic residues are missing due to a point mutation. The text similarly discusses these as if it is a point mutation. Overall, the ED5A alignment seems forced, dividing into two non-homologous groups; there are limited sequence matches, excessive gapping, and no information on percent identity or similarity. This could indicate convergent evolution to a stable fold. If so, one could expect relatively poor sequence identity with helices and strands in the same order, but with conserved anchor residues necessary for this fold. An example is in Fig 1 of PMID 8880921, which is an early analysis of convergent evolution of Ig folds. Visual assessment of the structural alignment and side-by-side structures makes it appear that these have little in common beyond two helices and a size/shape similarity. It is difficult to tell for certain because 3D, ED5B, and ED5D are incredibly small. The pair-wise RMS deviation of $>10 \text{ \AA}$ would also support these not sharing a fold. While the pruned similarity may indicate a shared fold, the authors do not specify the pruning level, residue count, or show the elements remaining after pruning, making it hard to evaluate potential over-pruning. If the analysis is retained, more supporting information would be necessary. The sequence alignment needs to be fully structure-based and not additionally manipulated. Anchor residues are the most critical part of the analysis of convergent evolution and need to be marked in the alignment, secondary structural elements need to be labeled in numerical order on both the sequence alignment and the structural panels, the size of the panels that show structural comparisons need to be increased, and the pruned core fold from each comparator needs to be shown side-by-side.

Given that the authors indicate that their major biological question was about evolutionary adaptations that affect torque, the rationale for not including more detailed analyses that leverage available structures of different parts of the low-torque motors as comparators was not clear. Are the major differences between this motor and lower-torque motors the larger gear size and a buffering piece on the membrane? If so, this reduces the number of changes that an organism needs to make to move from low- to high-torque, particularly when it can co-opt the buffer via horizontal transfer. And it's obviously easier to move from high- to low-torque. The only comparisons shown were in tomogram side views, which lack the detail needed to make these interpretations.

MotA/B. The description of MotA/B in the text is confusing and the accompanying figures currently don't do a good job of illustrating the authors' points. What is meant by rotational register? Do the authors mean that the asymmetry of the pentameric MotA, as is induced by the central MotB, always aligns so that structurally similar face engages FliG? If so, this is not only unclear but is expected given the information in the literature. MotA rotates around MotB during transmembrane proton/Na transfer in a way that is conceptually analogous to the ATP synthase, as concomitantly described by both the Lea and Taylor groups (ref 9/10) and as supported by the isolated and symmetric MotA structure (PMID 36179499). If the authors do mean that the same conformational face of MotA/B engages the C-ring across the particle, this would be the first time that it is directly shown around the intact ring because the (unmentioned) MotA/B-FliG structure from the Lea group (ref 8) only use a resected FliG domain to show that a consistent conformational face of MotA interacting with FliG. This might just be a text and presentation issue.

The manuscript is under-referenced throughout, and sometimes the findings of the cited papers are overgeneralized. As an example (not limited to this, but this is the first one in the text), the introduction cites refs 7/8 for the isolated C-ring, but the isolated C-ring citations should be reference 8 and PMID 39179739. Ref 7 does contain an isolated C-ring but also contains the combined MS- and C-ring. Reference 8 contains the MotA/B-FliG structure, and the unreferenced PMID 39179739 also contains CheY. Following the introduction, the text only uses reference 8 when referring to the C-ring but all three citations are appropriate each time. Figure legends are fully unreferenced.

Many figure legends did not adequately describe the figure components, sometimes instead stating the authors' interpretations or conclusions. This made it difficult to understand many figure panels. For example, Fig. 3A appears to show a cross-section of the motor perpendicular to the membrane. The legend states: "The medial disk is situated between the basal disk and proximal disk."

Along the same lines, many figures contain red boxes, but it is not always clear what those are. One can eventually figure out that these point to another panel, but the view isn't always the same and there is not clarification when this view is rotated. The first example is Fig 2A, which has a red dotted box labeled 'B'. There is a red box in panel B that appears to be on the same protein, but a different view and different part of the ring (or rotated -90° around the y axis?). The rationale for red boxes in 2F is unclear, as is the reasons that FliG is on a different scale, and that the proteins are not shown in the same order they are discussed in the text.

Many figure panels were too small to see.

The main text accompanying Fig 2I implies that the $1-2\text{\AA}$ difference is statistically significant at 8\AA resolution without coordinates assigned (although the legend indicates these are similar). This should be within error. See PMID 33559604.

PflC/D are described as proteins of previously unknown function, with the discovery of their role reported here. I find this quite interesting. Can bioinformatics identify how widespread these are? Are there homologs in other organisms, are these common in genomes of most motile bacteria, or are these unique to high-torque motors? How does this inform on the question of torque? Would this predict a low- versus high-torque bacterium?

The FlgQ analysis also feels tangential. It basically involved experiments with negative results and the authors indicate that FlgQ was not observed in this structure.

If molecular models are shown in figures, the standard in the field is that PDB files are deposited at www.rcsb.org. These would be limited to alpha-carbons or polyanines given the reported resolution. For the authors consideration, both a full ring and an individual unit could be separately deposited, with the latter essential in making these results useful to the scientific community. Currently, only EMDB and EMPIAR codes are given in the manuscript. Anyone wishing to make comparisons to this structure would need to do an independent de novo chain trace and map interpretation. The full ring is important for less experienced users who will have trouble building this from a single unit. Given the number of domain swaps, an individual unit is also not all that helpful.

In the title and throughout, the authors indicate that they are investigating torque. Most of the text revolves around speed and power. Torque is a vector, so a nod to the directional component could be appropriate in the discussion.

It is not clear from the text that only the top half of the structure is assigned with models. The decision to prioritize this aspect of the assignment may have been pragmatic due to resolution, but it is also a reasonable choice for prioritization because the newly assigned section is unique as compared to high-resolution structures of component parts. Nevertheless, adding text clarifying what is and is not assigned with molecular models is important in the context of the title and introduction, which indicates that the manuscript investigates ways that *C. jejuni* will "increase" torque. But the models, and most of the text, focuses on the adaptations that function to buffer (not increase) torque. The power-generating part of the motor, which actually increases the torque, does not appear to be assigned. It doesn't necessarily need molecular assignment as the analysis about gear size stands without it, but the concept could be better highlighted. I am hoping that the resolution of the maps is sufficient to allow the authors to complete the model assignment for a future publication; it would be amazing to see the full machine some day.

Because of the way that most figure panels are deconstructed to highlight one of the authors' points, in some cases, it can be hard to get a sense of where the specific features are within the intact motor. I realize that some of the figure panels make a strong attempt at providing this context and it may not be possible to improve on this. Is it reasonable to include a few global views of the maps with different rotations and an overview in the ED that is an intact motor highlighting what each main text figure will focus on. Also is it reasonable to have a map in the ED that is colored by resolution?

When referring to other structures, PDB codes are not associated with their citations throughout the manuscript. In some cases, it was not clear whether models were alphaFold predictions or experimentally determined structures. For example, lines 186-189 describe PflC using language normally reserved for experimental structures, but the authors also report that they have discovered the role of this protein and I don't see either PflC or cj1643 when searching the PDB. No citation is given. If these are alphaFold predictions, did the authors use a confidence cutoff? Also, if alphaFold, the program should be referenced or the DOI given if the prediction is among those in the alphaFold database.

Line 234 and Figure 4. This is quite hard to interpret likely because the color used for MotB blends into its surroundings and it is not highlighted. Because of this, it took some time to find MotB in the figure. Also, the periplasmic domain is not associated with the past cryoEM structures of isolated MotA/B. As readers of this paper may be familiar with that work, the authors could consider providing additional clarification about why MotB appears to differ from what people are used to seeing (i.e. that domain is not in past work).

Figure 5 – What is the rationale for labeling PDB 6YKM labeled as MotA5 rather than MotA5B2 (which is what is contained in 6YKM)?

Text that labeled Fig 3F was too small.

Style and clarity. The authors' style is fully their choice, but if they would be willing to consider... my opinion is that some of the flowery language obscured the scientific advance and made the work less accessible to a broad audience. Language choices may have also prevented the authors themselves from identifying underlying organizational and grammatical problems that decreased clarity.

Reviewer #3

(Remarks to the Author)

I have reviewed this manuscript previously for another journal. The current version is substantially refocused on the strengths of the earlier manuscript and provides a compelling description of the *C. jejuni* flagellar assembly, its evolution and function. The structures revealed by the in situ approach are fascinating and clearly described, the strengths and limitations of the data are also very clearly described. The manuscript seems highly suitable for publication in *Nature Microbiology* as is.

Susan M. Lea, Frederick, MD, USA.

Decision Letter:

19th November 2024

Dear Morgan,

Thank you very much for your patience while your manuscript "Molecular model of a bacterial flagellar motor in situ reveals a "parts-list" of protein adaptations to increase torque" was under peer-review at Nature Microbiology. It has now been seen by 3 referees, whose expertise and comments you will find at the end of this email. Although they find your work of some potential interest, they have raised a number of concerns that will need to be addressed before we can consider publication of the work in Nature Microbiology.

In particular, you will see that while Referee #3 is satisfied with the manuscript, Referee #2 raises some questions over the alignment and fitting of HtrA with PflC, asks whether it is possible to make further comparisons with structures from low torque systems to strengthen conclusions regarding adaptations that affect torque, and asks for bioinformatic analyses to identify whether PflC and PflD are more widespread. There are also additional points raised by both Referee #2 and Referee #1 which we also feel are important points that should be addressed for us to further consider the manuscript. The rest of the referees' reports are clear and the remaining issues should be straightforward to address.

Should further experimental data allow you to address these criticisms, we would be happy to look at a revised manuscript.

Please include a data availability statement as a separate section after Methods but before references, under the heading "Data Availability". This section should inform readers about the availability of the data used to support the conclusions of your study. This information includes accession codes to public repositories (data banks for protein, DNA or RNA sequences, microarray, proteomics data etc...), references to source data published alongside the paper, unique identifiers such as URLs to data repository entries, or data set DOIs, and any other statement about data availability. At a minimum, you should include the following statement: "The data that support the findings of this study are available from the corresponding author upon request", mentioning any restrictions on availability. If DOIs are provided, we also strongly encourage including these in the Reference list (authors, title, publisher (repository name), identifier, year). For more guidance on how to write this section please see: <http://www.nature.com/authors/policies/data/data-availability-statements-data-citations.pdf>

* If you have not done so already we suggest that you begin to revise your manuscript so that it conforms to our Article format instructions at <http://www.nature.com/nmicrobiol/info/final-submission>. Refer also to any guidelines provided in this letter.

When submitting the revised version of your manuscript, please pay close attention to our [href="https://www.nature.com/nature-portfolio/editorial-policies/image-integrity">Digital Image Integrity Guidelines](https://www.nature.com/nature-portfolio/editorial-policies/image-integrity) and to the following points below:

Link Redacted

Note: This url links to your confidential homepage and associated information about manuscripts you may have submitted or be reviewing for us. If you wish to forward this e-mail to co-authors, please delete this link to your homepage first.

Nature Microbiology is committed to improving transparency in authorship. As part of our efforts in this direction, we are now requesting that all authors identified as 'corresponding author' on published papers create and link their Open Researcher and Contributor Identifier (ORCID) with their account on the Manuscript Tracking System (MTS), prior to acceptance. This applies to primary research papers only. ORCID helps the scientific community achieve unambiguous attribution of all scholarly contributions. You can create and link your ORCID from the home page of the MTS by clicking on 'Modify my Springer Nature account'. For more information please visit www.springernature.com/orcid.

If you wish to submit a suitably revised manuscript we would hope to receive it within 6 months. If you cannot send it within this time, please let us know. We will be happy to consider your revision, even if a similar study has been accepted for publication at Nature Microbiology or published elsewhere (up to a maximum of 6 months).

Yours sincerely,

Reviewer Expertise:

Referee #1: cryo-EM flagellar structure and function

Referee #2: cryo-EM flagellar structure and function

Referee #3: structural biology, flagella

Reviewer Comments:

Reviewer #1 (Remarks to the Author):

This is a very nice piece of work revealing the in situ flagellar motor structure of *Campylobacter jejuni* in sufficient detail to allow identification of component proteins by docking AlphaFold predicted models into the cryoEM density map. The authors have done a good job producing minicells of *Campylobacter jejuni* at minimal sizes close to 200 nm so that they can carry out cryoEM single particle analysis of the flagellar motor structure, instead of tomography data collection and subtomogram averaging as usually done for in situ structural analysis, to successfully achieve a sub nanometer resolution. The density map is impressively visualizing the secondary structures or domains of many of the component proteins to allow unambiguous docking of predicted models. Although the depth of biological and functional insights into the mechanism of torque generation by the flagellar motor is limited due to the limited resolution, this study still reveals much greater detail of the flagellar motor structure of *Campylobacter jejuni* than those obtained in previous studies by tomography of normal cells to advance our understanding of the structural and functional variations of the flagellar motor between bacterial species.

The authors also faithfully revised their original manuscript submitted to other journal according to the comments by reviewers, and their responses are satisfactory. I have a few minor points listed below for the final revision of the manuscript.

1. Lines 135-137: The authors predict that each subsequent rings of FlgP incorporates 11 additional protomers, but this could be examined by applying +11-fold symmetry to the 17-fold of the innermost ring. Has this been tried?
2. Lines 142 and 336: "second beta-sheet" should be "second strand of the beta-sheet".
3. Paragraph from line 308: A reference should be cited for "approximately three times the torque of the E. coli and Salmonella motors". Also, if the structure of C. jejuni flagellar motor can explain the three times higher torque, the authors should give a quantitative mechanical explanation by referring to the number of stator units and the diameter of the C-ring.
4. Paragraph from line 341: The consistent rotational register of the MotA pentamer to the C-ring edge could also be interpreted as all the MotA pentamers are bound to the C-terminal torque helix of FliG. This possibility should be considered and discussed here.
5. Line 353: It is not clear what is meant by "our projection imaging approach".
6. Line 358: "such as the MS-ring"?
7. Lines 367-368: "provide invaluable information on the mechanisms of torque generation" seems to be an overstatement.
8. Lines 369-371: It may be good to insert the words "without sectioning" after "subnanometer resolution" to make is a bit more precise.

Keiichi Namba

Reviewer #2 (Remarks to the Author):

The manuscript "Molecular model of a bacterial flagellar motor in situ reveals a parts-list of protein adaptations to increase torque," from the Beeby lab uses a delta-flhG mutation to produce large numbers of C jejuni flagellar motors in mini-cells, then determines the cryoEM structures. This is a major tour de force and provides the highest resolution view of a fully intact flagellar motor in situ. Part of this work discovers roles for two previously unassigned proteins, now FlgC/D. The introduction sets up biological questions about how the C jejuni motor achieves high torque and how motors might have evolved. The authors also appear to want to discuss accompanying technological advances. This turns the results section into a true 'parts-list' that

describes the different regions of the motor, the methods to collect data, and the strategy used to assign and interpret the maps. The authors seem to run out of space and don't squarely address the biological questions that they pose in the intro. I recognize that it is a challenge to pick and choose when there is a massive amount of information and a short report format. Despite these concerns and the detailed comments below, I think that this manuscript is well suited to a broad audience and could help explain differences in chemotaxis behavior between different motile bacteria. The structure itself is an absolutely fantastic accomplishment. Most concerns directly or indirectly relate to the lack of focus of the text in the context of a short format, which makes this major advance feel smaller than it actually is.

Points for the authors to consider.

HtrA. The HtrA analysis felt tangential to the main manuscript and lacks clarity on whether the authors are proposing convergent or divergent evolution. Moreover, the figures do not convincingly illustrate the authors' claims of a shared fold and some of the depictions do not match the legend. For instance, in the ED5A legend, the term "local sequence realignments" suggests manual manipulation, which absolutely compromises a structure-based alignment. Closer inspection identifies that this almost certainly cannot be a proper structure-based sequence alignment. Specifically, ED5C shows the HtrA secondary structural elements containing the catalytic residues, i.e. a helix containing the catalytic His and a loop containing the Ser. These secondary structure elements appear to be absent in the overlaid PflC in the same ED5C panel. But panel ED5A shows the HtrA sequences aligned to something in the PflC sequence as if the catalytic residues are missing due to a point mutation. The text similarly discusses these as if it is a point mutation. Overall, the ED5A alignment seems forced, dividing into two non-homologous groups; there are limited sequence matches, excessive gapping, and no information on percent identity or similarity. This could indicate convergent evolution to a stable fold. If so, one could expect relatively poor sequence identity with helices and strands in the same order, but with conserved anchor residues necessary for this fold. An example is in Fig 1 of PMID 8880921, which is an early analysis of convergent evolution of Ig folds. Visual assessment of the structural alignment and side-by-side structures makes it appear that these have little in common beyond two helices and a size/shape similarity. It is difficult to tell for certain because 3D, ED5B, and ED5D are incredibly small. The pair-wise RMS deviation of $>10 \text{ \AA}$ would also support these not sharing a fold. While the pruned similarity may indicate a shared fold, the authors do not specify the pruning level, residue count, or show the elements remaining after pruning, making it hard to evaluate potential over-pruning. If the analysis is retained, more supporting information would be necessary. The sequence alignment needs to be fully structure-based and not additionally manipulated. Anchor residues are the most critical part of the analysis of convergent evolution and need to be marked in the alignment, secondary structural elements need to be labeled in numerical order on both the sequence alignment and the structural panels, the size of the panels that show structural comparisons need to be increased, and the pruned core fold from each comparator needs to be shown side-by-side.

Given that the authors indicate that their major biological question was about evolutionary adaptations that affect torque, the rationale for not including more detailed analyses that leverage available structures of different parts of the low-torque motors as comparators was not clear. Are the major differences between this motor and lower-torque motors the larger gear size and a buffering piece on the membrane? If so, this reduces the number of changes that an organism needs to make to move from low- to high-torque, particularly when it can co-opt the buffer via horizontal transfer. And it's obviously easier to move from high- to low-torque. The only comparisons shown were in tomogram side views, which lack the detail needed to make these interpretations.

MotA/B. The description of MotA/B in the text is confusing and the accompanying figures currently don't do a good job of illustrating the authors' points. What is meant by rotational register? Do the authors mean that the asymmetry of the pentameric MotA, as is induced by the central MotB, always aligns so that structurally similar face engages FliG? If so, this is not only unclear but is expected given the information in the literature. MotA rotates around MotB during transmembrane proton/Na transfer in a way that is conceptually analogous to the ATP synthase, as concomitantly described by both the Lea and Taylor groups (ref 9/10) and as supported by the isolated and symmetric MotA structure (PMID 36179499). If the authors do mean that the same conformational face of MotA/B engages the C-ring across the particle, this would be the first time that it is directly shown around the intact ring because the (unmentioned) MotA/B-FliG structure from the Lea group (ref 8) only use a resected FliG domain to show that a consistent conformational face of MotA interacting with FliG. This might just be a text and presentation issue.

The manuscript is under-referenced throughout, and sometimes the findings of the cited papers are overgeneralized. As an example (not limited to this, but this is the first one in the text), the introduction cites refs 7/8 for the isolated C-ring, but the isolated C-ring citations should be reference 8 and PMID 39179739. Ref 7 does contain an isolated C-ring but also contains the combined MS- and C-ring. Reference 8 contains the MotA/B-FliG structure, and the unreferenced PMID 39179739 also contains CheY. Following the introduction, the text only uses reference 8 when referring to the C-ring but all three citations are appropriate each time. Figure legends are fully unreferenced.

Many figure legends did not adequately describe the figure components, sometimes instead stating the authors' interpretations or conclusions. This made it difficult to understand many figure panels. For example, Fig. 3A appears to show a cross-section of the motor perpendicular to the membrane. The legend states: "The medial disk is situated between the basal disk and proximal disk."

Along the same lines, many figures contain red boxes, but it is not always clear what those are. One can eventually figure out that these point to another panel, but the view isn't always the same and there is not clarification when this view is rotated. The first example is Fig 2A, which has a red dotted box labeled 'B'. There is a red box in panel B that appears to be on the same protein, but a different view and different part of the ring (or rotated -90° around the y axis?). The rationale for red boxes in 2F is unclear, as is the reasons that FliG is on a different scale, and that the proteins are not shown in the same order they are discussed in the text.

Many figure panels were too small to see.

The main text accompanying Fig 2I implies that the 1-2Å difference is statistically significant at 8Å resolution without coordinates assigned (although the legend indicates these are similar). This should be within error. See PMID 33559604.

PflC/D are described as proteins of previously unknown function, with the discovery of their role reported here. I find this quite interesting. Can bioinformatics identify how widespread these are? Are there homologs in other organisms, are these common in genomes of most motile bacteria, or are these unique to high-torque motors? How does this inform on the question of torque? Would this predict a low- versus high-torque bacterium?

The FlgQ analysis also feels tangential. It basically involved experiments with negative results and the authors indicate that FlgQ was not observed in this structure.

If molecular models are shown in figures, the standard in the field is that PDB files are deposited at www.rcsb.org. These would be limited to alpha-carbons or polyalanines given the reported resolution. For the authors consideration, both a full ring and an individual unit could be separately deposited, with the latter essential in making these results useful to the scientific community. Currently, only EMDB and EMPIAR codes are given in the manuscript. Anyone wishing to make comparisons to this structure would need to do an independent de novo chain trace and map interpretation. The full ring is important for less experienced users who will have trouble building this from a single unit. Given the number of domain swaps, an individual unit is also not all that helpful.

In the title and throughout, the authors indicate that they are investigating torque. Most of the text revolves around speed and power. Torque is a vector, so a nod to the directional component could be appropriate in the discussion.

It is not clear from the text that only the top half of the structure is assigned with models. The decision to prioritize this aspect of the assignment may have been pragmatic due to resolution, but it is also a reasonable choice for prioritization because the newly assigned section is unique as compared to high-resolution structures of component parts. Nevertheless, adding text clarifying what is and is not assigned with molecular models is important in the context of the title and introduction, which indicates that the manuscript investigates ways that *C. jejuni* will "increase" torque. But the models, and most of the text, focuses on the adaptations that function to buffer (not increase) torque. The power-generating part of the motor, which actually increases the torque, does not appear to be assigned. It doesn't necessarily need molecular assignment as the analysis about gear size stands without it, but the concept could be better highlighted. I am hoping that the resolution of the maps is sufficient to allow the authors to complete the model assignment for a future publication; it would be amazing to see the full machine some day.

Because of the way that most figure panels are deconstructed to highlight one of the authors' points, in some cases, it can be hard to get a sense of where the specific features are within the intact motor. I realize that some of the figure panels make a strong attempt at providing this context and it may not be possible to improve on this. Is it reasonable to include a few global views of the maps with different rotations and an overview in the ED that is an intact motor highlighting what each main text figure will focus on. Also is it reasonable to have a map in the ED that is colored by resolution?

When referring to other structures, PDB codes are not associated with their citations throughout the manuscript. In some cases, it was not clear whether models were alphaFold predictions or experimentally determined structures. For example, lines 186-189 describe PflC using language normally reserved for experimental structures, but the authors also report that they have discovered the role of this protein and I don't see either PflC or cj1643 when searching the PDB. No citation is given. If these are alphaFold predictions, did the authors use a confidence cutoff? Also, if alphaFold, the program should be referenced or the DOI given if the prediction is among those in the alphaFold database.

Line 234 and Figure 4. This is quite hard to interpret likely because the color used for MotB blends into its surroundings and it is not highlighted. Because of this, it took some time to find MotB in the figure. Also, the periplasmic domain is not associated with the past cryoEM structures of isolated MotA/B. As readers of this paper may be familiar with that work, the authors could consider providing additional clarification about why MotB appears to differ from what people are used to seeing (i.e. that domain is not in past work).

Figure 5 – What is the rationale for labeling PDB 6YKM labeled as MotA5 rather than MotA5B2 (which is what is contained in 6YKM)?

Text that labeled Fig 3F was too small.

Style and clarity. The authors' style is fully their choice, but if they would be willing to consider... my opinion is that some of the flowery language obscured the scientific advance and made the work less accessible to a broad audience. Language choices may have also prevented the authors themselves from identifying underlying organizational and grammatical problems that decreased clarity.

Reviewer #3 (Remarks to the Author):

I have reviewed this manuscript previously for another journal. The current version is substantially refocused on the strengths of the earlier manuscript and provides a compelling description of the *C. jejuni* flagellar assembly, its evolution and function. The structures revealed by the in situ approach are fascinating and clearly described, the strengths and limitations of the data are also very clearly described. The manuscript seems highly suitable for publication in *Nature Microbiology* as is.

Susan M. Lea, Frederick, MD, USA.

Version 1:

Reviewer comments:

Reviewer #1

(Remarks to the Author)

The authors adequately revised the manuscript in response to my comments, except for just one point. It would be nice to add an explicit quantitative explanation in the discussion paragraph starting from line 370 on the three times torque of the E. coli and Salmonella motors based on the radial position and number of stator units of the C. jejuni motor.

Reviewer #2

(Remarks to the Author)

Even though my original comments were extensive, this is absolutely fantastic work and a pleasure to read - both the original and the revision. I was absolutely wowed by the sheer magnitude of this impressive accomplishment and hope to see the remainder of the assignment in the future. I appreciate the clarifications in the text and response to reviewers - and particularly the deposition of PDB files. Because the clarity is enhanced in many places, the revision could be read reasonably quickly. I think the paper will offer something to both junior researchers and experts in the field. I can't wait to cover this in our journal club course!

Decision Letter:

Our ref: NMICROBIOL-24092874A

19th February 2025

Dear Dr. Beeby,

Thank you for submitting your revised manuscript "Molecular model of a bacterial flagellar motor in situ reveals a "parts-list" of protein adaptations to increase torque" (NMICROBIOL-24092874A). It has now been seen by the original referees and their comments are below. The reviewers find that the paper has improved in revision, and therefore we'll be happy in principle to publish it in Nature Microbiology, pending minor revisions to satisfy the referees' final requests and to comply with our editorial and formatting guidelines.

Thank you again for your interest in Nature Microbiology Please do not hesitate to contact me if you have any questions.

Sincerely,

Reviewer #1 (Remarks to the Author):

The authors adequately revised the manuscript in response to my comments, except for just one point. It would be nice to add an explicit quantitative explanation in the discussion paragraph starting from line 370 on the three times torque of the E. coli and Salmonella motors based on the radial position and number of stator units of the C. jejuni motor.

Reviewer #2 (Remarks to the Author):

Even though my original comments were extensive, this is absolutely fantastic work and a pleasure to read - both the original and the revision. I was absolutely wowed by the sheer magnitude of this impressive accomplishment and hope to see the remainder of the assignment in the future. I appreciate the clarifications in the text and response to reviewers - and particularly the deposition of PDB files. Because the clarity is enhanced in many places, the revision could be read reasonably quickly. I think the paper will offer something to both junior researchers and experts in the field. I can't wait to cover this in our journal club course!

Version 2:

Decision Letter:

11th April 2025

Dear Dr Beeby,

I am very pleased to accept your Article "In situ structure of a bacterial flagellar motor at subnanometre resolution reveals adaptations for increased torque" for publication in Nature Microbiology. Thank you for having chosen to submit your work to us and many congratulations.

Authors may need to take specific actions to achieve [compliance](https://www.springernature.com/gp/open-research/funding/policy-compliance-faqs) with funder and institutional open access mandates. If your research is supported by a funder that requires immediate open access (e.g. according to [Plan S principles](https://www.springernature.com/gp/open-research/plan-s-compliance)) then you should select the gold OA route, and we will direct you to the compliant route where possible. For authors selecting the subscription publication route, the journal's standard licensing terms will need to be accepted, including [self-archiving policies](https://www.nature.com/nature-portfolio/editorial-policies/self-archiving-and-license-to-publish). Those licensing terms will supersede any other terms that the author or any third party may assert apply to any version of the manuscript.

To assist our authors in disseminating their research to the broader community, our SharedIt initiative provides you with a unique shareable link that will allow anyone (with or without a subscription) to read the published article. Recipients of the link with a

subscription will also be able to download and print the PDF.

With kind regards,

P.S. Click on the following link if you would like to recommend Nature Microbiology to your librarian
<http://www.nature.com/subscriptions/recommend.html#forms>

** Visit the Springer Nature Editorial and Publishing website at http://editorial-jobs.springernature.com?utm_source=ejP_NMicro_email&utm_medium=ejP_NMicro_email&utm_campaign=ejp_NMicro for more information about our career opportunities. If you have any questions please click [here](mailto:editorial.publishing.jobs@springernature.com).

Reviewer Expertise:

Referee #1: cryo-EM flagellar structure and function

Referee #2: cryo-EM flagellar structure and function

Referee #3: structural biology, flagella

Reviewer Comments:

Reviewer #1 (Remarks to the Author):

This is a very nice piece of work revealing the in situ flagellar motor structure of *Campylobacter jejuni* in sufficient detail to allow identification of component proteins by docking AlphaFold predicted models into the cryoEM density map. The authors have done a good job producing minicells of *Campylobacter jejuni* at minimal sizes close to 200 nm so that they can carry out cryoEM single particle analysis of the flagellar motor structure, instead of tomography data collection and subtomogram averaging as usually done for in situ structural analysis, to successfully achieve a sub nanometer resolution. The density map is impressively visualizing the secondary structures or domains of many of the component proteins to allow unambiguous docking of predicted models. Although the depth of biological and functional insights into the mechanism of torque generation by the flagellar motor is limited due to the limited resolution, this study still reveals much greater detail of the flagellar motor structure of *Campylobacter jejuni* than those obtained in previous studies by tomography of normal cells to advance our understanding of the structural and functional variations of the flagellar motor between bacterial species.

The authors also faithfully revised their original manuscript submitted to other journal according to the comments by reviewers, and their responses are satisfactory. I have a few minor points listed below for the final revision of the manuscript.

1. Lines 135-137: The authors predict that each subsequent rings of FlgP incorporates 11 additional protomers, but this could be examined by applying +11-fold symmetry to the 17-fold of the innermost ring. Has this been tried?

We have tried this. First, though, some background to explicitly describe why +11 subunits per ring best accounts for the progressive increase in radius. By extrapolating from intersubunit spacing of the first 51-subunit, 286 Å-diameter ring, we predict that the tenth, 150-subunit ring, would have a diameter of 841 Å, which is a close match to the observed 849 Å. It is also much better than +10 or +12 (a 58 Å difference of the +10 subunits per ring prediction of a tenth, 141-subunit ring of 791 Å, or a 43 Å difference of the +12 subunits per ring prediction of a tenth, 159-subunit ring of 892 Å diameter).

This suggests two scenarios: (1) the rotational phase relationship of two rings is independent of adjacent rings, or (2) that there is a deterministic architecture to the entire disk that could be discernible in a C1 reconstruction.

To tackle option (1), we would need to independently refine each ring separately, which we were unsure was possible given the low signal in the basal disk: FlgP subunits are 11.5kDa, are separated by only 2 nm, and lack prominent features to assist alignment. To assess whether our goal was possible, we first performed a positive control of testing whether we could refine the innermost, 51-subunit ring *ab initio*. We masked around this first ring and refined in C1, but were unable to align particles, and produced a blurred

ring. We also attempted this while imposing C50, C51, and C52 symmetries to see if we could retrieve the known C51 symmetry, but these refinements produced only blurred rings, reproducing their imposed periodicity, and lacking secondary structural features, indicating that all three were symmetrisation artefacts. We concluded that there is insignificant signal in FlgP to align it, and our ability to resolve the 51 copies of FlgP in our published map is because the alignment relied upon the considerably larger 17-fold-symmetric PflABC structures. Therefore it is not possible to query option (1) because individual rings cannot be refined in isolation in our current dataset.

To tackle option (2), we calculated a C1 map. While this was successful, we still could not resolve distinct FlgP protomers in any ring except the first. Unfortunately this is inconclusive: this is consistent with lack of deterministic architecture, but is also consistent with a deterministic architecture that lacks sufficient signal to drive alignment for structure determination.

It's worth noting that the rings may also be heterogeneously structured between motors. While the discrepancy between our predicted tenth-ring diameter of 841 Å versus the observed 849 Å diameter may derive from accumulated errors, it may be described by a more complicated scenario of non-uniform additional subunits per ring – indeed, adding an average of +11.15 subunits would recapitulate the observed 849 Å diameter. If this is the case, it may be due to sporadic deterministic addition of one additional subunit (for example +11,+11,+11,+11,+12,+11,+11,+11,+12 between rings). Alternatively, there may be a stochastic likelihood of addition of an additional subunit. Indeed, a back-of-envelope experiment in which there is a 15% chance that each ring can insert +12 subunits instead of a default of +11 predicts an average tenth ring of 849 Å diameter, albeit with 8 Å standard deviation – i.e., this suggests that wider-diameter rings will appear radially smeared, which is difficult to assess, given that the disk has some inherent flexibility and wider-diameter rings are inevitably of lower resolutions.

Because these considerations are somewhat esoteric, we prefer to leave them for a future disk-specific publication instead of further complicating our already dense manuscript! We have, however, added a sentence succinctly summarising this: “*Attempts at focused refinement of discrete rings failed, presumably due to insufficient signal from the 11.5kDa FlgP subunits separated by only 2 nm, and lacking prominent features that would assist alignment.*”

2. Lines 142 and 336: “second beta-sheet” should be “second strand of the beta-sheet”.

Good catch. Apologies. We have corrected this, and taken the opportunity to simplify and clarify minor aspects of the figure and text.

3. Paragraph from line 308: A reference should be cited for “approximately three times the torque of the E. coli and Salmonella motors”. Also, if the structure of C. jejuni flagellar motor can explain the three times higher torque, the authors should give a quantitative mechanical explanation by referring to the number of stator units and the diameter of the C-ring.

This suggestion aligns with suggestions from Reviewer #2. To these suggestions we have therefore restructured and expanded our Introduction section to fully introduce the biological and theoretical basis for our statement that the C. jejuni motor produces three

times the torque of the *E. coli* and *Salmonella* motors. We summarise and explicitly reference the quantitative calculations.

4. Paragraph from line 341: The consistent rotational register of the MotA pentamer to the C-ring edge could also be interpreted as all the MotA pentamers are bound to the C-terminal torque helix of FliG. This possibility should be considered and discussed here.

Good suggestion – we have added text describing this possibility in this Discussion paragraph.

5. Line 353: It is not clear what is meant by “our projection imaging approach”.

Apologies – we have replaced this with the clearer, “our single particle analysis approach”.

6. Line 358: “such as the MS-ring”?

Apologies for this ambiguous wording – we *did* resolve the symmetry of the C-ring, but using subtomogram averaging. We could *not* resolve the symmetry of the C-ring using our main single particle analysis approach. We have clarified this by editing the passage to read, “*but we did not resolve symmetries for other substructures such as the C-ring (which instead required a conventional subtomogram averaging approach)*”.

7. Lines 367-368: “provide invaluable information on the mechanisms of torque generation” seems to be an overstatement.

We’ve reworded this by replacing the word “invaluable” with “new”.

8. Lines 369-371: It may be good to insert the words “without sectioning” after “subnanometer resolution” to make is a bit more precise.

This is a helpful suggestion that we have applied to the text. Thank you.

Keiichi Namba

Thank you, Keiichi!

Reviewer #2 (Remarks to the Author):

The manuscript “Molecular model of a bacterial flagellar motor in situ reveals a parts-list of protein adaptations to increase torque,” from the Beeby lab uses a delta-flhG mutation to produce large numbers of C jejuni flagellar motors in mini-cells, then determines the cryoEM structures. This is a major tour de force and provides the highest resolution view of a fully intact flagellar motor in situ. Part of this work discovers roles for two previously unassigned proteins, now FlgC/D. The introduction sets up biological questions about how the C jejuni motor achieves high torque and how motors might have evolved. The authors also appear to want to discuss accompanying technological advances. This turns the results section into a true ‘parts-list’ that describes the different regions of the motor, the methods to collect data, and the strategy used to assign and interpret the maps. The authors seem to run out of space and don’t squarely address the biological questions that they pose in the intro. I recognize that it is a challenge to pick and choose when there is a massive amount of

information and a short report format. Despite these concerns and the detailed comments below, I think that this manuscript is well suited to a broad audience and could help explain differences in chemotaxis behavior between different motile bacteria. The structure itself is an absolutely fantastic accomplishment. Most concerns directly or indirectly relate to the lack of focus of the text in the context of a short format, which makes this major advance feel smaller than it actually is.

Thanks for your encouraging words. We have absolutely struggled to articulate all that we wanted to, but many of your comments were invaluable in addressing some of our blind spots.

We have streamlined the Introduction and Discussion to more coherently describe our questions and the insights into them that our study yielded, which revolve primarily around “*Understanding how and why these additional proteins [found in the C. jejuni flagellar motor] were incorporated into the motor, and how they contribute to function*”.

Points for the authors to consider.

(Where appropriate we have broken and continued paragraphs with “//” to address individual points.)

HtrA. The HtrA analysis felt tangential to the main manuscript and lacks clarity on whether the authors are proposing convergent or divergent evolution. //

For us, the PfIC:HtrA homology is one of the most exciting finds of our study, although your comments rightly highlight that our treatment was overly superficial. While in-depth coverage of the PfIC:HtrA homology is the subject of another paper currently in preparation, we have added substantial analyses on the homology of PfIC and HtrA based on your suggestions to the current manuscript. Point-by-point responses below.

// Moreover, the figures do not convincingly illustrate the authors' claims of a shared fold and some of the depictions do not match the legend. //

We have reworked the relevant figures (Fig. 3, and Extended Data Figs. 7 and 8), including new views, explicit description of our methodology, a depiction of the core residues involved in structural alignment, a topology schematic, and a figure illustrating RMSD as a function of residue. Together we hope this convincingly argues for both structural similarity and homology, as well as an illustration of the divergence of the structure of the active site region.

For instance, in the ED5A legend, the term “local sequence realignments” suggests manual manipulation, which absolutely compromises a structure-based alignment. Closer inspection identifies that this almost certainly cannot be a proper structure-based sequence alignment. Specifically, ED5C shows the HtrA secondary structural elements containing the catalytic residues, i.e. a helix containing the catalytic His and a loop containing the Ser. These secondary structure elements appear to be absent in the overlaid PfIC in the same ED5C panel. But panel ED5A shows the HtrA sequences aligned to something in the PfIC sequence as if the catalytic residues are missing due to a point mutation. The text similarly discusses these as if it is a point mutation. Overall, the ED5A alignment seems forced, dividing into two non-homologous groups; there are limited sequence matches, excessive gapping, and no information on percent identity or similarity. This could indicate convergent evolution to a stable fold. If so, one could expect relatively poor sequence identity with helices and strands in the same order, but with conserved anchor residues necessary for this fold. An example is in Fig 1 of PMID 8880921, which is an early analysis of convergent evolution of Ig folds.

Visual assessment of the structural alignment and side-by-side structures makes it appear that these have little in common beyond two helices and a size/shape similarity. It is difficult to tell for certain because 3D, ED5B, and ED5D are incredibly small. The pair-wise RMS deviation of >10 Å would also support these not sharing a fold. While the pruned similarity may indicate a shared fold, the authors do not specify the pruning level, residue count, or show the elements remaining after pruning, making it hard to evaluate potential over-pruning.
//

We have revised the text and figures extensively to address your concerns. As we detail fully below, both sequence and structural comparisons make it clear that PflC is homologous with HtrA. We have removed the multiple sequence alignment of many sequences because this goes beyond the scope of the current manuscript, and restricted comparison of *C. jejuni* PflC to *C. jejuni* HtrA (for simplicity, we arbitrarily choose to restrict to sequences within the same organism– the phylogeny of PflC and HtrA is complicated by their considerable divergence, which we prefer to defer for a future manuscript). Therefore we show pairwise alignment of PflC_N to HtrA_N, and PflC_C to HtrA_C, concatenated into a single panel for clarity. We have included information on percent identity, pruning level, and the core fold after pruning, and we have enlarged figures appropriately. We hope that these figures convince that there is substantially more in common than two helices and a size and shape similarity.

In the text we have removed mention of point mutations and made it clear that the entire region around the active site has atrophied.

You are justified in saying that we did not provide good evidence of homology, which conventionally would be ascertained by sequence comparisons. We therefore include additional supporting evidence as you suggested in your next comment, informed by the Bateman, Eddy, and Chothia approach. More below. We have *not* included an analysis of ‘anchor’ residues, as we consider our new analysis compelling, and it is our understanding that anchor residues remain poorly understood and/or controversial (see, e.g., Nooney, C., Gusnanto, A., Gilks, W. R. & Barber, S. *Do protein structures evolve around ‘anchor’ residues?* in *Geometry Driven Statistics* 311–336 (John Wiley & Sons, Ltd, 2015). doi:[10.1002/9781118866641.ch16](https://doi.org/10.1002/9781118866641.ch16).) (We also didn’t follow your sentence, “An example is in Fig 1 of PMID 8880921, which is an early analysis of convergent evolution of Ig folds” – Chothia argues that these sequences are homologous [i.e, divergent], not analogous [i.e., convergent].)

// If the analysis is retained, more supporting information would be necessary. //

We have added two different sequence-based searches that together strongly argue for PflC and HtrA homology:

1. Although our discovery that PflC is similar to HtrA was initially from a structure-based search with DALI against the PDB, the updated structural comparison that follows this discovery in the manuscript is based on the Needleman-Wunsch global sequence alignment initially generated by the ChimeraX matchmaker tool. We have removed the previous alignment and now include this global alignment. This global alignment is used by ChimeraX’s Matchmaker tool to align corresponding C α atoms. The resulting excellent structural alignment of the pruned core would not be possible if this sequence alignment resulted from chance alignments. (from the manuscript” “*Sequence-guided structural alignment of PflC_N to C. jejuni HtrA yielded alignment of a core of 118 pruned residue pairs*

(of 216) with RMSD of 2.448 Å (Extended Data Fig. 7A,B). Because PflC_C is connected to PflC_N by a flexible linker, we aligned it separately to the C-terminal PDZ domain of HtrA with RMSD between 42 pruned residue pairs (of 64) of 1.425 Å”)

2. The Bateman, Eddy, and Chothia paper is based on the premise that distantly-related homologous sequences can be detected by techniques that are fine-tuned to the peculiarities of conservation in a specific family. In that paper they constructed a hidden Markov model (HMM) based on a pre-existing alignment. This paper predated the advent of PSI-BLAST, which dynamically iteratively constructs a position-specific substitution matrix based on an input sequence. With each iteration, the matrix becomes more sensitive to distant relatives, and the matrix is thus refined. We used this approach to build a multiple alignment starting with the PflC sequence, and we subsequently used this multiple sequence alignment to construct an HMM. Both steps gave strong statistical support for PflC and HtrA homology: using *C. jejuni* PflC as the seed for a PSI-BLAST search against the NCBI clustered non-redundant protein sequence database (nr_clustered, 5th Dec 2024) using default settings immediately yielded hits to sequences annotated as serine proteases with e-values as small as 5×10^{-59} on the first iteration. We took the top hits (250, default setting) from the 4th iteration of PSI-BLAST (each iteration using the default top 500 hits) to build an HMM using hmmbuild from the hmmer 3.3 package. We used this HMM with hmmsearch to search protein sequences from *C. jejuni* 81-176 (ASM1552v1), which returned four significant hits: PflC (CJJ81176_1634, e-value of 2.4×10^{-108}), HtrA (CJJ81176_1242, e-value of 3×10^{-57}), a zinc protease (CJJ81176_1086, with e-value of 6.7×10^{-08}), and a carboxyl-terminal protease (CJJ81176_0539, e-value of 3.2×10^{-06}). PflC and HtrA are composed of a protease domain followed by two PDZ domains; the two lower-scoring hits only have one PDZ domain after their protease domain.

Although these sequence-based findings are compelling, there are also three other similarities consistent with shared ancestry: PflC and HtrA have the same three-domain protease:PDZ:PDZ arrangement strung along the same polypeptide, both are periplasmically co-localised, and both form higher-order oligomers. While far from compelling in isolation, these commonalities further argue for common ancestry, and we have incorporated this reasoning into the manuscript. (While it is most parsimonious to speculate that the last common ancestor was a protease instead of a flagellar component, the large differences between the sequences means that the root falls between the two subfamilies, rendering inference of ancestry intractable using our current approaches.)

// The sequence alignment needs to be fully structure-based and not additionally manipulated. Anchor residues are the most critical part of the analysis of convergent evolution and need to be marked in the alignment, secondary structural elements need to be labeled in numerical order on both the sequence alignment and the structural panels, the size of the panels that show structural comparisons need to be increased, and the pruned core fold from each comparator needs to be shown side-by-side.

The ultimate test of homology vs. analogy is sequence-based, and as highlighted in our responses above, our structural alignment approach is now *sequence* based, and not additionally manipulated.

We have added a pruned core fold to Extended Data Figure 7B, and highlighted which residues comprised the pruned core fold in the pairwise alignment in Extended Data Figure 7A.

We have added Extended Data Fig. 8 to include an analysis of topology, with secondary structural elements labelled as you suggest on both the protein structure, and also a topology schematic. We think that this offers a satisfactory demonstration of common topology.

Given that the authors indicate that their major biological question was about evolutionary adaptations that affect torque, the rationale for not including more detailed analyses that leverage available structures of different parts of the low-torque motors as comparators was not clear. Are the major differences between this motor and lower-torque motors the larger gear size and a buffering piece on the membrane? If so, this reduces the number of changes that an organism needs to make to move from low- to high-torque, particularly when it can co-opt the buffer via horizontal transfer. And its obviously easier to move from high- to low-torque. The only comparisons shown were in tomogram side views, which lack the detail needed to make these interpretations.

Yes, the major differences between this motor and low-torque motors are the larger gear size (the C-ring) and the recruitments of the proteins (FlgP, PflABCD) that form the periplasmic scaffold. (While the basal disk is indeed a “buffering piece”, we prefer to refer to PflABCD as a scaffold that positions additional stator complexes at a wider radius). These structures are absent from low-torque motors, so comparison is limited to our current description of presence in *C. jejuni* vs. absence in low-torque motors. As such, there are no available structures to compare to.

It is true that this reduces the number of changes an organism needs to make to move from low- to high-torque, although this conclusion predates this manuscript and I have discussed it previously in Beeby et al, PNAS 201518952 (2016). (<https://doi.org/10.1073/pnas.1518952113>).

Regarding your observation that “*The only comparisons shown were in tomogram side views, which lack the detail needed to make these interpretations.*”, we did include relatively superficial such comparisons of C-ring and MS-ring architecture in a previous iteration of this manuscript, but reviewers believed it too speculative and unimportant to merit conclusions. There is little more that we can say beyond the fact that the *C. jejuni* C-ring is 38-fold symmetric while the *E. coli* / *Salmonella* C-ring is 34-fold symmetric.

We have refined our text text to confirm that the major core differences between this motor and lower-torque motors is the larger gear size, but that also the additional proteins are essential to scaffold the wider ring of (more) stator complexes to enable continued engagement of the stator complexes with the C-ring.

MotA/B. The description of MotA/B in the text is confusing and the accompanying figures currently don't do a good job of illustrating the authors points. What is meant by rotational register? Do the authors mean that the asymmetry of the pentameric MotA, as is induced by the central MotB, always aligns so that structurally similar face engages FliG? If so, this is not only unclear but is expected given the information in the literature. MotA rotates around MotB during transmembrane proton/Na transfer in a way that is conceptually analogous to the ATP synthase, as concomitantly described by both the Lea and Taylor groups (ref 9/10) and as supported by the isolated and symmetric MotA structure (PMID 36179499). If the authors do mean that the same conformational face of MotA/B engages the C-ring across the

particle, this would be the first time that it is directly shown around the intact ring because the (unmentioned) MotA/B-FliG structure from the Lea group (ref 8) only use a resected FliG domain to show that a consistent conformational face of MotA interacting with FliG. This might just be a text and presentation issue.

Yes, the text was not as clear as it could have been. I have extensively rewritten that section and split the relevant paragraph in two to fully unpack the argument. I have removed the term “rotational register” in favour of a more descriptive prose, and added substantially to the logical exposition.

We do indeed mean “*that the asymmetry of the pentameric MotA, as is induced by the central MotB, always aligns so that structurally similar face engages FliG*”, and I hope that the text is now clearer. I do not, however, agree that this is *entirely* expected from the literature. Yes, on the one hand “*MotA ‘rotates’ around MotB during transmembrane proton/Na transfer*” (‘rotates’ is my emphasis because I maintain there is zero experimental evidence for this elegant and appealing hypothesis – although we all agree that MotA assembles around two MotB helices), but on the other hand the dominant cogwheel model argues strongly for the orientation of MotA to be dictated by its meshing with the cogteeth of the C-ring.

I hope you’ll find the new text clearer in explanation, and be accepting of my questioning the dominant cogwheel model. I may be wrong, but that is the essence of science. My description tries to couch the argument is sufficient uncertainty for a broad audience to appreciate that we are not outright rejecting the cogwheel model.

The manuscript is under-referenced throughout, and sometimes the findings of the cited papers are overgeneralized. As an example (not limited to this, but this is the first one in the text), the introduction cites refs 7/8 for the isolated C-ring, but the isolated C-ring citations should be reference 8 and PMID 39179739. Ref 7 does contain an isolated C-ring but also contains the combined MS- and C-ring. Reference 8 contains the MotA/B-FliG structure, and the unreferenced PMID 39179739 also contains CheY. Following the introduction, the text only uses reference 8 when referring to the C-ring but all three citations are appropriate each time. Figure legends are fully unreferenced.

Apologies – we had not included PMID 39179739 because we originally submitted our manuscript (long) before that paper was published. We have added citations accordingly. *All of the C-ring papers feature an MS-ring (whether they model it or not), albeit in some cases with small truncations. The C-ring cannot assemble correctly without the MS-ring.*

We have added references to all figure legends – thank you and apologies for this omission.

Many figure legends did not adequately describe the figure components, sometimes instead stating the authors’ interpretations or conclusions. This made it difficult to understand many figure panels. For example, Fig. 3A appears to show a cross-section of the motor perpendicular to the membrane. The legend states: “The medial disk is situated between the basal disk and proximal disk.”

Apologies – we have gone through all legends and rewritten those sections that did not adequately describe the respective figure components.

Along the same lines, many figures contain red boxes, but it is not always clear what those are. One can eventually figure out that these point to another panel, but the view isn’t always

the same and there is not clarification when this view is rotated. The first example is Fig 2A, which has a red dotted box labeled 'B'. There is a red box in panel B that appears to be on the same protein, but a different view and different part of the ring (or rotated -90° around the y axis?). The rationale for red boxes in 2F is unclear, as is the reasons that FlgT is on a different scale, and that the proteins are not shown in the same order they are discussed in the text.

We have added full descriptions of the identities of the red boxes, and the relative orientations of their referenced panels across all figures.

(We have also tidied up descriptions of proteins that share the FlgP fold in the body text and figure legend, including FlgT description.)

Many figure panels were too small to see.

Figure 3 was the main offender for too small figures. I have enlarged the entire figure and rearranged panels to remedy this. In my view, other figures do not share this problem.

The main text accompanying Fig 2I implies that the 1-2Å difference is statistically significant at 8Å resolution without coordinates assigned (although the legend indicates these are similar). This should be within error. See PMID 33559604.

Agreed that these are within error, and as the legend indicates, we view these as similar/comparable. We have not altered the text for this, as I think we have been clear that these are comparable.

PfIC/D are described as proteins of previously unknown function, with the discovery of their role reported here. I find this quite interesting. Can bioinformatics identify how widespread these are? Are there homologs in other organisms, are these common in genomes of most motile bacteria, or are these unique to high-torque motors? How does this inform on the question of torque? Would this predict a low- versus high-torque bacterium?

A truly excellent question...

We've been focused on just these questions on PfIC for the last two years. Unfortunately, the answer is "it's complicated", and although our recent findings do not contradict or problematise anything we report in the current manuscript, we'd prefer not to add anything more to the already bulky current manuscript.

In brief, PfIC appears to be widely present in the Campylobacterota *but* it does not always form a lattice-like structure as we report for *C. jejuni*. We have preliminary results confirming that it is still a motor component in species as distal as the *Helicobacters*, but it forms a much simpler structure. PfIC, therefore, was not simply recruited to the Campylobacterota motors, but rather was first recruited (long ago), and subsequently underwent extensive remodelling to become the PfIC we see in *C. jejuni* (more recently). At what point PfIC became incorporated into the motor needs to be interrogated by further comparative studies of more distal species, but this is a substantial project in its own right.

The FlgQ analysis also feels tangential. It basically involved experiments with negative results and the authors indicate that FlgQ was not observed in this structure.

Yes, it is indeed a negative result (and appears tangential), but we've been writing about FlgQ for many years now, so (if only for our own sense of completion), we really want

to retain this negative result in the manuscript. We also suspect that it will make sense some day (as we write: we wonder if FlgQ may be a chaperone, based on its structural similarity to FlgP? Time will tell, we hope...).

If molecular models are shown in figures, the standard in the field is that PDB files are deposited at www.rcsb.org. These would be limited to alpha-carbons or polyanines given the reported resolution. For the authors consideration, both a full ring and an individual unit could be separately deposited, with the latter essential in making these results useful to the scientific community. Currently, only EMDB and EMPIAR codes are given in the manuscript. Anyone wishing to make comparisons to this structure would need to do an independent de novo chain trace and map interpretation. The full ring is important for less experienced users who will have trouble building this from a single unit. Given the number of domain swaps, an individual unit is also not all that helpful.

We have now deposited the α -carbon coordinates of the atomic model of the basal disk and periplasmic scaffold to the PDB with public accession code 9HMF, and included references to this under the **Data Availability** section. We deposited the coordinates of an individual asymmetric unit along with appropriate symmetry operators to produce the C17 scaffold, and linked the entry to the corresponding map on EMDB (EMD-16724).

In the title and throughout, the authors indicate that they are investigating torque. Most of the text revolves around speed and power. Torque is a vector, so a nod to the directional component could be appropriate in the discussion.

The directionality of the torque vector is used to define the plane of rotation (i.e., the plane that the vector is perpendicular to). Given that the plane of rotation is universally understood to be perpendicular to the rod, and given that our target audience are microbiologists, we think defining the fundamentals of torque may only confuse the text, and would prefer to omit it. Readers who do not have a strong background in physics will intuitively understand torque as the magnitude of the vector; readers with a background in physics will not need our explanation.

We carefully avoided using the word “power” throughout the text.

It is not clear from the text that only the top half of the structure is assigned with models. The decision to prioritize this aspect of the assignment may have been pragmatic due to resolution, but it is also a reasonable choice for prioritization because the newly assigned section is unique as compared to high-resolution structures of component parts.

Thanks - we have explicitly clarified this in the manuscript: *“Given that these symmetry-mismatched regions comprise universally-conserved structures such as the P-, L-, MS-, and C-rings, and the stator complexes, which have already been structurally characterised by purification from model organisms, we focused on the 17-fold symmetric novel structures from C. jejuni.”*

Nevertheless, adding text clarifying what is and is not assigned with molecular models is important in the context of the title and introduction, which indicates that the manuscript investigates ways that C jejuni will "increase" torque. But the models, and most of the text, focuses on the adaptations that function to buffer (not increase) torque. The power-generating part of the motor, which actually increases the torque, does not appear to be assigned. It doesn't necessarily need molecular assignment as the analysis about gear size stands without it, but the concept could be better highlighted. I am hoping that the resolution

of the maps is sufficient to allow the authors to complete the model assignment for a future publication; it would be amazing to see the full machine some day.

The maximum torque output of a flagellar motor is cross product of the lever radius (i.e., C-ring radius) and the force applied by a stator complex, all multiplied by the number of stator complexes. (There are many assumptions made in this statement, but it is approximately true – see p. E1924 of PMID 26976588 for full exposition.)

The power generating part of the motor (the stator complexes and C-ring) do not in themselves increase torque – rather, it is the assembly of the stator complexes. As such, the basal disk and periplasmic scaffold (“buffer”) *do* facilitate increased torque: presuming the stator complexes exert the same force in all organisms, the periplasmic scaffold both increases the lever radius at which stator complexes apply force to the C-ring, *and* increases the number of stator complexes that assemble into the motor. We have previously argued that *C. jejuni* assembles 17 stator complexes that are statically anchored into the motor, in contrast to *E. coli* and *Salmonella*, whose stator complexes assemble as a function of motor load.

We therefore stand by our text, but appreciate that this could have been articulated more clearly in the introduction. We have revised that section of the text to read, “*By increasing the number of stator complexes, and increasing the radius at which the stator complexes exert leverage to rotate the C-ring, these adaptations increase the magnitude of the motor’s torque output*”.

Because of the way that most figure panels are deconstructed to highlight one of the authors' points, in some cases, it can be hard to get a sense of where the specific features are within the intact motor. I realize that some of the figure panels make a strong attempt at providing this context and it may not be possible to improve on this. Is it reasonable to include a few global views of the maps with different rotations and an overview in the ED that is an intact motor highlighting what each main text figure will focus on. Also is it reasonable to have a map in the ED that is colored by resolution?

We liked your suggestion of a figure globally contextualising what each main text figure focused on. This has now been added as Extended Data Figure 2.

When referring to other structures, PDB codes are not associated with their citations throughout the manuscript. In some cases, it was not clear whether models were alphaFold predictions or experimentally determined structures. For example, lines 186-189 describe PflC using language normally reserved for experimental structures, but the authors also report that they have discovered the role of this protein and I don't see either PflC or cj1643 when searching the PDB. No citation is given. If these are alphaFold predictions, did the authors use a confidence cutoff? Also, if alphaFold, the program should be referenced or the DOI given if the prediction is among those in the alphaFold database.

The structure of PflC is an AlphaFold2 prediction, and we agree this was not sufficiently clear in the highlighted lines. To avoid confusion, we have now clarified this in the main text, and describe modelling of all chains in detail in Methods.

Line 234 and Figure 4. This is quite hard to interpret likely because the color used for MotB blends into its surroundings and it is not highlighted. Because of this, it took some time to find MotB in the figure. Also, the periplasmic domain is not associated with the past cryoEM structures of isolated MotA/B. As readers of this paper may be familiar with that work, the

authors could consider providing additional clarification about why MotB appears to differ from what people are used to seeing (i.e. that domain is not in past work).

Thanks. We have now highlighted both MotB_N and MotB_C (i.e., panel 4C features both MotB_N and MotB_C, while the inset panel only features MotB_C), and elaborated in the main text for readers to better understand the context, including a sentence highlighting that poor resolution of MotB_C in our structure “*This is consistent with the MotB_C domain not being resolved in recently structures of purified MotA₅B₂ complexes.*”

Figure 5 – What is the rationale for labeling PDB 6YKM labeled as MotA₅ rather than MotA₅B₂ (which is what is contained in 6YKM)?

We have now explicitly explained that this is MotA₅ from 6YKM with the N-terminal helices of MotB removed. (The motivation for this figure is to highlight the thimble-like structure of MotA₅.)

Text that labeled Fig 3F was too small.

Very true – we’ve extensively reformatted Fig. 3 to full-page, remedying this problem and other aspects of poor visual communication.

Style and clarity. The authors' style is fully their choice, but if they would be willing to consider... my opinion is that some of the flowery language obscured the scientific advance and made the work less accessible to a broad audience. Language choices may have also prevented the authors themselves from identifying underlying organizational and grammatical problems that decreased clarity.

Together with all of the above we have reworked and streamlined as best we can – I hope that this is evident upon rereading the manuscript.

Reviewer #3 (Remarks to the Author):

I have reviewed this manuscript previously for another journal. The current version is substantially refocused on the strengths of the earlier manuscript and provides a compelling description of the *C.jejuni* flagellar assembly, its evolution and function. The structures revealed by the in situ approach are fascinating and clearly described, the strengths and limitations of the data are also very clearly described. The manuscript seems highly suitable for publication in *Nature Microbiology* as is.

Susan M. Lea, Frederick, MD, USA.

Thank you, Susan!

Reviewer #1 (Remarks to the Author):

The authors adequately revised the manuscript in response to my comments, except for just one point. It would be nice to add an explicit quantitative explanation in the discussion paragraph starting from line 370 on the three times torque of the E. coli and Salmonella motors based on the radial position and number of stator units of the C. jejuni motor.

Thanks. We've now added an additional section in the second paragraph of the Discussion as you suggest.

Reviewer #2 (Remarks to the Author):

Even though my original comments were extensive, this is absolutely fantastic work and a pleasure to read - both the original and the revision. I was absolutely wowed by the sheer magnitude of this impressive accomplishment and hope to see the remainder of the assignment in the future. I appreciate the clarifications in the text and response to reviewers - and particularly the deposition of PDB files. Because the clarity is enhanced in many places, the revision could be read reasonably quickly. I think the paper will offer something to both junior researchers and experts in the field. I can't wait to cover this in our journal club course!

Thank you!